# INFERENCE-TIME DIFFUSION MODEL ALIGNMENT VIA RANDOM ORDINARY EQUATIONS

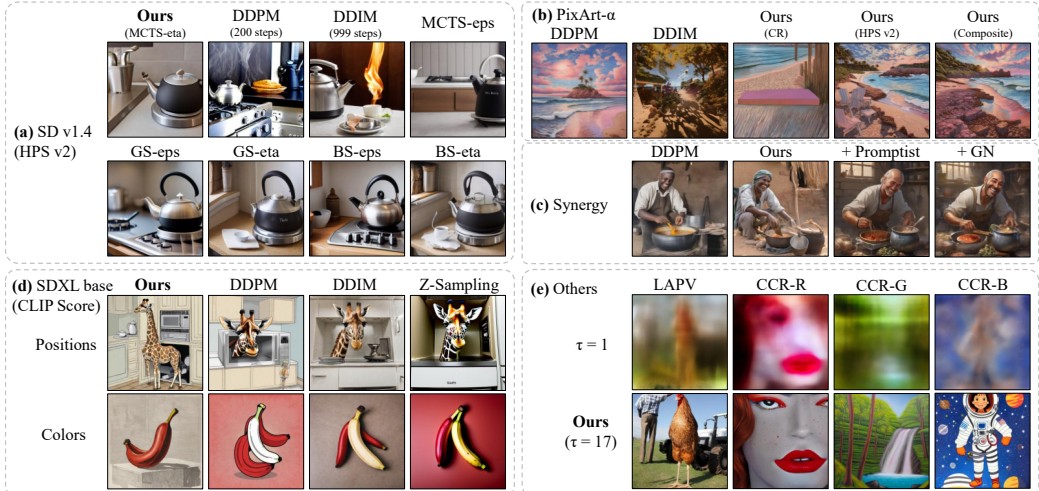

Figure 1: Our method successfully aligns diverse pre-trained diffusion models with various reward functions, such as aesthetics (a), semantics (d), compressibility (b), and sharpness and colors (e). It can also synergize with community-provided modules to improve the aesthetics of samples (c).

## ABSTRACT

Aligning diffusion models (DM) with human preferences is a challenging yet practical task. Recent efforts focus on training-free methods, but usually adopt high-dimensional action spaces or require differentiable rewards. To address these issues, we propose a novel inference-time alignment framework based on random ordinary differential equation sampling. Specifically, we first formulate DM alignment as a max-encountered-reward optimal control problem. Then, by fixing the process noise and optimizing the perturbation strength, we obtain a 1-D action space, which integrates naturally with Monte Carlo tree search. We can thus perform trajectory search to derive the optimal control in a gradient-free manner, therefore supporting non-differentiable rewards. We also provide theoretical guarantees and empirical evidence to support and validate our method. Experiments show that our method demonstrates sufficient sample diversity and successfully aligns pre-trained DMs with reward functions defined on clean image domains. Our method outperforms traditional inference-step scaling, achieving higher best rewards. Meanwhile, it has significantly higher parameter efficiency than existing approaches adopting high-dimensional action spaces. Our approach can be plug-and-play integrated into any multi-step inference DMs.

## 1 INTRODUCTION

Diffusion models (DMs) (Ho et al., 2020; Song et al., 2021) can fit the data distribution, but often misaligns with human preferences, such as aesthetics (Fan et al., 2023), compressibility (Black et al., 2023), and colors (Eyring et al., 2024). To overcome this challenge, researchers (Lee et al., 2023; Black et al., 2023) model the denoising process as a Markov decision process (MDP), and train reward models to guide the training of DMs, thereby aligning them with specific rewards. But these approaches often require intensive training. A more favorable avenue is *inference-time scaling*, which employ inference-time guidance or optimization methods to achieve training-free DM alignment (Liu et al., 2024). However, these methods usually necessitate differentiable reward

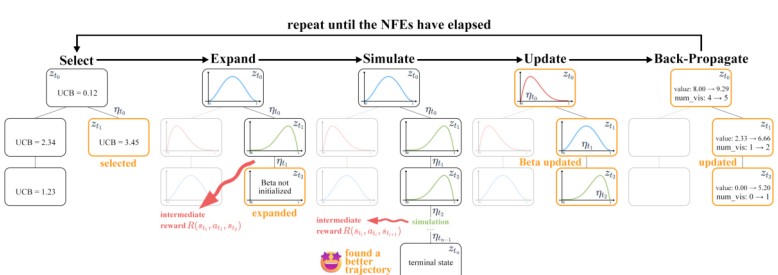

Figure 2: **Illustration of our MCTS.** The better trajectories found and the Beta policy updates in this figure occurred in the simulation phase. Remind that updates can also occur during the expansion phase.

functions (Bansal et al., 2023) and involve costly back-propagation (Eyring et al., 2024). Recently, researchers have turned to gradient-free methods to search noise or latents for inference-time alignment (Liu et al., 2024; Oshima et al., 2025), which, however, are always parameter-inefficient due to high-dimensional action spaces (Sec. 5.4). Besides, previous work also suffers from sparse rewards during the denoising process, resulting in low data efficiency (Zhang et al., 2024a).

To address these issues, we propose a novel inference-time scaling framework that alters the denoising trajectory to align DMs with any off-the-shelf reward function (unnecessarily differentiable). Specifically, our framework starts from a random ordinary differential equation (RODE) (Han et al., 2017) sampling extracted from DDIM (Song et al., 2020), which has lower variance and introduces controllable randomness to ODE sampling for exploration (Sec. 3.2). Then, we model the denoising process as a deterministic episodic MDP with a 1-D action space and dense rewards (Sec. 3.3), and formulate inference-time DM alignment as a max-encountered-reward optimal control problem (Quah & Quek, 2006) (Sec. 3.1, Sec. 3.4). Finally, we employ a value-based Monte Carlo tree search (MCTS) (Coulom, 2006; Browne et al., 2012) with online-updated Beta polices (Fig. 2) to derive the optimal control (Fleming & Rishel, 2012), thus achieving DM alignment (Sec. 4).

In experiments, we first verify that RODE sampling offers reasonable, sufficient and controllable diversity (Sec. 5.2). Then, we demonstrate that our combination of latent reward shaping, max-reward modeling and max-value update outperforms traditional ones (Sec. 5.3). Next, we compare different methods on aligning DMs to various reward functions, such as aesthetics (Sec. 5.4), semantics (Sec. 5.5) and compressibility (Sec. 5.6), highlighting the advantage of our MCTS with augmented RODE-based scaling. Finally, we present ablations and applications of our method (Sec. 5.7).

Our contributions: (i) We propose a novel parameter-efficient inference-time scaling framework for aligning DMs with any Lipschitz continuous reward functions (unnecessarily differentiable) by altering the denoising trajectory; (ii) **We are the first to** extract an RODE sampling from the DDIM's SDE sampling, and model the denoising process as a MDP with a **1-D action space** and dense rewards; (iii) **We are the first to** formulate the inference-time DM alignment as a **max-encountered-reward optimal control problem**, and solve it using MCTS with augmented RODE-based scaling; Our approach can be plug-and-play integrated into any multi-step inference DMs.

## 2 PRELIMINARIES

### 2.1 DDIM

A stochastic DDIM (Song et al., 2020) step writes

$$z_{t_{i+1}} = \sqrt{\overline{\alpha}_{t_{i+1}}}\hat{z}_{t_n} + \sqrt{1 - \overline{\alpha}_{t_{i+1}} - \sigma_{t_i}^2}\epsilon_\theta^{(t_i)}(z_{t_i}) + \sigma_{t_i}\epsilon_{t_i}, \tag{1}$$

where the process noise $\epsilon_{t_i} \sim \mathcal{N}(0, I)$ is re-sampled at each timestep $t_i$, contributing stochasticity via the perturbation term $\sigma_{t_i}\epsilon_{t_i}$; and the *posterior mean*

$$\hat{z}_{t_n} \triangleq \mathbb{E}[z_{t_n} \mid z_{t_i}] = (z_{t_i} - \sqrt{1 - \overline{\alpha}_{t_i}}\epsilon_\theta^{(t_i)}(z_{t_i}))/\sqrt{\overline{\alpha}_{t_i}}. \tag{2}$$

The standard deviation $\sigma_{t_i}$ is defined as the product of $\eta_{t_i}$ and a time-dependent constant $\omega_{t_i}$:

$$\sigma_{t_i}(\eta_{t_i}) \triangleq \eta_{t_i}\omega_{t_i} \triangleq \eta_{t_i}\sqrt{(1 - \overline{\alpha}_{t_{i-1}})/(1 - \overline{\alpha}_{t_i})}\sqrt{1 - \overline{\alpha}_{t_i}/\overline{\alpha}_{t_{i-1}}}. \tag{3}$$

$\eta_{t_i}$ is conventionally stipulated to be time-invariant, *i.e.*, $\eta_{t_i} = \eta$ $(0 \leq i < n)$. The value of $\eta$ determines the stochasticity of the DDIM step: $\eta = 0$ makes the update process fully deterministic, referred to as *deterministic DDIM*; When $\eta \in (0, 1]$, the update process incorporates some randomness, termed *stochastic DDIM*. Particularly, it is called *DDPM* (Ho et al., 2020) when $\eta = 1$.

## 2.2 DDPO-Style Modeling

Most existing RL-based DM alignment methods (Black et al., 2023; Fan et al., 2023; Lee et al., 2023; Xu et al., 2023a; Clark et al., 2023; Prabhudesai et al., 2023) are based on the DDPO-style modeling (Black et al., 2023). Specifically, given a timestep schedule $\{t_0, t_1, \cdots, t_n\}$, where $T = t_0 > t_1 > \cdots > t_n = 0$, in which $T$ represents the number of training steps of the DMs, and $n$ denotes the number of inference steps. Meanwhile, given a DM $p_\theta$, and a *clean reward model* $\phi(\cdot)$ (defined on clean image domains). Condition information (*e.g.*, text prompts) is omitted for simplicity. The denoising process of $p_\theta$ is modeled as a multi-step MDP $(\mathcal{S}, \mathcal{A}, P, R, \gamma, K)$, where:

$$s_{t_i} \triangleq (z_{t_i}, t_i), \ \pi(a_{t_i}, s_{t_i}) \triangleq p_\theta(z_{t_{i+1}} \mid z_{t_i}, t_i), \ a_{t_i} \triangleq z_{t_{i+1}}, \ P(s_{t_{i+1}} \mid s_{t_i}, a_{t_i}) \triangleq (\delta_{z_{t_{i+1}}}, \delta_{t_{i+1}})$$

$$R(s_{t_i}, a_{t_i}, s_{t_{i+1}}) \triangleq \begin{cases} \mathbb{E}_{z_{t_n} \sim p_\theta(z_{t_{i+1}})}[\phi(z_{t_n})], & \text{if } i+1 = n \\ 0, & \text{otherwise} \end{cases}, \ \gamma = 1, \ K \triangleq n. \tag{4}$$

Refer to Appx. D.1 for symbol description and more discussion.

Given an initial noise $z_{t_0}$, the objective of RL is to maximize

$$J(\theta) \triangleq \mathbb{E}_{z_{t_n} \sim p_\theta(z_{t_0})} \left[ \sum_{i=0}^{n-1} R(s_{t_i}, a_{t_i}) \right] \xrightarrow{\text{if DDPO}} \mathbb{E}_{z_{t_n} \sim p_\theta(z_{t_0})} [\phi(z_{t_n})]. \tag{5}$$

## 2.3 Latent Reward Shaping

To obtain dense rewards for the denoising process for faster convergence and better performance, training-free *latent reward shaping* utilizes the final reward (obtained with *clean reward models*) as a *proxy* for intermediate rewards (Appx. C.3). Specifically, to estimate the *latent reward* for an intermediate latent $z_{t_i}$ $(0 \leq i < n)$, denoted as $\hat{\phi}(z_{t_i})$, we first estimate the *pseudo-final sample* $\hat{z}_{t_n}$, from $z_{t_i}$ in a certain way $\mathcal{F}$, *i.e.*, $\hat{z}_{t_n} = \mathcal{F}(z_{t_i})$. The estimated reward for $z_{t_i}$ $(0 \leq i \leq n)$ writes

$$\hat{\phi}(z_{t_i}) = \begin{cases} \phi(\mathcal{F}(z_{t_i})) = \phi(\hat{z}_{t_n}), & \text{if } i < n \\ \phi(z_{t_n}), & \text{if } i = n \end{cases}, \tag{6}$$

and the reward function in Eq. 4 is modified to

$$R(s_{t_i}, a_{t_i}, s_{t_{i+1}}) \triangleq \mathbb{E}_{z_{t_{i+1}} \sim p_\theta(\cdot | z_{t_i}, t_i)} \left[ \hat{\phi}(z_{t_{i+1}}) \right]. \tag{7}$$

For example, setting $\mathcal{F}$ as the posterior mean of DDIM, *i.e.*, $\mathcal{F}(z_{t_i}) = \mathbb{E}[z_{t_n} \mid z_{t_i}]$ (Eq. 2), we can derive the *immediate-DDIM* paradigm latent reward shaping. Refer to Appx. D.2 for more details.

# 3 Problem Statement and Our Formulation

## 3.1 Problem Formulation

Given a DM $p_\theta$ and a clean reward model $\phi(\cdot)$, the most straightforward problem formulation is to maximize the rewards of the final samples, *i.e.*, the *final reward* of the MDP, regardless of whether sparse or dense rewards are adopted. In the sparse reward scenario, the final reward is the only available object to optimize, which means the objectives of maximizing *cumulative rewards*, *final rewards*, and *rewards encountered during the process* are equivalent. However, we argue that, the above formulation is sub-optimal when adopting dense rewards (Zhang et al., 2024b;a) due to the inconsistency between intermediate rewards and final rewards, regardless of whether the final rewards are optimized directly (*e.g.*, intermediate rewards as heuristic) or indirectly (*e.g.*, maximizing

cumulative rewards), since intermediate rewards would mislead the search process due to similar scale and *inadmissibility* (Appx. E.1). To circumvent this issue, inspired by the *de novo* drug design (Gummesson Svensson et al., 2024), we treat the pseudo-final samples obtained during the denoising process as valid samples, thereby transforming the problem into maximizing the encountered rewards within an episode, *i.e.*, a *max-encountered-reward (max-reward for short) control/RL problem* (Quah & Quek, 2006; Gottipati et al., 2020) (Appx. D.3). Note that this is different from purely maximizing the final reward, i.e., *max-final-reward modeling*. To the best of our knowledge, we are the first to model DM alignment as a max-reward control/RL problem.

Specifically, we adopt the DDIM schedule. Given an initial noise $z_{t_0}$ and the NFE budget for calculating transition dynamics (*i.e.*, maximum inference steps) $N$, we aim to find the sample $\widetilde{z}_{t_N}$ within $N$ DDIM steps that maximizes $\phi(\widetilde{z}_{t_N})$, where $\widetilde{z}_{t_N}$ is defined as the derived sample $z_{t_N}$ if $N$ DDIM steps are strictly performed sequentially; otherwise, a pseudo-final sample $\widetilde{z}_{t_N} = \mathcal{F}(z_{t_i})$. We do **not** enforce a full use of $N$ steps, because: (i) Sabour et al. (2024); Ye et al. (2024) found that, the fixed hand-crafted timestep schedule might be sub-optimal; (ii) Nichol & Dhariwal (2021); Li et al. (2023) showed that, increasing inference steps might cause sample quality degradation.

**Geometric Interpretation.** Fig. 3 provides a geometric interpretation of the conventional and our formulation. Under the conventional formulation, the MDP transfer graph $G$ of the DM's inference process is a directed tree due to strictly unidirectional and stepwise transitions. Instead, our formulation treats the in-process pseudo-final samples as valid samples, introducing *shortcuts* from states at timestep $t_i \in [t_1, t_{n-2}]$ to the terminal states at timestep

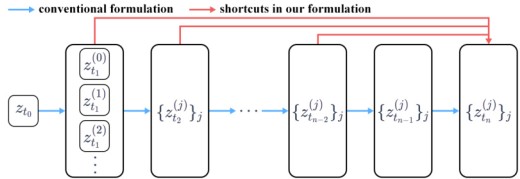

Figure 3: **Geometric interpretation of the conventional and our formulation.** Note that ours contains both blue and red edges.

$t_n$. This transforms $G$ into a directed acyclic graph (DAG) $G'$ (Bang-Jensen & Gutin, 2008), where nodes in each layer of $G$ now have edges to both the next layer and the leaves.

**Enable Pseudo-Final Samples as Valid Samples.** We empirically observe reward hacking in our max-reward modeling due to early pseudo-final samples (Sec. 5.7). To mitigate this, we only consider pseudo-final samples with depth no less than $\tau \in [1, n]$ as valid samples. For samples with depth in the range $[1, \max\{\tau - 1, 1\}]$, we still compute intermediate rewards.

## 3.2 Augmented RODE Sampling for DDIM

Existing inference-time scaling methods suffer from high-dimensional action spaces (Appx. C.2). To overcome this, we propose a random ODE-based (RODE-based) (Han et al., 2017) sampling for DDIM schedule to obtain a low-dimensional action space. It is directly extracted from the pre-trained SDE sampling without extra training or fine-tuning. Specifically, we fix a set of process noises $\{\epsilon_{t_i}\}_{i=0}^{n-1}$ for the DDIM step (Eq. 1), converting SDE sampling to ODE sampling, which is deterministic once given $\{\sigma_{t_i}\}_{i=0}^{n-1}$. Then, we propose to reintroduce randomness to the sampling process by treating the process variances $\{\eta_{t_i}\}_{i=0}^{n-1}$ in Eq. 3 as random variables, which are sampled from some certain probability density functions (PDFs). In the continuous-time perspective, this eliminates the $\mathrm{d}\overline{w}$ term in the standard SDE sampling (Appx. D.4.1), transforming it into an RODE

$$\text{SDE: } \mathrm{d}z = [f(z, t) - g(t)^2 \nabla_z \log p_t(z)]\mathrm{d}t + g(t)\mathrm{d}\overline{w}$$
$$\text{RODE: } \mathrm{d}z = \left[f(z, t) - \omega(t)^2 \eta(t)^2 \nabla_z \log p_t(z)\right] \mathrm{d}t + \omega(t)\eta(t)h(t)\mathrm{d}t \text{,} \tag{8}$$

where $\overline{w}$ is an inverse Wiener process, $\eta(t) \in [0, 1]$ is a scalar stochastic process, and $h(t)$ is the fixed direction field induced by the realized process noise ($\epsilon_{t_k}$ in discrete representation).

Intuitively, our RDOE sampling fixes the perturbation direction at each timestep for all samples, and introduces randomness via random scaling along these directions. This yields lower variance than standard SDE sampling, where the process noise is resampled at each timestep:

**Proposition 1.** *In the discrete analog, the variances of SDE and RODE step writes*

$$Var_{SDE} = \omega_{t_i}^2 \eta^2 \Delta t I, \quad Var_{RODE} = \omega_{t_i}^2 Var(\eta_{t_i})(\Delta t)^2 \epsilon_{t_i} \epsilon_{t_i}^\top, \tag{9}$$

*respectively. Thus, $Var_{RODE} < Var_{SDE}$ in Loewner order (Horn & Johnson, 2012), and both sampling schemes display exponential tail decay and strong concentration around their means.*

Despite lower variance, RODE sampling can provide sufficient sample diversity in multi-step inference scenarios (Sec. 5.2). Besides, let $p^{(S)}$ and $p^{(R)}$ be the distribution induced by SDE and RODE sampling, respectively. $p^{(R)}$ approximates the true data distribution as $p^{(S)}$ does because:

**Proposition 2.** *The Wasserstein-1 distance (Kolouri et al., 2017) $W_1(p^{(S)}, p^{(R)})$ is bounded:*

$$W_1(p^{(S)}, p^{(R)}) \leq \sum_{k=0}^{n-1} M_{t_{k+1}} \left( \omega_{t_k} \cdot C_{\epsilon,t_k} \cdot \mathbb{E}[\Delta \eta_{t_k}] \right) + \mathcal{O}\left( \sum_{k=0}^{n-1} M_{t_{k+1}} \gamma_{t_k} \right), \qquad (10)$$

where: (i) $M_{t_{i+1}} = \prod_{j=i+1}^{n-1} L_{t_j}$ $(0 \leq j \leq n-1)$ is the the product of a sequence of Lipschitz constants $L_{t_j}$ with a uniform upper-bound $L < \infty$. We have $M_{t_{i+1}} \leq e^{LT}$, where $T$ is the total time horizon (Gronwall, 1919); (ii) $C_{\epsilon,t_k} = \mathbb{E}[\|\epsilon_{t_k}\|]$; (iii) $\Delta \eta_{t_k} = |\eta_{t_k} - \eta| \leq 1$; (iv) the score-estimation error at timestep $t_i$ is upper-bounded by a sub-Gaussian perturbation of scale $\gamma_{t_i}$.

Hence, the discrepancy between samples generated from the two paradigms can be controlled, and the $\eta$-bias is the dominant source of discrepancy between RODE and SDE trajectories.

Additionally, reusing the SDE-trained score network $s_\theta(z_t, t) \approx \nabla_z \log p_t(z_t)$ within RODE sampling does **not** significantly increase score-estimation error since $s_\theta(\cdot)$ approximates marginal distributions and is independent of sampling trajectories. Specifically,

**Proposition 3.** *The difference of score-estimation error writes*

$$\left| \mathbb{E}_{p_t^{(R)}} \mathcal{E}(z, t) - \mathbb{E}_{p_t^{(S)}} \mathcal{E}(z, t) \right| \leq L_{\mathcal{E},t} \cdot W_1(p_t^{(R)}, p_t^{(S)}), \qquad (11)$$

where $\mathcal{E}(z, t) = \|s_\theta(z, t) - s(z, t)\|$ is the point-wise score-estimation error at timestep $t$, and $s(z, t)$ is the GT score function. Refer to Appx. F for assumptions, proofs and more discussion.

Prop. 3 is empirically supported by the zero-shot FIDs (Yu et al., 2021; Podell et al., 2023) (Sec. 5.2). Fortunately, in our scenario, RODE sampling aims to introduce randomness for exploration, instead of exactly approximates the true data distribution. Besides, the theoretically larger error in RODE is visually imperceptible, *i.e.*, it is hard to distinguish RODE-sampled results from standard SDE/ODE-sampled ones (*e.g.*, Fig. 1). It indicates that, RODE sampling can still generate reasonable samples under larger error, which we attribute to the robustness of DMs and the drift term dominance (Appx. F.3). Still, we can design the process noise to span the $\epsilon_{t_i}$ space more effectively, enhancing exploration and reducing error compared to SDE sampling. This can be readily done by generating each process noise with distinct seeds, since high-dimensional random vectors are almost always nearly orthogonal to each other (Cai et al., 2013). Our RODE sampling introduces randomness for exploration, while avoiding dealing with the stochastic optimal control problem (Evans, 1983; Fleming & Rishel, 2012) and its notorious high variance (Sec. 3.4).

**Stronger proposition.** Under stronger assumptions, we show mathematically that $p^{(R)}$ and $p^{(S)}$ are sufficiently close (strengthened Prop. 2). Refer to Appx. F.3.3 for more discussion.

**Benefits to the Search Process.** Standard stochastic DDIM sampling adopts $\eta_{t_i} = \eta$ for all $i \in [0, n-1]$, thereby anchoring a data manifold of a single noise-level $\sigma_{t_i}(\eta)$ for each timestep $t_i$ (Eq. 3). In contrast, our method traverses multiple neighboring noise-level manifolds $\sigma_{t_i}(\eta_{t_i}^{(0)})$, $\sigma_{t_i}(\eta_{t_i}^{(1)}), \cdots$ by adapting different $\sigma_{t_i}(\eta_{t_i}^{(j)})$ on every visit to timestep $t_i$, rather than being confined to a single manifold as in Oshima et al. (2025).

**Augmented RODE Sampling.** When only one set of process noise is realized, RODE sampling may under-explore directions that are under-represented (*e.g.*, directions orthogonal to all noise vectors), leading to sub-optimal rewards. To overcome this, we propose *augmented RODE sampling* (Aug. RODE sampling), which adopts several sets of process noise, and retains the optimal result *per sample*. Formally, for an initial noise $z_{t_0}$, we take $m$ independent sets of process noise $\{\{\epsilon_{t_i}^{(j)}\}_{i=0}^{n-1}\}_{j=0}^{m-1}$. We then search for each set and obtain the maximum rewards $\{r^{(j)}\}_{j=0}^{m-1}$, and

take $r^* = \max_j \{r^{(j)}\}_{j=0}^{m-1}$ as the maximum achievable reward for $z_{t_0}$. Intuitively, adopting more sets of process noise can better span the $\epsilon_{t_i}$ space, thus alleviating insufficient exploration.

## 3.3 MDP MODELING

Treating the $\eta_{t_i}$s as low-dimensional actions and their PDFs as policies forms a novel MDP modeling paradigm. We continue the symbols in Sec. 2.2 and Sec. 2.3, and only indicate the modified parts:

(b) The action space $\mathcal{A} = [0, 1]$, in which the policy and the action at time step $t_i$ is modified to

$$\pi(a_{t_i}, s_{t_i}) \triangleq \widetilde{\pi}(\eta_{t_i} \mid z_{t_i}, t_i), \;\; a_{t_i} \triangleq \eta_{t_i}; \tag{12}$$

(c) The transition dynamics $P$ is modified to

$$P(s_{t_{i+1}} \mid s_{t_i}, a_{t_i}) \triangleq p_\theta(z_{t_{i+1}} \mid z_{t_i}, \eta_{t_i}). \tag{13}$$

Although inherent stochastic in $p_\theta$, the transition becomes deterministic when the process noise $\{\epsilon_{t_i}\}_{i=0}^{n-1}$ is fixed as done in RODE sampling. We further specify that $P(s_{t_{n+1}} \mid s_{t_n}, a) = \delta_{s_{t_n}}$ for any action $a$. We subsequently treat $P$ as a deterministic function induced by a Lipschitz continuous model class (Asadi et al., 2018), and use $s_{t_{i+1}} = P(s_{t_i}, a_{t_i})$ for simplicity;

(d) The reward function is modified to

$$R(s_{t_i}, a_{t_i}, s_{t_{i+1}}) = \hat{\phi}(z_{t_{i+1}}). \tag{14}$$

The expectation is omitted due to the fully deterministic transition dynamics and the sole dependence on states (*i.e.*, trajectory-independent) for the rewards. We stipulate $R(s_{t_n}, a, \cdot) = 0$ for any action $a$. The reward function is assumed to be Lipschitz continuous;

(e) The discount factor $\gamma$ is fixed to 1 because: (i) we treat pseudo-final samples as valid ones in the max-reward modeling, thus no discount is needed; (ii) the episodic MDP is finite, so the Bellman update (Appx. D.3) converges without requiring $\gamma < 1$. Refer to Appx. E.2 for more discussion.

## 3.4 OPTIMAL CONTROL

We formulate the training-free inference-time DM alignment as an online planning problem (Efroni et al., 2020). Specifically, let $\eta_{t_0, \cdots, t_{n-1}} = \{\eta_{t_0}, \cdots, \eta_{t_{n-1}}\}$, and the optimization object be $J(\eta_{t_0, \cdots, t_{n-1}})$. Our goal is to find an optimal control sequence $\eta_{t_0, \cdots, t_{n-1}}^*$ that maximizes $J$, *i.e.*,

$$\eta_{t_0, \cdots, t_{n-1}}^* = \underset{\eta_{t_0, \cdots, t_{n-1}}}{\arg\max} \; J(\eta_{t_0, \cdots, t_{n-1}}) \; s.t. \; s_{t_{i+1}} = P(s_{t_i}, a_{t_i}). \tag{15}$$

We keep the DM's parameters unchanged during the optimization process (*i.e.*, as a fixed environment model), and only optimize the policies used to sample $\eta_t$s, which prevents fatal deviations from the data distribution and unreasonable over-optimization for higher rewards (Black et al., 2023; Zhang et al., 2024b). Unlike He et al. (2025), our modeling does **not** require an additional KL divergence term for regularization. In summary, our RODE-based scaling adopts a low-dimensional action space that can be efficiently covered, enabling controllable trajectory modulation and smooth navigation of the continuous action space. Refer to Appx. E.3 and Appx. E.4 for more discussion.

## 4 ONLINE PLANNING USING MCTS

We employ MCTS as a training-free optimal control solver to derive the $\eta_{t_0, \cdots, t_{n-1}}^*$. The overview is presented in Fig. 2. Specifically, to handle the continuous action space, we select node with the highest *value-based* UCB (Sec. 4.1), then expand it with Beta policies (Sec. 4.2). In the expansion and simulation phases, the *unimodal Beta policies* of the tree nodes are updated online if superior trajectories are found. Finally, the roll-out is backpropagated with *max-value policies* (Sec. 4.1).

### 4.1 SELECTION POLICIES

**Value-Based UCB.** We propose a value-based UCB to facilitate the calculation of UCB in continuous action space based on the insight that, the state value function $V(s)$ aggregates the performance

of all sampled actions at state $s$. Specifically, we propose to replace the $Q(s, a)$ in traditional UCB (Appx. D.5) with $V(s)$, and modify the exploration term in terms of $s$. Formally,

$$\text{exploitation}(s) = V(s), \ \ \text{exploration}(s) = \sqrt{\ln N(\text{parent}(s))/N(s)}, \tag{16}$$

where $\text{parent}(s)$ and $N(s)$ denote the parent node and the visit times of state $s$, respectively.

**Max-Value Policies.** In cumulative-reward RL, the $V(s)$ is typically estimated with the average return from all roll-outs passing through $s$, which is updated with the *average-value policy* during the backpropagation phase (Appx. D.5). However, our max-reward modeling leads us to a novel *max-value policy*, where the $V(s)$ is defined and updated as

$$V(s) = \max_{i=0}^{N(s)-1} R_i, \ \ V(s) \leftarrow \max\{V(s), R\}, \tag{17}$$

in which $R$ is the cumulative or maximum return from the expansion or simulation phases.

**Selection Depth Limit.** During the MCTS selection phase, the depth of the node selected for expansion is limited to the range $[0, n']$, where $n' \leq n$. This is based on the observation from Cao et al. (2023); Yang et al. (2023) that, DMs recover the layout of images first, and the details later. We allocate more computational resources to shallower nodes in MCTS, which focuses on early-to-mid denoising steps, enabling more significant changes for potentially higher rewards.

### 4.2 Beta Policy for Continuous Action Space

**Unimodal Beta Policies.** For continuous action spaces, uniform random sampling (Sec. 5.7) or discretization (Chou et al., 2017) can lead to sub-optimal performance. Following Ye et al. (2024), we maintain a unimodal (for policy stability) Beta distribution (Gupta & Nadarajah, 2004) for each MCTS node, and sample actions $\eta_t$ from it, *i.e.*, we adopt Beta policies. To do this, we re-parameterize the shape parameters $\alpha$ and $\beta$ (**not** the variance parameters $\alpha_{t_i}$ and $\beta_{t_i}$ in DMs) of the Beta distribution as $\alpha = 1 + e^a$, $\beta = 1 + e^b$, and maintain the $a, b > 0$ for each MCTS node.

**Mode Re-Parameterization.** The Beta policies should be updated online during the search process, allocating more computational resources to promising actions. Directly update the shape parameters is hard to balance the exploration and exploitation of the Beta policy (Appx. G.2). To overcome this, we note that, the goal of the update of Beta policies is to maximize the likelihood of sampling (near-)optimal actions, and the main probability density of a unimodal Beta distribution is concentrated near its mode. This leads us to update with the mode. Specifically, we re-parameterize the Beta distribution in terms of its mode, and introduce a hyper-parameter $\zeta > 0$ to control its concentration. Specifically, let $\rho = (\alpha - 1)/(\alpha + \beta - 2)$ be the mode, then the $a, b$ are re-parameterized as $a = \ln \zeta$, $b = \ln \zeta + \ln((1 - \rho)/\rho)$. Refer to Appx. G.2 for more discussion.

**Initialization and Online Update.** The policies for sampling actions are initialized to uniform distributions at the beginning, and then updated to unimodal Beta distributions with the first-sampled action. When a trajectory superior to the best-known one is found, the Beta policies from the current node to the root node in the MCTS tree are softly updated (Appx. G.3). Then, the $a$ and $b$ are updated correspondingly. In MCTS, such updates may occur within both the expansion (when expanding to a terminal state) and simulation phases, *i.e.*, we also treat the trajectories obtained via simulation as valid ones for updating Beta policies. Particularly, in our max-reward modeling, updates might also occur during the expansion phase — updated with pseudo-final samples when calculating the intermediate rewards for the expanded nodes. Regardless of where the update occurs, only the tree portion of the trajectory (real tree nodes) will have their Beta policies updated.

## 5 Experiments

### 5.1 Settings and Evaluations

**Settings.** We adopt the following settings unless specified. We generate $N = 2$ images per prompt using $N$ fixed independent initial noise. We fix $m = 3$ sets of process noise for Aug. RODE sampling. For SDE sampling, DDPM adopts $\eta_{t_i} = 1.0$ for all $i \in [0, n-1]$, and is run with $m = 3$

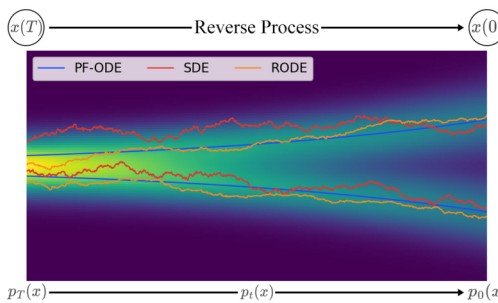 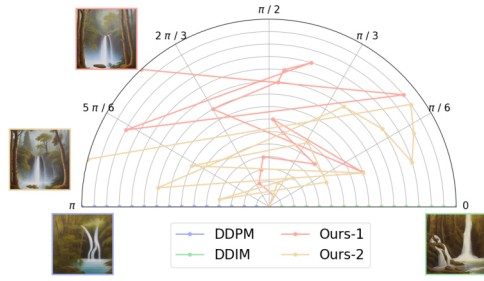

Table 1: **PF-ODE/SDE/RODE trajectories.**

Figure 4: **Visualization of denoising trajectories of 1× DDPM, 1× DDIM, and 2× ours.**

Table 2: **Policies and MPDs (↑) of different inference-time scaling paradigms.** Cells with a red / orange / yellow background: the best / second-best / third-best performance for each column.

| Paradigm | Sampler | Components | | | SD-Turbo | SD v1.4 | | |
| | | $\{z_{t_0}^{(j)}\}_{j=0}^{N-1}$ | $\{\epsilon_{t_i}\}_{i=0}^{n-1}$ | $\{\eta_{t_i}\}_{i=0}^{n-1}$ | 2-step | 15-step | 20-step | 25-step |
|---|---|---|---|---|---|---|---|---|
| **a)** initial-noise | ODE | random | × | $\eta_{t_i} = 0$ | 0.5867 | 0.7542 | 0.7584 | 0.7594 |
| | SDE | random | random | $\eta_{t_i} = 1.0$ | 0.6437 | 0.7653 | 0.7617 | 0.7650 |
| | Aug. RODE | random | $m$ sets | $\eta_{t_i} = 1.0$ | 0.6200 | 0.6861 | 0.6887 | 0.6914 |
| **b)** process-noise | SDE | fixed | random | $\eta_{t_i} = 1.0$ | 0.3823 | 0.6848 | 0.6939 | 0.6946 |
| **c)** process-noise | Aug. RODE | fixed | $m$ sets | $\eta_{t_i} \sim \mathcal{U}([0,1])$ | **0.1628** | **0.4717** | **0.4852** | **0.4899** |
| -variance | Aug. RODE | fixed | $m$ sets | $\eta_{t_i} \sim \mathcal{U}([0.5,1])$ | 0.1295 | 0.3556 | 0.3649 | 0.3686 |

random sets of process noise, and retain the best results across them. In our max-reward formulation, $\tau = \lceil n/3 \rceil$. For MCTS, $n' = \lceil (4n)/5 \rceil$, and simulation actions are uniformly sampled from $[0, 1]$. For Beta policies, $\zeta = 3$. We adopt *immediate-ddim* policy, and the combination of latent reward shaping, max-reward modeling and max-value update for its best performance (Sec. 5.3).

**Evaluations.** We adopt "*Best Reward*" (the highest reward achieved), "*Parameter Efficiency*" and "*Time Cost*" (the averaged wall-clock time required per sample to obtain *Best Reward*) for evaluation. Refer to Appx. H for details on settings, evaluations, and experimental composition.

## 5.2 RODE SAMPLING

**Trajectory Simulation.** Fig. 1 visualizes the sampling trajectories

Table 3: **FIDs (↓) for different sampling paradigms.**

| Paradigm / Step | 15 | 20 | 25 | 30 | 50 |
|---|---|---|---|---|---|
| ODE | 29.4050 | 29.3332 | 29.4872 | 29.5351 | 29.7702 |
| SDE | 25.0427 | 24.9356 | 24.9540 | 24.6273 | 24.6893 |
| Aug. RODE | 28.0226 | 27.2009 | 27.1311 | 27.0817 | 26.9312 |

produced by three paradigms. PF-ODE (Song et al., 2021) yields smooth trajectories, whereas both SDE and RODE generate jagged paths, where the RODE trajectories exhibits markedly smaller oscillations than the SDE ones, empirically supporting Prop. 1. Refer to Appx. I.1 for details.

**Empirical Distribution.** We compute zero-shot FID (Podell et al., 2023) on the MS-COCO 5k validation set (Lin et al., 2014) for differnt sampling paradigms. Results in Tab. 3 demonstrate that, our novel paradigm does **not** excessively deviate from the real data distribution, empirically corroborating Prop. 2. Refer to Appx. I.3 for more details.

**Sample Diversity.** We generate images under the conditions in Tab. 2, and compute the *mean pairwise distance* (MPD ↑) (Appx. H.3.1) to evaluate the sample diversity. It demonstrate that: (i) SDE sampling sustains the highest diversity, while our Aug. RODE sampling exhibits lower sample diversity than SDE sampling, which empirically validated Prop. 1; (ii) Our RODE sampling provides mid-to-high and controllable diversity when $n$ is large, suggesting that, our method is more suitable for multi-step inference scenarios due to broader exploration. Refer to Appx. I.2 for more details.

**Trajectory Visualization.** The sampling trajectory of RODE sampling can be efficiently visualized with $\eta_{t_0,\cdots,t_{n-1}}$ (Appx. I.4). Fig. 4 shows that, RODE sampling explores various paths from the origin to the $n$-th level data manifold (the outmost semi-circle), seeking for higher rewards.

Table 4: **Comparison between different reward shaping, MDP modeling and value update policies.** "*latent*": latent reward shaping; "*sparse*": w/o reward shaping. "*cum.*": cumulative-reward modeling. "*aver.*": average-value update.

| Reward Shaping | MDP Modeling | Value Update | Best Reward |
|---|---|---|---|
| latent | cum. | aver. | 28.9145 |
| latent | cum. | max | 28.8445 |
| latent | max | aver. | 28.8480 |
| latent | max | max | 29.1152 |
| sparse | cum. | aver. | 28.6840 |
| sparse | cum. | max | 29.0621 |

Table 5: **Comparison between different methods when optimizing HPS v2.** "*-eps*" / "*-eta*": $\epsilon_{t_i}$s / $\eta_{t_i}$s as actions. Note that Best-of-N fall **outside** our setting.

| Method | Best Reward | Relative Impro. (%) | Param. Efficiency | Time Cost |
|---|---|---|---|---|
| DDIM (15 steps, baseline) | 26.2122 | / | / | 147 |
| Best-of-N (DDIM) | 28.4056 | 8.37 | 2.03e-6 | 4480 |
| Best-of-N (DDPM) | 28.4297 | 8.46 | 2.05e-6 | 4513 |
| DDIM (999 steps) | 26.5362 | 1.24 | / | 6694 |
| DDPM (200 steps) | 27.6256 | 5.39 | 1.4378e-7 | 1382 |
| GS-eps | 29.7386 | 13.45 | 4.7830e-6 | 5684 |
| BS-eps | 30.2095 | 15.25 | 5.4217e-6 | 5469 |
| **MCTS-eps** | 29.0241 | 10.73 | 3.8139e-6 | 1437 |
| GS-eta | 27.8116 | 6.10 | 0.0355 | 5697 |
| BS-eta | 28.0833 | 7.14 | 0.0416 | 5273 |
| **Ours** | 28.1644 | 7.45 | 0.0434 | 2736 |

## 5.3 Comparison between Reward Shaping, MDP Modeling and Value Policies

This section compares various combinations of reward shaping, MDP modeling, and value update policies. Specifically, we adopt *uniform* expansion policy for all combinations, and align 15-step SD v1.4 (Rombach et al., 2022) to HPS v2 (Wu et al., 2023a). Results in Tab. 4 highlights the advantage of adopting latent reward shaping, max-reward modeling and max-value update.

## 5.4 Aligning with Aesthetics

We align 15-step SD v1.4 (Rombach et al., 2022) to HPS v2 (Wu et al., 2023a), and make comparison between: (i) vanilla 15-step DDIM as baseline; (ii) best-of-N (Ma et al., 2025) with $N = 66$, which is **not** directly comparable as it falls outside our setting, but is included here as a widely used baseline; (iii) the inference-step scaling paradigm (Nichol & Dhariwal, 2021); (iv) beam search (BS) (Oshima et al., 2025) with $\epsilon_{t_i}/\eta_{t_i}$ actions; (v) greedy search (GS, the BS with a single beam); (vi) our MCTS. Tab. 5 demonstrates that: (i) Ours (MCTS-eta) surpasses traditional inference-step scaling. Note that scaling DDPM is essentially an $\epsilon_{t_i}$-action method; (ii) When scaling with $\eta_{t_i}$, ours outperform both GS and BS, highlighting MCTS's ability to allocate NFEs judiciously, and Beta policies' capacity for accurate low-dimensional search; (iii) Although $\epsilon_{t_i}$-action methods achieve the highest absolute score, its relative improvement is only $\sim 2\times$ better than $\eta_{t_i}$-action ones, but with $\sim 16\text{k}\times$ more parameters, revealing severe parameter redundancy; (iv) When naively transferring our MCTS to $\epsilon_{t_i}$-actions, performance drops below GS and BS, underscoring the limitation of vanilla MCTS in high-dimensional continuous action spaces (Bianchi et al., 2023). Qualitative results are shown in Fig. 1 (a), where trajectory search-based methods yield samples that are visually more appealing than those produced by inference-step scaling. Refer to Appx. J for more details.

Besides, our method can generalize to other latent reward policies, see Appx. M.

## 5.5 Aligning with Semantics

We align 30-step SDXL base (Podell et al., 2023) to CLIP score (Hessel et al., 2021), and benchmark our method against DDPM, DDIM and Z-Sampling (Bai et al., 2024). Qualitative results in Fig. 1 (d) are generated with text prompts "A giraffe underneath a microwave." and "A red colored banana.", respectively. It can be observed that, our method renders accurate positional relationship (line 1) and colors (line 2), while baselines fail to capture. Quantitative results are shown in Tab. 6. Refer to Appx. K for more details.

Table 6: **Comparison between different methods when optimizing CLIP Score.**

| Method | Ours | DDPM | DDIM | Z-Sampling |
|---|---|---|---|---|
| CLIP score | 0.3716 | 0.3569 | 0.3328 | 0.3676 |
| Improvement (%) | 11.66 | 7.24 | / | 10.46 |
| HPS v2 | 29.0685 | 28.6943 | 27.8625 | 29.5635 |
| Improvement (%) | 4.33 | 2.99 | / | 6.10 |

Table 7: **Comparison between different optimization objectives.** "*Ours (R)*": aligning with reward $R$ with our method.

| Method | CR (↑) | HPS v2 (↑) | CLIP Score (↑) |
|---|---|---|---|
| Ours (CR) | 2.9308 | 26.7430 | 0.3404 |
| Ours (HPS v2) | 2.9261 | 27.8809 | 0.3731 |
| Ours (Composite) | 2.9282 | 27.7684 | 0.3743 |

## 5.6 ALIGNING WITH COMPOSITE REWARDS

We align 50-step Pixart-$\alpha$ (Chen et al., 2023b) (DiT-architecture (Peebles & Xie, 2023)) to compressibility reward (CR) (Appx. H.3.4). Samples displayed in Fig. 1 (b) are generated with the text prompt "A beautiful blue and pink sky overlooking the beach.", which introduces unreasonable smooth to inflate the reward. To overcome this, we optimize a *composite reward* that combines CR and HPS v2. Qualitative results demonstrate that, optimizing the composite reward yields samples that are perceptually smoother yet retain clear semantics. Quantitative results in Tab. 7 also highlights its balanced performance than optimizing either reward in isolation. Refer to Appx. L for more details.

## 5.7 ABLATIONS AND APPLICATIONS

**Reward Hacking and Effects of $\tau$.** We align 50-step SD v1.4 to Laplacian variance (LAPV) and color channel reward (CCR) (Appx. H.3.4) with $\tau = 17$ (default) and $\tau = 1$ (enabling pseudo-final samples throughout the entire process), respectively. Fig. 1 (e) illustrates the reward hacking phenomena

Table 8: **Ablations.** "*w/o pseudo-final*": disabling pseudo-final samples as valid samples. "*w/o Beta policy*": uniform policy.

| Method | Best Reward |
|---|---|
| **Ours** | 29.2582 |
| Ours w/o pseudo-final | 29.0199 |
| Ours w/o expansion depth limit | 29.1082 |
| Ours w/o online update | 29.1547 |
| Ours w/o Beta policy | 29.1152 |

that leverage the noisy and blurred pseudo-final samples in the early denoising for higher rewards. This can be mitigated by adjusting $\tau$ to determine when to enable treating pseudo-final samples as valid ones. Refer to Appx. N.1 for more details. In summary, our method generalizes across diverse DM architectures, sampling conditions, inference steps, reward functions, *et al.*

**Main Ablations.** We conduct ablations on key components of our method. Results in Tab. 8 underscore advantage of our max-encountered-reward formulation over the conventional max-final-reward formulation. Refer to Appx. N and Appx. P for more ablations and failure cases, respectively.

**Applications.** Our method can synergize with Promptist (Hao et al., 2023) and Golden Noise (GN) (Zhou et al., 2024b) to improve sample aesthetics (Fig. 1 (c), Appx. O.1). Additionally, it can be used to quantitatively assess the robustness of reward models (Appx. O.2).

## 6 RELATED AND CONCLUSION

**Related.** Appx. C presents literature review on DM alignment, inference-time scaling for DMs and latent reward shaping. Our method belongs to training-free alignment via trajectory search.

**Conclusion.** This study makes the first step towards scaling DMs with 1-D actions. We present an augmented RODE-based scaling paradigm, and achieve parameter-efficient inference-time DM alignment by solving a max-reward optimal control problem using MCTS. Our method do **not** necessitate differentiable reward functions, and is particularly suitable for multi-step inference scenarios. Refer to Appx. A for use of LLMs, and Appx. Q for broader impacts and limitations.

REPRODUCIBILITY STATEMENT

Refer to Appx. H for general experimental settings. Detail settings for Sec. 5 are detailed in Appx. I to Appx. P. Refer to *supplementary materials* for code implementation.

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

# Contents

# Appendix

## A    USE OF LARGE LANGUAGE MODELS

We use ChatGPT 4.0 and Grok-Expert to aid and polish writing.

## B    ABBREVIATIONS

Tab. 9 presents abbreviations and their corresponding full names in this paper.

Table 9: **Abbreviations and their corresponding full names.**

| Abbreviation | Full Name |
|:---:|:---:|
| DM | diffusion model |
| RL | reinforcement learning |
| MDP | Markov decision process |
| MCTS | Monte Carlo tree search |
| UCB | upper confidence bound |
| ODE | ordinary differential equation |
| SDE | stochastic differential equation |
| RODE | random ordinary differential equation |
| Aug. RODE sampling | augmented RODE sampling |
| NFE | number of function evaluation |
| MPD | mean pairwise distance |
| LAPV | Laplacian variance |
| CCR | color channel reward |

## C    RELATED

### C.1    DIFFUSION MODEL ALIGNMENT

#### C.1.1    HUMAN PREFERENCE QUANTIFICATION

Human preferences should be quantified first before achieving DM alignment. For example, CLIP Score (Radford et al., 2021; Hessel et al., 2021) is widely adopted to measure the semantic alignment of generated samples. However, some preferences are hard to formulate, for example, aesthetics. To overcome this challenge, researchers turn to collect preference data to train models to embed these preferences implicitly. The most representative ones are the models that predict human aesthetic preferences, such as AesScore (Romain Beaumont, 2022), PickScore (Kirstain et al., 2023), ImageReward (Xu et al., 2023a) and Human Preference Score v2 (HPS v2) (Wu et al., 2023b;a).

#### C.1.2    TRAINING-BASED ALIGNMENT

Training-based DM alignment techniques like Lee et al. (2023); Black et al. (2023); Fan et al. (2023) model the inference process of DMs as a multi-step MDP, and fine-tune or train DMs to align them to the reward functions (existing or trained alongside DMs), such as image aesthetics (Black et al., 2023; Fan et al., 2023; Lee et al., 2023; Xu et al., 2023a; Clark et al., 2023), semantic alignment (Fan et al., 2023; Lee et al., 2023), (in)compressibility (Black et al., 2023; Clark et al., 2023), object detection and removal (Clark et al., 2023), *et al.*

We omit to review a category of training-based DM alignment methods — direct preference optimization (DPO) (Rafailov et al., 2023; Wallace et al., 2024; Yang et al., 2024a; Li et al., 2024a; Yang et al., 2024b; Liang et al., 2024), as they lack explicit rewards, and are incompatible with our framework.

### C.1.3 Training-free Alignment

Training-free techniques freeze the DM's parameters, and achieve alignment by altering its inputs or the denoising trajectory. They can be categorized into gradient-based and gradient-free methods.

**Gradient-based Methods.** ReNO (Eyring et al., 2024) optimizes the initial noise in a one-step inference DM using the gradient of a combination of reward objectives with respect to samples. Besides, Universal Guidance (Bansal et al., 2023) computes the gradient of the loss function with respect to latents, and applying gradient ascent to drive the intermediate latents to high-reward regions. Although training-free, these methods necessitate differentiable reward functions and often limited to few-step or single-step inference DMs, as gradient propagation through DMs or reward models is computationally intractable for multi-step reasoning scenarios (Xu et al., 2023a; Clark et al., 2023; Prabhudesai et al., 2023; Eyring et al., 2024).

**Gradient-free Methods.** To mitigate the necessity of gradient computation, researchers recently turn to gradient-free methods. For example, Z-Sampling (Bai et al., 2024) injects information via the difference in CFG (Ho & Salimans, 2022) scales between denoising and inversion to enhance both aesthetics and semantic alignment, which is not applicable to scenarios that disable CFG (*e.g.*, unconditional generation). Our method likewise belongs to the gradient-free family, operating without requiring differentiable reward functions or CFG.

We refer readers to Liu et al. (2024) for more literature on DM alignment.

### C.2 Inference-time Scaling for Diffusion Models

Inspired by Ma et al. (2025) and Liu et al. (2024), we categorize existing DM's inference-time scaling techniques as follows:

1. *Inference step paradigm*: Boost performance with more inference steps, *e.g.*, IDDPM (Nichol & Dhariwal, 2021) and ADM-IP (Li et al., 2023);

2. *Initial noise selection paradigm*: Use a Best-of-N strategy to pick samples with highest verifier scores from multiple initial noise candidates for better visual quality, diversity, and content consistency (for video generation), *e.g.*, SeedSelect (Samuel et al., 2024), Scaling-Noise (Yang et al., 2025), the noise-level scaling of ReflectionFlow (Zhuo et al., 2025), Ahn et al. (2024), Qi et al. (2024), Samuel et al. (2024), Ma et al. (2025);

3. *Initial noise construction paradigm*: Mao et al. (2024) directly creates initial noise from known winning tickets for each concept mentioned in the prompt;

4. *Initial noise optimization paradigm*: Optimize the initial noise itself (*e.g.*, InitNO (Guo et al., 2024), ReNO (Eyring et al., 2024), Golden Noise (Zhou et al., 2024b), DNO (Tang et al., 2024)) or its sampling distribution (*e.g.*, InitNO (Guo et al., 2024)) via gradient-free (*e.g.*, zero-order search (Ma et al., 2025)) or gradient-based (*e.g.*, first-order search (Ma et al., 2025)) approaches;

5. *Prompt optimization paradigm*: Refine the user-provided text prompts to model-preferred ones that contain richer, finer-grained descriptions (*e.g.*, Reprompt (Wang et al., 2023), Promptist (Hao et al., 2023), OPT2I (Mañas et al., 2024), the prompt-level scaling of ReflectionFlow (Zhuo et al., 2025)), and adaptively adjust their injection timesteps and strengths (*e.g.*, PAE (Mo et al., 2024));

6. *Trajectory search paradigm*: Select multiple process noises as expansion directions at each timestep, and retain the top $N$ candidates. For example, BeamDiffusion (Fernandes et al., 2025) and DBLS (Oshima et al., 2025) utilizes beam search (BS) for sampling trajectory search, where candidates are generated by sampling a set of process noise for stochastic DDIM sampling (Song et al., 2020) at each expanded node. EvoSearch (He et al., 2025) reinterprets the denoising trajectory as an evolutionary path, and mutates both the initial noise an intermediate latents for exploration. Video-T1 (Liu et al., 2025a) proposes Tree of Frame (ToF) Search to efficiently scale video generation, which adaptively expands and prunes branches in autoregressive manner. Ctrl-Z Sampling (Mao et al., 2025) conducts trajectory search with zig-zag operation (Bai et al., 2024). FK Steering (Singhal et al.,

2025) scores and resamples particles according to their potentials during generation to steer them towards high-reward regions;

7. *Trajectory optimization paradigm*: Perform gradient-free information injection along the denoising trajectory (*e.g.*, Z-Sampling (Bai et al., 2024)), or apply gradient-based guidance (*e.g.*, Universal Guidance (Bansal et al., 2023));

8. *Correction paradigm*: The reflection-level scaling of ReflectionFlow (Zhuo et al., 2025) trains a corrector first, then uses it to progressively refine the generated images throughout the inference process.

However, we highlight the following limitations:

1. IDDPM (Nichol & Dhariwal, 2021) and ADM-IP (Li et al., 2023) found that, increasing inference steps yields diminishing FID (Yu et al., 2021) improvement, and can even degrade performance with too many steps;

2. Initial noise selection or construction methods are mainly limited to boost sample quality (*e.g.*, FID (Yu et al., 2021) and aesthetics) and semantic alignment, which is not suitable for broader reward functions. Besides, they adopt a best-of-N procedure, *i.e.*, independently sample multiple initial noise, and discard the sub-optimal ones along the search process. Such strategies, while acknowledging the significant influence of initial noise (Appx. I.2), neglect the contribution from the denoising trajectory, thus fail to fully exploit the generation potential in any single initial noise;

3. Initial noise optimization methods requires costly gradient calculation or approximation, limiting their deployment in multi-step inference scenarios. Besides, they also overlook trajectories;

4. Gradient-free trajectory optimization methods mainly limited to enhance the sample quality and semantic alignment (*e.g.*, Z-Sampling (Bai et al., 2024)), which is always inapplicable to other reward functions;

5. Since initial noise, process noise (*e.g.*, DBLS (Oshima et al., 2025)), and intermediate latents are high-dimensional vector, all methods except *inference step paradigm* adopt high-dimensional action spaces in MDP contexts, leading to low sampling efficiency and high variance.

To overcome the above challenges, we first model the DM's inference process as a MDP with dense rewards, then fix the initial noise and perform efficient trajectory search with MCTS (Coulom, 2006). Our method belongs to the *trajectory search paradigm*. Besides, existing inference-time scaling approaches that adopt high-dimensional action spaces (*e.g.*, initial noise, process noise, intermediate latents) suffer from low parameter efficiency (Sec. 5.4, Appx. J). Recent works (Zhou et al., 2024a; Chen et al., 2023a; 2025a; Wang et al., 2024; Zhu et al., 2025) designate novel distillation techniques with minimal learnable parameters, which inspire us to adopt an extremely low-dimensional action space — 1-D action space — for parameter-efficient scaling.

Besides, Parmar et al. (2025) marks a first step towards scaling a group of samples for better diversity and quality.

**Comparison.** Tab. 10 compares our method against some existing training-based and training-free methods (especially trajectory-based paradigms). Note that our method requires neither differentiable reward functions nor gradient propagation, which is applicable to a wide range of reward functions.

### C.3 LATENT REWARD SHAPING

Most existing alignment techniques adopt the DDPO-style modeling (Black et al., 2023) (Sec. 2.2), which suffers from the sparse reward issue in reinforcement learning (RL), *i.e.*, the reward signals emerge only at the end of the process, which causes unstable training, slow convergence, and inefficient data use (Hare, 2019). On the one hand, adopting sparse rewards lead the training or guidance process to focus too much on the rewards of the final samples, neglecting the contribution from the denoising trajectory. On the other hand, in the practices of existing RL-based DM alignment, methods using dense rewards (Zhang et al., 2024b;a; Ye et al., 2024; Zhang et al., 2025) often outperform

Table 10: **Comparison between popular DM alignment methods and ours.** "*Train.*" denotes whether training or fine-tuning is required; "*Dense.*" denotes whether the reward signals are dense; "*Diff.*" denotes whether necessitating differentiable reward functions; "*Gradient.*" denotes whether gradient propagation through the DMs or the reward models is needed beyond training. Methods that are more user-friendly (less restrictive, more flexible, more computationally tractable) are indicated in green. Particularly, ✓ denotes the need to train both the DM and an additional model; ✓ denotes the need to train an additional model rather than the DM itself; ✓ indicates dense rewards due to single-step inference.

| Methods | Train. | Dense. | Diff. | Gradient. |
|---|---|---|---|---|
| DDPO (Black et al., 2023) | ✓ | ✗ | ✗ | ✗ |
| DPOK (Fan et al., 2023) | ✓ | ✗ | ✗ | ✗ |
| Lee *et al.* (Lee et al., 2023) | ✓ | ✗ | ✗ | ✗ |
| ReFL (Xu et al., 2023a) | ✓ | ✓ | ✓ | 1 step |
| DRaFT-K (Clark et al., 2023) | ✓ | ✓ | ✓ | fixed $K$ steps |
| AlignProp (Prabhudesai et al., 2023) | ✓ | ✓ | ✓ | random steps |
| TDPO-R (Zhang et al., 2024b) | ✓ | ✓ | ✗ | ✗ |
| SDPO (Zhang et al., 2024a) | ✓ | ✓ | ✗ | ✗ |
| Schedule On the Fly (Ye et al., 2024) | ✓ | ✓ | ✗ | ✗ |
| LRM (Zhang et al., 2025) | ✓ | ✓ | ✗ | ✗ |
| ReNO (Eyring et al., 2024) | ✗ | ✓ | ✓ | 1 step |
| DNO (Tang et al., 2024) | ✗ | ✗ | ✓ | $n$ step |
| Universal Guidance (Bansal et al., 2023) | ✗ | ✓ | ✓ | multiple steps |
| DBLS (Oshima et al., 2025) | ✗ | ✓ | ✗ | ✗ |
| EvoSearch (He et al., 2025) | ✗ | ✓ | ✗ | ✗ |
| Video-T1 (Liu et al., 2025a) | ✗ | ✓ | ✗ | ✗ |
| Ctrl-Z Sampling (Mao et al., 2025) | ✗ | ✓ | ✗ | ✗ |
| FK Steering (Singhal et al., 2025) | ✗ | ✓ | ✗ | ✗ |
| **Ours** | ✗ | ✓ | ✗ | ✗ |

those with sparse rewards (Black et al., 2023; Fan et al., 2023; Lee et al., 2023; Xu et al., 2023a; Clark et al., 2023; Prabhudesai et al., 2023). We attribute it to the fact that, under the same computational resources, methods with dense rewards receive more supervision signals in the process.

Recently, researchers have tried designing intermediate rewards for the multi-step DM inference process to obtain dense supervision signals (Zhang et al., 2024b;a; Dhariwal & Nichol, 2021), which is called *reward shaping* (Laud, 2004; Grzes, 2017) in the RL context. The challenge is that, most popular reward models for image synthesis (*e.g.*, FID (Yu et al., 2021), IS (Barratt & Sharma, 2018), LPIPS (Zhang et al., 2018), aesthetics prediction models (Kirstain et al., 2023; Xu et al., 2023a; Romain Beaumont, 2022; Wu et al., 2023b;a)) are trained on clean image domains, which can not be directly transferred to the noisy latent domain to provide reliable intermediate rewards. Although ImageReward (Xu et al., 2023a) found that, rewards calculated from noisy images after 30-th denoising step in a 40-step inference are reliable enough, obtaining rewards for every denoising step, especially in the early stages, is still tough (Xu et al., 2023a; Clark et al., 2023; Prabhudesai et al., 2023; Eyring et al., 2024).

There are training-based and training-free methods that focus on deriving intermediate rewards for the noisy latent domain from reward models trained on clean image domains (we call them *clean reward models*), which we call *latent reward shaping*.

**Training-based Methods.** An immediate thoughts is to train en extra counterpart of the clean reward model in the latent domain to accept noisy inputs, such as classifier guidance (Dhariwal & Nichol, 2021) and LRM (Zhang et al., 2025). Recently, discount-based methods have been developed for latent reward acquisition. For example, TDPO-R (Zhang et al., 2024b) trains an extra model to discount the final rewards into intermediate rewards. Besides, vision language models (VLMs) have also be adopted as reward critics for intermediate rewards (Liu et al., 2025a). However, these

methods demand lots of training, may not generalize across datasets, and require retraining when reward function changes.

**Training-free Methods.** To overcome the limitations of training-based approaches, recent works design training-free paradigms to efficiently obtain intermediate rewards. The key idea of training-free latent reward shaping is to use the final reward as a *proxy* for intermediate rewards. For example, Universal Guidance (Bansal et al., 2023) computes guidance using the *pseudo-final samples* derived from the posterior mean of DDIM (Song et al., 2020). DBLS (Oshima et al., 2025) uniformly interpolates the remaining denoising steps into 2 or 3 DDIM steps for more accurate estimation. Schedule On the Fly (Ye et al., 2024) discounts the final rewards backward to obtain intermediate rewards. SDPO (Zhang et al., 2024a) obtains dense rewards via the cosine similarity between the intermediate latents and those at the initial, final, or other anchor steps. We present a general formulation for training-free latent reward shaping in Sec. 2.3 in the main paper and Appx. D.2.

# D SUPPLEMENTARY TO PRELIMINARY

## D.1 SUPPLEMENTARY TO DDPO-STYLE MODELING

Given a timestep schedule $\mathcal{T} = \{t_0, t_1, \cdots, t_n\}$, where $T = t_0 > t_1 > \cdots > t_n = 0$, in which $T$ represents the number of training steps of the DM, and $n$ denotes the number of inference steps. Meanwhile, given a DM $p_\theta$ parameterized by $\theta$, and a clean reward model $\phi(\cdot)$. Condition information (such as text prompts) is omitted for simplicity. The inference process of the DM is modeled as an MDP $(\mathcal{S}, \mathcal{A}, P, R, \gamma, K)$, where:

(a) The state space $\mathcal{S} \subseteq \mathbb{R}^{c \times h \times w} \times \mathcal{T}$ consists of all timesteps $t_i$ ($0 \leq i \leq n$) and their corresponding latent variables $z_{t_i}$ with varying noise levels. Here, $c, h, w$ are the number of channels, height, and width of the latent variables, respectively. The state at timestep $t_i$ is defined as

$$s_{t_i} \triangleq (z_{t_i}, t_i); \tag{18}$$

(b) The action space $\mathcal{A} \subseteq \mathbb{R}^{c \times h \times w}$ comprises all latent variables $z_{t_i}$ ($0 \leq i \leq n$) across noise levels. The action at timestep $t_i$ is defined as the latent variable corresponding to the noise level of the next timestep $t_{i+1}$. The policy and action are given by

$$\pi(a_{t_i}, s_{t_i}) \triangleq p_\theta(z_{t_{i+1}} \mid z_{t_i}, t_i), \quad a_{t_i} \triangleq z_{t_{i+1}}; \tag{19}$$

(c) The transition dynamics $P$ is determined by $p_\theta$. Formally,

$$P(s_{t_{i+1}} \mid s_{t_i}, a_{t_i}) \triangleq (\delta_{z_{t_{i+1}}}, \delta_{t_{i+1}}), \tag{20}$$

in which $\delta_y$ is the Dirac delta distribution with non-zero density only at $y$. The specific form when using DDIM (Song et al., 2020) writes Eq. 1 in Sec. 2.1;

(d) The reward function is defined as

$$R(s_{t_i}, a_{t_i}, s_{t_{i+1}}) \triangleq \begin{cases} \mathbb{E}_{z_{t_n} \sim p_\theta(z_{t_{i+1}})}[\phi(z_{t_n})] & \text{if } i+1 = n \\ 0 & \text{otherwise} \end{cases}; \tag{21}$$

(e) The discount factor $\gamma$ is within the interval $(0, 1]$. Since rewards are only obtainable at the final timestep (Eq. 21), $\gamma$ becomes useless in DDPO-style modeling. We still explicitly include it here to maintain consistency with subsequent representations;

(f) The time horizon $K$ is defined as the number of inference steps, *i.e.*,

$$K \triangleq n, \tag{22}$$

which is incorporated into the MDP tuple to emphasize that it's an episodic MDP (Grzes, 2017).

## D.2 SUPPLEMENTARY TO LATENT REWARD SHAPING

### D.2.1 VARIANTS OF $\mathcal{F}$ FOR LATENT REWARD SHAPING

This section presents several variations of the estimation policy $\mathcal{F}$ used for deriving the *pseudo-final latent* $\hat{z}_{t_n} = \mathcal{F}(z_{t_i})$ from an intermediate latent $z_{t_i}$ ($0 \le i < n$).

(a) *immediate-ddim*: The posterior mean in the DDIM (Song et al., 2020) step

$$\mathbb{E}[z_{t_n} \mid z_{t_i}] = \frac{z_{t_i} - \sqrt{1 - \overline{\alpha}_{t_i}} \epsilon_\theta^{(t_i)}(z_{t_i})}{\sqrt{\overline{\alpha}_{t_i}}} \tag{23}$$

can be used to derive pseudo-final latents, *i.e.*,

$$\mathcal{F}(z_{t_i}) = \mathbb{E}[z_{t_n} \mid z_{t_i}]; \tag{24}$$

(b) *immediate-score*: Similar to (a) that directly predicts $\hat{z}_{t_n}$ from $z_{t_i}$, not assuming $z_{t_i}$ is noise-free, Chung et al. (2022) propose a more accurate posterior mean estimation based on the score function (Song & Ermon, 2019), which writes

$$\mathbb{E}[z_{t_n} \mid z_{t_i}] = \frac{z_{t_i} + (1 - \overline{\alpha}_{t_i}) \nabla_{z_{t_i}} \log p_{t_i}(z_{t_i})}{\sqrt{\overline{\alpha}_{t_i}}} \approx \frac{z_{t_i} + (1 - \overline{\alpha}_{t_i}) s_\theta(z_{t_i})}{\sqrt{\overline{\alpha}_{t_i}}}, \tag{25}$$

where $s_\theta(z_{t_i}) = -\frac{\epsilon_\theta(z_{t_i})}{\sqrt{1 - \overline{\alpha}_{t_i}}}$ is the score function at time step $t_i$. The form of $\mathcal{F}$ remains as in Eq. 24;

(c) *look-ahead*: Unlike immediately estimating $\hat{z}_{t_n}$ from $z_{t_i}$, DBLS (Oshima et al., 2025) introduces a look-ahead (LA) estimator. Specifically, it equally interpolates the remaining timesteps from $t_i$ to $t_n = 0$ to 2 or 3 steps, and denoises with 2 or 3 deterministic DDIM steps to obtain the pseudo-final latent. We name they *LA-2* and *LA-3*, respectively.

Formally, denote the DDIM step from timestep $t_i$ to $t_{i+1}$ (Eq. 1 in Sec. 2.1) as $z_{t_{i+1}} = \mathcal{G}(z_{t_i}, \epsilon_{t_i}, \{t_i, t_{i+1}\})$, then the LA-2's derivation of pseudo-final latents can be expressed as

$$\mathcal{F}(z_{t_i}) = \mathcal{G}(z_{t_i}, \epsilon_\theta^{(t_i)}(z_{t_i}), \epsilon_{t_i}, \{t_i, t_j, t_n\}), \tag{26}$$

where $t_j = \left\lfloor \frac{t_i + t_n}{2} \right\rfloor = \left\lfloor \frac{t_i}{2} \right\rfloor$, while the LA-3's writes

$$\mathcal{F}(z_{t_i}) = \mathcal{G}(z_{t_i}, \epsilon_\theta^{(t_i)}(z_{t_i}), \epsilon_{t_i}, \{t_i, t_{j_0}, t_{j_1}, t_n\}), \tag{27}$$

where $t_{j_0} = \left\lfloor \frac{2t_i}{3} \right\rfloor, t_{j_1} = \left\lfloor \frac{t_i}{3} \right\rfloor$;

(d) *sequential*: Starting from $z_{t_i}$, apply deterministic DDIM steps sequentially to obtain $\hat{z}_{t_n}$, *i.e.*,

$$\mathcal{F}(z_{t_i}) = \mathcal{G}(z_{t_i}, \epsilon_\theta^{(t_i)}(z_{t_i}), \epsilon_{t_i}, \{t_i, \cdots, t_n\}). \tag{28}$$

When using *latent rewards* in multi-step inference, the number of intermediate reward queries far exceeds the number of final ones (Zhang et al., 2024a), regardless of whether training-based or training-free. Besides, the intermediate rewards derived from latent rewards suffer from inaccuracy in early denoising (Zhang et al., 2024a).

**Clarification on novelty.** We do **not** propose a new intermediate-reward estimator. Instead, we introduce a unified latent-reward-shaping formulation that encompasses a broad class of existing ODE-based value/intermediate-reward estimation methods. The mapping $\mathcal{F}$ is modular and replaceable, allowing practitioners to choose an appropriate instantiation based on computational or task requirements.

### D.2.2 LATENT REWARD SHAPING IN DIFFERENT STYLE

Differently, SDPO (Zhang et al., 2024a) obtains dense rewards via the cosine similarity between intermediate latents and those at the initial, final, or anchor steps. This policy can not be represented as the $\mathcal{F}$-style latent reward shaping in Sec. D.2.1.

We do **not** adopt this policy, because intermediate rewards should be estimated in a trajectory agnostic manner during the search process in our scenario.

### D.2.3 DISCOUNTED *vs.* UNDISCOUNTED

When using dense rewards, the selection of $\gamma$ assigns different preferences to the optimization of cumulative returns: (i) $\gamma = 1$ treats each denoising step equally (Zhang et al., 2024b); (ii) $\gamma < 1$ emphasizes the contributions from earlier (Zhang et al., 2024a) or later (Ye et al., 2024) denoising steps, depending on the discounting direction.

### D.3 MAX-REWARD CONTROL/RL

Unlike typical control/RL problems that aim to maximize the (discounted) cumulative returns, the *max-reward control/RL problem* focuses on maximizing the (discounted) rewards encountered throughout the process (**not** just the reward at the end of the episode).

**Deterministic Scenarios.** Quah & Quek (2006); Gottipati et al. (2020) introduces a Q-function for *deterministic* scenarios, which recursively optimizes for the expectation of maximum of reward at the current timestep and future expectations, expressed as

$$Q^\pi(s_t, a_t) = \mathbb{E}_{\substack{s_{t+1} \\ a_{t+1}}} \left[ \max \left\{ r_{t+1}, \gamma \mathbb{E}_{\substack{s_{t+2} \\ a_{t+2}}} [\max\{r_{t+2}, r_{t+3}, \cdots\}] \right\} \right], \tag{29}$$

in which $r_{t+1} \triangleq R(s_t, a_t, s_{t+1})$. This Q-function is proven to satisfy the Bellman-like equation

$$Q^\pi(s_t, a_t) = \mathbb{E}_{\substack{s_{t+1} \\ a_{t+1}}} [\max\{r_{t+1}, \gamma Q^\pi(s_{t+1}, a_{t+1})\}]. \tag{30}$$

Although we do **not** employ stochastic control/RL in this study, we still present a brief introduction to highlight the simplicity of deterministic counterpart.

**Stochastic Scenarios.** Veviurko et al. (2024) further extends this concept to *stochastic* environments. Firstly, the *max-reward return*

$$G_t = \max\{r_{t+1}, \gamma r_{t+2}, \gamma^2 r_{t+3}\} \tag{31}$$

in Quah & Quek (2006); Gottipati et al. (2020) is expanded to

$$\mathbb{E}_\pi[G_t] = \mathbb{E}_\pi[r_{t+1} \vee \gamma G_{t+1}], \tag{32}$$

where and $a \vee b \triangleq \max\{a, b\}$ does not commute with $\mathbb{E}$.

Then, with the help of an auxiliary real variable $y \in \mathbb{R}$ that propagates information about the past rewards, the *max-reward value functions* are defined as

$$V^\pi(s, y) = \mathbb{E}_\pi[y \vee G_t \mid s_t = s], \tag{33}$$

$$Q^\pi(s, a, y) = \mathbb{E}_\pi[y \vee G_t \mid s_t = s, a_t = a], \tag{34}$$

which are subject to the Bellman-like equations

$$V^\pi(s, y) = \gamma \mathbb{E}_{\substack{s_{t+1} \\ a_t}} [y' \vee V^\pi(s_{t+1}, y') \mid s_t = s], \tag{35}$$

$$Q^\pi(s, a, y) = \gamma \mathbb{E}_{\substack{s_{t+1} \\ a_{t+1}}} [y' \vee Q^\pi(s_{t+1}, a_{t+1}, y') \mid s_t = s, a_t = a], \tag{36}$$

where $y' \triangleq \dfrac{R(s, a, s_{t+1}) \vee y}{\gamma}$.

### D.4 SDE SAMPLING FOR DMS

### D.4.1 SDE SAMPLING FOR DDIM

The denoising process in DMs is governed by a reverse-time SDE (Song et al., 2021)

$$\mathrm{d}z = \left[ f(z, t) - g(t)^2 \nabla_z \log p_t(z) \right] \mathrm{d}t + g(t)\mathrm{d}\overline{w}, \tag{37}$$

where $\overline{w}$ is an inverse Wiener process.

For standard stochastic DDIM (Song et al., 2020) (Eq. 23),

$$f(z, t) = \frac{\mathrm{d}}{\mathrm{d}t} \left( \log \sqrt{\alpha_t} \right) z, \quad g(t) = \sigma_t, \tag{38}$$

where

$$\sigma_{t_i} = \sigma_{t_i}(\eta_{t_i}) \triangleq \eta_{t_i} \omega_{t_i} \triangleq \eta_{t_i} \sqrt{\frac{1 - \overline{\alpha}_{t_{i-1}}}{1 - \overline{\alpha}_{t_i}}} \sqrt{1 - \frac{\overline{\alpha}_{t_i}}{\overline{\alpha}_{t_{i-1}}}} \tag{39}$$

is the variance of the process noise in Eq. 37, and $\eta_{t_i} \in [0, 1]$ is sampled from a uniform distribution on $[0, 1]$ or a unimodal Beta distribution to introduce randomness, independent at each timestep.

In standard SDE sampling for DDIM, the process noise $\epsilon_{t_i}$ is re-sampled at each timestep, corresponding to the $\mathrm{d}\overline{w}$ term in Eq. 37, which provides isotropic and uncorrelated perturbation.

### D.4.2 VARIANCE ANALYSIS

Consider a discrete analog of the continuous SDE sampling. Using the Euler-Maruyama method (Oksendal, 2013), the discrete-time approximation of Eq. 37 is

$$z_{t_{i+1}} = z_{t_i} + [f(z_{t_i}, t_i) - g(t_i)^2 \nabla_z p_{t_i}(z_{t_i})]\Delta t + g(t_i)\sqrt{\Delta t}\xi_{t_i}, \tag{40}$$

where $\xi_{t_i} \sim \mathcal{N}(0, I)$, and $g(t_i)\sqrt{\Delta t}\xi_{t_i}$ approximates the $\mathrm{d}\overline{w}_t$ over the interval $[t_i, t_{i+1})$.

The conditional expectation

$$\mathbb{E}[z_{t_{i+1}} \mid z_{t_i}] = z_{t_i} + [f(z_{t_i}, t_i) - g(t_i)^2 \nabla_z p_{t_i}(z_{t_i})]\Delta t. \tag{41}$$

The conditional variance

$$\begin{aligned} \mathrm{Var}_{\mathrm{SDE}} &= \mathrm{Var}(z_{t_{i+1}} \mid z_{t_i}) \\ &= \mathbb{E}\left[ \left( g(t_i)\sqrt{\Delta t}\xi_{t_i} \right) \left( g(t_i)\sqrt{\Delta t}\xi_{t_i} \right)^\top \Big| z_{t_i} \right] \\ &= g(t_i)^2 \Delta t I = \omega_{t_i}^2 \eta^2 \Delta t I, \end{aligned} \tag{42}$$

where $\eta_{t_i} = \eta$ $(0 \leq i \leq n - 1)$, and $\mathbb{E}[\xi_{t_i}\xi_{t_i}^\top] = I$.

This variance is a full-rank matrix, which reflects the isotropic perturbation in different directions, and ensures diverse sample generation. Besides, it scales linearly with $\Delta t$ in discrete terms.

### D.4.3 ERROR ANALYSIS

Xu et al. (2023b) points out that, the *discretization error* and the *approximation error* are the the primary errors involved in the SDE sampling for DDIM. It demonstrates that, the diffusive nature of the stochasticity in standard SDE sampling contracts accumulated errors due to its isotropic noise, *i.e.*, it helps to "*reset*" or "*forget*" their cumulative impact. This contraction effect allows SDE samplers to achieve better sample quality in the large NFE regime by mitigating error accumulation across steps.

### D.5 UCB IN MCTS

**UCB Definition.** MCTS (Coulom, 2006; Browne et al., 2012; Rivière et al., 2024) is a heuristic search algorithm for optimizing decision-making processes, where the upper confidence bound (UCB) during the selection phase for a MDP can be written as

$$\mathrm{UCB}(s, a) = \mathrm{exploitation}(s, a) + \lambda_{\mathrm{exploration}} \cdot \mathrm{exploration}(s, a), \tag{43}$$

where $s$ is the state, $a$ is an action used for expansion, and $\lambda_{\mathrm{exploration}} > 0$ is a hyper-parameter used to balance exploration and exploitation.

In conventional implementations, the exploitation and exploration term are respectively given as

$$\mathrm{exploit}(s, a) = Q(s, a), \quad \mathrm{explore}(s, a) = \sqrt{\frac{\ln N(s)}{N(s, a)}}, \tag{44}$$

where $N(s)$ is the number of times state $s$ is visited, and $N(s, a)$ is the number of times action $a$ is chosen for expansion.

**Average-Value Policies.** In cumulative-reward RL, the $V(s)$ is typically estimated with the average return from all simulations passing through $s$. Formally,

$$V(s) = \left( \sum_{i=0}^{N(s)-1} R_i \right) / N(s). \tag{45}$$

Let $R$ be the cumulative or maximum return from the simulation phase, then $V(s)$ is updated as

$$V(s) \leftarrow V(s) + \frac{R - V(s)}{N(s)}, \tag{46}$$

during the backpropagation phase. We refer to this value policy as *average-value policy*.

**Dealing with Continuous Action Spaces.** Standard MCTS only supports discrete action spaces, which are unavailable in our scenario, since two points in continuous spaces are different with probability 1 (Shynk, 2012). It causes $N(s, a) = 1$ almost surely and the exploration term becomes $\sqrt{\ln N(s)}$ consistently, which renders the UCB formula ineffective for balancing exploitation and exploration among actions, as it relies heavily on visit counts to under-explored options.

Recent works have employed progressive widening (Coulom, 2007; Couëtoux et al., 2011) to tackle continuous action spaces (Moerland et al., 2018; Bianchi et al., 2023; Lee et al., 2020) (*e.g.*, DTS (Jain et al., 2025)). However, they only sample candidate actions for selection, which limits the exploration across the entire action space, and are incompatible with our stochastic Beta policy. Instead, we propose a value-based UCB to overcome this challenge (Sec. 4.1), together with Beta policies for exploration (Sec. 4.2).

# E    MORE DISCUSSION ON OUR FORMULATION

## E.1    FLAWS OF THE CONVENTIONAL FORMULATION

We argue that, the *max-final reward formulation* is sub-optimal when adopting dense rewards, regardless of whether the final rewards are optimized directly or indirectly:

- *Indirectly*: Use latent rewards as dense rewards to train or fine-tune DMs by optimizing the cumulative rewards. It is based on the insight that, greater cumulative rewards often lead to greater final rewards. However, latent rewards usually have similar scales to the final rewards, potentially overwhelming the final reward, especially when $n$ is large;

- *Directly*: Using latent rewards as a heuristic to guide the optimization process of the final rewards like FK Steering (Singhal et al., 2025) could potentially mitigate the above misleading. However, the *admissibility* of latent rewards as a valid heuristic is not guaranteed, since pseudo-final samples may achieve higher rewards than the terminal samples, *i.e.*, over-estimation exists. This can mislead the optimization process into over-exploiting paths that appear promising but are actually suboptimal for final reward maximization.

## E.2    COMPARISON WITH EXISTING INFERENCE-TIME SCALING PARADIGMS

Existing inference-time scaling paradigms for DMs that adopt high-dimensional action space (Appx. C.2) have several limitations:

1. Adopting the initial noise $z_{t_0}$ or process noise $\epsilon_{t_i}$ as actions makes a high-dimensional action space $\mathcal{A} \subseteq \mathbb{R}^{c \times h \times w}$, which is hard to enumerate efficiently. Here, $c, h, w$ are the number of channels, height and width of the latent variables, respectively;

2. High-dimensional spaces are sparse, making effective coverage difficult with limited samples. Quantitatively, sampling $M$ points from $\mathbb{R}^d$ faces the sphere-covering problem (Vershynin, 2018), that is, covering a ball of radius $r$ requires $M \propto \left(\frac{r}{\delta}\right)^d$ samples, where $\delta$ is the covering precision. Even for small $r$ and $\delta$, the required $M$ is impractically large due to the high dimension $d = c \times h \times w$;

3. The deviation between the $\epsilon_{t_i}$s resampled at each timestep is relatively large, leading to high variance due to variant perturbation direction at the same timesteps in different runs, which results in slow convergence;

4. The manifold hypothesis (Fefferman et al., 2016) posits that, high-dimensional data lies near a low-dimensional manifold. Thus many parameters of the high-dimensional actions may be redundant.

In contrast, our RODE-based inference-time scaling paradigm (Sec. 3.4 in the main paper) offers the following advantages:

1. Our 1-D actions are easy to sample with 1-D probability density functions;

2. It becomes convenient to visualize the high-dimensional denoising trajectories using polar plots (Sec. 5.2, Appx. I.4) by parameterizing on low-dimensional action sequences;

3. Our low-dimensional compact action space $\mathcal{A} = [0, 1]$ can be efficiently covered with limited samples. Specifically, sampling $M$ points $\{x^{(i)}\}_{i=0}^{M-1}$ in $[0, 1]$ ensures that, for any $x^* \in [0, 1]$, there exists $x^{(j)}$ s.t. $|x^* - x^{(j)}| \leq \frac{1}{2M}$ (Kuipers & Niederreiter, 2012);

4. The standard deviation $\sigma_{t_i} = \eta_{t_i} \cdot \omega_{t_i}$ in DDIM (Eq. 3) is a linear function of $\eta_{t_i}$, where

$$\omega_{t_i} = \sqrt{\frac{1 - \overline{\alpha}_{t_{i-1}}}{1 - \overline{\alpha}_{t_i}}} \sqrt{1 - \frac{\overline{\alpha}_{t_i}}{\overline{\alpha}_{t_{i-1}}}}$$ is a time-variant constant, making changes in $\eta_{t_i}$ smoothly

adjust $\sigma_{t_i}$. Fixing the process noise $\{\epsilon_{t_i}\}_{i=0}^{n-1}$, the perturbation direction at each timestep is fixed, with its strength controlled by $\eta_{t_i}$. In other words, our modeling decouples noise direction ($\epsilon_{t_i}$) and magnitude ($\eta_{t_i}$), enabling controllable trajectory modulation and smooth navigation of the continuous action space;

5. Our low-dimensional actions make it easy to optimize the likelihood of sampling a certain action and the actions nearby (Sec. 4.2).

### E.3 COMPARISON WITH EXISTING METHODS

**Comparison with an Existing Method that Frames the Data Generation as a Control Problem.** Previously, Fan & Lee (2023) frames the data generation as a control problem, and fine-tunes DDPM (Ho et al., 2020) samplers to find a shorter path to the data distribution, instead of strictly follow the DM's backward process. Our method differs in these aspects:

1. We only augment each timestep of the original backward process with a shortcut to the terminal step, rather than identifying a completely novel denoising path;

2. We align pretrained DMs in a training-free manner by optimizing the action sequence $\{\eta_{t_i}\}_{i=0}^{n-1}$ online, rather than tuning the DM parameters.

Besides, treating pseudo-final samples as valid samples in our max-reward formulation, can be viewed as a novel implementation for deviating from the DM's pre-defined trajectory for better performance.

**Comparison with an Existing Method that Leverage MCTS for Trajectory Search.** Concurrently, Ramesh & Mardani (2025) models the denoising process as a MDP with a terminal reward, and perform trajectory search using MCTS with contextual bandits. Our method differs in these aspects:

1. We scale low-dimensional $\eta_{t_i}$ actions (Sec. 3.2) rather than the high-dimensional $\epsilon_{t_i}$ actions employed by Ramesh & Mardani (2025);

2. We adopt dense rewards throughout the entire denoising process, rather than the solely final rewards employed by Ramesh & Mardani (2025);

3. We adopt dense rewards for the MDP, and model DM alignment as an max-encountered-reward optimal control problem (Sec. 3.3, Sec. 3.4) rather than the max-final reward problem in Ramesh & Mardani (2025);

4. Ramesh & Mardani (2025) discretizes the continuous action space by pre-fixing a candidate set of process noise for every timestep, whereas ours directly explores the continous action space via Beta polices.

**Comparison with Diffusion Tree Sampling.**  Also concurrently, Diffusion Tree Sampling (DTS) (Jain et al., 2025) introduce a MCTS-based approach that samples from the reward-aligned target density by propogating terminal rewards back through the diffusion chain, and progressively refining value estimation. Our method differs in these aspects:

1. We select candidate tree nodes for expansion based on our proposed value-based UCB (Sec. 4.1), rather than the Boltzmann distribution employed by DTS;

2. We adopt a low-dimensional action space (Sec. 3.2), and expand nodes with actions sampled from onlne-updated Beta distributions (Sec. 4.2), instead of directly sampling from high-dimensional distributions as in DTS. Besides, we do **not** employ progressive widening (Coulom, 2007; Couëtoux et al., 2011) to deal with continuous action spaces as done in DTS;

3. Beyond terminal rewards, we compute intermediate rewards at every tree node (Sec. 3.3), and introduce max-reward modeling to fully exploit these intermediate supervision signals (Sec. 3.1);

4. We update nodes's values with our proposed max-value policy (Sec. 4.1), which differs from the soft value estimation adopted in DTS.

**Comparison with Demons.**  Demons (Yeh et al., 2024) also seems to involve scaling scalars. Our method differs in these aspects:

1. **Difference in the role of scalar variables.**  (I) After randomly sampling a set of candidate noise vectors $\{z^{(k)}\}_{k=1}^{K}$, Demons forms a new noise vector $z^*$ by weighting them with a set of scalars $\{b_k\}_{k=1}^{K}$ (Sec. 4.2 in [2]). Importantly, each $b_k$ *is computed adaptively* according to the relative advantage of $z^{(k)}$ among $\{z^{(k)}\}_{k=1}^{K}$ rather than from an optimization process; (II) Differently, we pre-sample and fix a set of process-noise directions, and the MCTS optimization explicitly searches for the scalar sequence $\{\eta_{t_i}\}_{i=0}^{n-1}$. These scalars in our formulation are **not** adaptively computed, instead, they are variables that *being optimized*;

2. **Clarification regarding RODE sampling.**  (I) The candidate noise set of Demons $\{z^{(k)}\}_{k=1}^{K}$ is resampled at every expansion step. From a continuous-time perspective, this repeated re-sampling induces a stochastic term — a Wiener increment $\mathrm{d}w$ — when writing its differential-equation formulation. Consequently, the continuous-time interpretation of Demons corresponds to an SDE sampling process; (II) Differently, we pre-sample and fix a set of process-noise realizations and reuse them during search instead of resampling (Appx. F.3.4). This replaces the stochastic term $g(t)\mathrm{d}w$ with an anisotropic drift term $\omega(t)\eta(t)h(t)\mathrm{d}t$ (Eq. 8). With process noise fixed, both $\omega(t)$ and $h(t)$ are deterministic, and the only stochastic component is the scalar-valued process $\eta(t)$. Consequently, the resulting dynamics contain no Wiener-process term, making the formulation a random ODE (RODE) rather than an SDE. To the best of our knowledge, prior works have not formalized sampling via pre-fixed process-noise realizations as a RODE for inference-time alignment.

Besides, our modeling decouples noise direction ($\epsilon_{t_i}$) and magnitude ($\eta_{t_i}$), enabling controllable trajectory modulation and smooth navigation of the continuous action space, while Demons (Yeh et al., 2024) does **not**.

**Comparison with methods that handle non-differentiable rewards.**  DNO (Tang et al., 2024) does explicitly involve gradients of the reward functions, but it adopts gradient estimation to relax the constraint of differentiability. In contrast, our method does **not** necessitate differentiable reward functions because it is entirely simulation-based and *does not incorporate any reward gradients*.

An intuitive way to understand this distinction: Gradient-based navigation tends to be more beneficial or even necessary to efficiently explore high-dimensional action spaces. However, our action

space is extremely low-dimensional, allowing MCTS to cover it effectively with a limited search budget (Appx. E.2). As a result, gradient computation or estimation is unnecessary for improving navigation efficiency. Moreover, involving gradient estimation in such a low-dimensional space may waste NFE dynamics and NFE reward evaluations, potentially leading to suboptimal performance.

### E.4 RELATIONSHIP TO EDM FRAMEWORK

Solving for the optimal control sequence $\{\eta_{t_i}\}_{i=0}^{n-1}$ is similar to indirectly performing an adaptive search for a time-variant $S_{\text{churn}}$ sequence in the stochastic sampling of EDM (Karras et al., 2022) framework.

## F MORE DISCUSSION ON AUGMENTED RODE SAMPLING

### F.1 RODE SAMPLING FOR DDIM

Recall that

$$
\begin{aligned}
\text{SDE: } \mathrm{d}z &= [f(z,t) - g(t)^2 \nabla_z \log p_t(z)]\mathrm{d}t + g(t)\mathrm{d}\overline{w} \\
\text{RODE: } \mathrm{d}z &= \left[f(z,t) - \omega(t)^2 \eta(t)^2 \nabla_z \log p_t(z)\right] \mathrm{d}t + \omega(t)\eta(t)h(t)\mathrm{d}t
\end{aligned}
\tag{47}
$$

The insight to sample $\eta_{t_i}$ from some certain probability density functions (PDFs) is inspired by sampling the scatter directions from PDFs at each hit point in ray tracing (Glassner, 1989).

**Assumption 1.** *Consider a fixed timestep schedule $T = t_0 > t_1 > \cdots > t_n = 0$. The SDE-based and RODE-based sampler share the same discretization grid and numerical integrator, thus the discretization error is the same for both schemes. We ignore the common discretization error in the subsequent analysis.*

**Assumption 2.** *The timestep interval $\Delta t$ in the discrete timestep schedule is sufficiently small.*

**Assumption 3.** *For all $k$, $\eta_{t_k}$ is independent of $\epsilon_{t_k}$, and $\eta_{t_k}s$ are independent across steps. All constants appearing below are finite, and do **not** blow up as $\Delta t \to 0$.*

The exact solution of the RODE in Eq. 47 writes

$$
z_{t_{i+1}} = z_{t_i} + \int_{t_i}^{t_{i+1}} [f(z_t, t) - \omega(t)^2 \eta(t)^2 \nabla_z \log p_t(z_t) + \omega(t)\eta(t)e(t)]\mathrm{d}t ,
\tag{48}
$$

which is approximated by

$$
z_{t_{i+1}} \approx z_{t_i} + [f(z_{t_i}, t_i) - \omega_{t_i}^2 \eta_{t_i}^2 s_\theta(z_{t_i}, t_i) + \omega_{t_i}\eta_{t_i}\epsilon_{t_i}](t_{i+1} - t_i),
\tag{49}
$$

where $s_\theta(z, t)$ is the score model (Song & Ermon, 2019).

The RODE sampling adopts anisotropic and non-diffusive noise, which lacks the properties of a Wiener process (*e.g.*, uncorrelated increments and isotropic diffusion), suggesting poorer coverage of the $\epsilon_{t_i}$ space. The noise term is confined to the direction of $\epsilon_{t_i}$ and randomly scaled by $\eta_{t_i}$ for each time step $t_i$, resembling a randomized perturbation along fixed paths rather than a true stochastic diffusion.

After switching to $\eta(t)$ as the stochastic process in sampling, the drift term $[f(z,t) - g(t)^2 \nabla_z \log p_t(z)]\mathrm{d}t$ in the original SDE sampling also becomes random. If we assume that, reusing the SDE-trained score network $s_\theta(z,t) \approx \nabla_z \log p_t(z_t)$ within RODE sampling does **not** increase score-estimation error (this assumption will be relaxed in Prop. 6), then

- On the one hand, we argue that stochastic DDIM (Song et al., 2020) can naturally accommodate this randomness without special handling, since the learned score function (Song et al., 2021) is robust to the noise schedule $\sigma_{t_i}$ (Song et al., 2021). That is, any error introduced at the current step is automatically corrected by the model in the subsequent step;

- On the other hand, this randomness mitigates the problem of reduced diversity caused by restricted process noise, which is empirically validated in Sec. 5.2 of the main paper.

We will ignore this term in the following discussion, and focus on the impact of the noise term whose stochastic is fundamentally changed. Under the above assumptions, the DDIM step can be written as

$$z_{t_{i+1}} \approx \Psi_{t_i}(z_{t_i}) + \omega_{t_i}\eta_{t_i}\epsilon_{t_i} , \tag{50}$$

where $\Psi_{t_i}(\cdot)$ is a deterministic map.

**Assumption 4.** *Each deterministic map $\Psi_{t_i}(\cdot)$ is $L_{t_i}$-Lipschitz in z. Then the forward-propagation is upper bound by*

$$M_{t_{i+1}} \triangleq \prod_{j=i+1}^{n-1} L_{t_j} \ (0 \le j \le n-1), M_{t_n} \triangleq 1. \tag{51}$$

## F.2 VARIANCE ANALYSIS

Note that $\eta(t) \in (0,1]$ for stochastic DDIM sampling (Song et al., 2020). If $\mathrm{Var}(\eta(t))$ exists, it can be further guaranteed that, $v_t \triangleq \mathrm{Var}(\eta(t)) < \frac{1}{4}$ according to *Popoviciu's inequality* on variances (Bencze et al., 2010), thus finite.

The contribution of the noise term in Eq. 48 is

$$\int_{t_0}^{t_n} \omega(t)\eta(t)e(t)\mathrm{d}t. \tag{52}$$

At time step $t_i$ in the discrete analog, the variance it brings writes

$$\mathrm{Var}_{\mathrm{RODE}} = \mathrm{Var}\left(\omega_{t_i}\eta_{t_i}\epsilon_{t_i}\Delta t\right) = \omega_{t_i}^2(\Delta t)^2\mathrm{Var}(\eta_{t_i})\epsilon_{t_i}\epsilon_{t_i}^\top. \tag{53}$$

$\epsilon_{t_i}\epsilon_{t_i}^\top$ is rank-1 since it has only one linearly independent column, unlike the full-rank variance in $\mathrm{Var}_{\mathrm{SDE}}$ (Eq. 42). Intuitively, this variance is anisotropic and rank-deficient, with smaller magnitude due to $\mathrm{Var}(\eta_{t_i}) < 1$ and $(\Delta t)^2$, which results in lower variance in directions orthogonal to $\epsilon_{t_i}$, providing lower variance and reducing sample diversity. Prop. 4 describes this property:

**Proposition 4.** (Prop. 1 in the main paper) *The variance of our RODE sampling is strictly less than that of standard SDE sampling in Loewner order* (Horn & Johnson, 2012).

***Proof.*** Consider the *trace* of the variances of both sampling paradigm, which reflect the overall dispersion from the mean.

The trace of the standard SDE sampling's variance (Eq. 42) at times step $t_i$ writes

$$\mathrm{Tr}(\mathrm{Var}_{\mathrm{SDE}}) = d\omega_{t_i}^2\eta^2\Delta t, \tag{54}$$

where $d = c \times h \times w$ denotes the dimension of $\epsilon_{t_i}$ or $I$.

The trace of our RODE sampling's variance (Eq. 53) writes

$$\mathrm{Tr}(\mathrm{Var}_{\mathrm{RODE}}) = \omega_{t_i}^2\mathrm{Var}(\eta_{t_i})(\Delta t)^2||\epsilon_{t_i}||^2, \tag{55}$$

since its eigenvalues are $||\epsilon_{t_i}||^2$ (with multiplicity 1) and 0 (with multiplicity $(n-1)$).

Eq. 54 scales with $\Delta t$, while Eq. 55 scales with $(\Delta t)^2$. As $\Delta t \to 0$, $\mathrm{Tr}(\mathrm{Var}_{\mathrm{RODE}}) \le \mathrm{Tr}(\mathrm{Var}_{\mathrm{SDE}})$ since all the other items are finite.

More precisely, consider the *Loewner order* (Horn & Johnson, 2012) of these two variances. Specifically, for

$$\begin{aligned} \Delta\mathrm{Var} &= \mathrm{Var}_{\mathrm{SDE}} - \mathrm{Var}_{\mathrm{RODE}} \\ &= \omega_{t_i}^2\eta^2\Delta t I - \omega_{t_i}^2\mathrm{Var}(\eta_{t_i})(\Delta t)^2\epsilon_{t_i}\epsilon_{t_i}^\top, \end{aligned} \tag{56}$$

In directions orthogonal to $\epsilon_{t_i}$, the $\epsilon_{t_i}\epsilon_{t_i}^\top$ term contributes 0, so the eigenvalues of $\Delta\mathrm{Var}$ in these directions are $\omega_{t_i}^2\eta^2\Delta t > 0$; In the direction of $\epsilon_{t_i}$, the eigenvalue

$$\omega_{t_i}^2\eta^2\Delta t - \omega_{t_i}^2\mathrm{Var}(\eta_{t_i})(\Delta t)^2||\epsilon_{t_i}||^2 > 0 \tag{57}$$

since $\eta^2 > \mathrm{Var}(\eta_{t_i})\|\epsilon_{t_i}\|^2 \Delta t$ as $\Delta t \to 0$ (this is because $\eta > 0$ for SDE sampling).

Therefore, $\Delta \mathrm{Var}$ is positive definite, making $\mathrm{Var}_{\mathrm{RODE}} < \mathrm{Var}_{\mathrm{SDE}}$ in Loewner order. □

The lower variance makes the distribution derived from our RODE sampling more concentrated than that from the standard SDE sampling. More precisely, under some sub-Gaussian assumptions (detailed in Appx. F.3), for constants $C, c > 0$, the vector concentration yields

$$\Pr\left(\|\Delta_{t_n}^{(\bullet)}\| \geq C\sqrt{V_\bullet} + u\right) \leq 2\exp\left(-c\frac{u^2}{V_\bullet}\right), \tag{58}$$

where $\bullet = \{\mathrm{SDE}, \mathrm{RODE}\}$, $\Delta_{t_n}^{(\bullet)} = z_{t_n}^{(\bullet)} - \mathbb{E}[z_{t_n}^{(\bullet)}]$ is the deviation of samples around their mean. Thus, both schemes display exponential tail decay and strong concentration around their means.

### F.3 ERROR ANALYSIS

#### F.3.1 DISTRIBUTION DISTANCE

As a novel sampling paradigm, we need to explore whether our RODE sampling can approximate the true data distribution $p_{\mathrm{data}}$ as the standard SDE sampling (Song et al., 2021) does. To do this, we consider the Wasserstein-1 distance (Kolouri et al., 2017) $W_1(\cdot, \cdot)$ between the distribution $p^{(\mathrm{S})} \triangleq p_{t_n}^{(\mathrm{S})}$ and $p^{(\mathrm{R})} \triangleq p_{t_n}^{(\mathrm{R})}$ induced by SDE and RODE sampling, respectively, where $p_t^{(\bullet)}$ is the marginal distribution at timestep $t$.

**Assumption 5.** $p^{(S)}$ and $p^{(R)}$ *have finite first moments.*

**Assumption 6.** *The score model $s_\theta(z, t)$ is well-trained, so that the score-estimation error does **not** dominate the stochastic error budget. Precisely, the contribution from the model error at timestep $t_i$ is upper-bounded by a sub-Gaussian perturbation of scale $\gamma_{t_i}$. Perturbations at different timesteps are assumed independent.*

We perform local linearization for SDE and RODE steps to propagate point-wise error:

$$z_{t_{i+1}}^{\mathrm{S}} = \Psi_{t_i}(z_{t_i}^{\mathrm{S}}) + \omega_{t_i}\eta\epsilon_{t_i} + \delta_{t_i}^{\mathrm{score,\,S}}, \tag{59}$$

$$z_{t_{i+1}}^{\mathrm{R}} = \Psi_{t_i}(z_{t_i}^{\mathrm{R}}) + \omega_{t_i}\eta_{t_i}\epsilon_{t_i} + \delta_{t_i}^{\mathrm{score,\,R}}, \tag{60}$$

where $\delta_{t_i}^{\mathrm{score},\bullet}$ is the equivalent perturbation induced by the score-estimation error.

**Proposition 5.** (Prop. 2 in the main paper) *The Wasserstein-1 distance of distributions induced by the two paradigms is bounded:*

$$W_1(p^{(S)}, p^{(R)}) \leq \sum_{k=0}^{n-1} M_{t_{k+1}}\left(\omega_{t_k} \cdot C_{\epsilon,t_k} \cdot \mathbb{E}[\Delta\eta_{t_k}]\right) + \mathcal{O}\left(\sum_{k=0}^{n-1} M_{t_{k+1}}\gamma_{t_k}\right), \tag{61}$$

*where $C_{\epsilon,t_k} \triangleq \mathbb{E}[\|\epsilon_{t_k}\|]$, and $\Delta\eta_{t_k} \triangleq |\eta_{t_k} - \eta| \leq 1$.*

***Proof.*** Consider the difference of samples.

Define the point-wise trajectory error as $\Delta_{t_i} \triangleq z_{t_i}^{\mathrm{R}} - z_{t_i}^{\mathrm{S}}$. A first-order Taylor expansion yields

$$\begin{aligned}\Delta_{t_{i+1}} &= \Psi_{t_i}(z_{t_i}^{\mathrm{R}}) - \Psi_{t_i}(z_{t_i}^{\mathrm{S}}) + (\omega_{t_i}\eta_{t_i} - \omega_{t_i}\eta)\epsilon_{t_i} + (\delta_{t_i}^{\mathrm{score,\,R}} - \delta_{t_i}^{\mathrm{score,\,S}}) \\ &= J_{t_i}\Delta_{t_i} + r_{t_i}(\Delta_{t_i}) + \omega_{t_i}(\eta_{t_i} - \eta)\epsilon_{t_i} + \Delta_{t_i}^{\mathrm{score}}\end{aligned}, \tag{62}$$

where $J_{t_i}$ is the Jocabian of $\Psi_{t_i}(\cdot)$ evaluated at the reference point in the local linearization, $r_{t_i}(\Delta_{t_i})$ is the higher-order residual term, and $\Delta_{t_i}^{\mathrm{score}} \triangleq \delta^{\mathrm{score,R}} - \delta^{\mathrm{score,S}}$, $\|\Delta_{t_i}^{\mathrm{score}}\| \leq 2\gamma_{t_i}$.

**Assumption 7.** *The Jocabian $J_{t_i}$ is bounded, and $\|J_{t_i}\| \leq L_{t_i}$.*

**Assumption 8.** *The mapping $\Psi_{t_i}(\cdot)$ is twice continuously differentiable with bounded second derivatives. Thus, there exists a constant $C_{t_i} > 0$ such that $\|r_{t_i}(\Delta_{t_i})\| \leq C_{t_i}\|\Delta_{t_i}\|^2$.*

Neglecting the quadratic residual, the main components of the point-wise error writes

$$||\Delta_{t_{i+1}}|| \leq L_{t_i}||\Delta_{t_i}|| + \omega_{t_i} \cdot |\eta_{t_i} - \eta| \cdot ||\epsilon_{t_i}|| + ||\Delta_{t_i}^{\text{score}}|| \,. \tag{63}$$

To make the bound more interpretable, we separate the contribution of the $\eta$-bias term $\mathbb{E}[\Delta\eta_{t_k}]$ from the remaining score-estimation error $\gamma_{t_k}$. Starting from $\Delta_{t_0} = 0$ (*i.e.*, shared initial noise in SDE and RODE sampling), the difference of samples writes

$$\begin{aligned}
\mathbb{E}||\Delta_{t_n}|| &\leq \sum_{k=0}^{n-1} M_{t_{k+1}}(\omega_{t_k} \cdot C_{\epsilon,t_k} \cdot \mathbb{E}[\Delta\eta_{t_k}] + 2\gamma_{t_k}) + (\text{higher-order residual}) \\
&\leq \sum_{k=0}^{n-1} M_{t_{k+1}}(\omega_{t_k} \cdot C_{\epsilon,t_k} \cdot \mathbb{E}[\Delta\eta_{t_k}]) + \mathcal{O}\left(\sum_{k=0}^{n-1} M_{t_{k+1}}\gamma_{t_k}\right)
\end{aligned} \,, \tag{64}$$

The first term corresponds to the main contribution arising from the bias of the scalar process $\eta(t)$, while the second term collects all residual errors, including the score-estimation error and higher-order discretization terms. Hence, the discrepancy between samples generated from the two paradigms can be controlled, and the $\eta$-bias is the dominant source of discrepancy between RODE and SDE trajectories (Assump. 6).

Let $\Pi(p, q)$ be the set of coupling distributions with marginals $p$ and $q$. Couple $p^{(\text{S})}$ and $p^{(\text{R})}$ on the same probability space (*i.e.*, using the same initial noise and process noise realization $\{\epsilon_{t_i}\}_{i=0}^{n-1}$). Note that

$$\begin{aligned}
W_1(p^{(\text{S})}, p^{(\text{R})}) &= \inf_{\pi \in \Pi} \mathbb{E}_{\pi, X \sim p^{(\text{S})}, Y \sim p^{(\text{R})}} ||X - Y|| \\
&\leq \mathbb{E}_{\pi_0, z_{t_n}^{(\text{S})} \sim p^{(\text{S})}, z_{t_n}^{(\text{R})} \sim p^{(\text{R})}} ||z_{t_n}^{(\text{S})} - z_{t_n}^{(\text{R})}|| \\
&= \mathbb{E}||\Delta_{t_n}|| \,,
\end{aligned} \tag{65}$$

where $\pi_0$ is a specific coupling distribution. Thus the distance of distributions induced by the two paradigms is also controllable. $\qquad\square$

### F.3.2    SCORE-ESTIMATION ERROR

Furthermore, we relax the assumption made in Appx. F.1 by explicitly analysing the score-estimation error that arises when RODE sampling reuses the score network $s_\theta(\cdot)$ originally trained for SDE sampling. Intuitively,

1. $s_\theta(\cdot)$ is trained with a time-invariant $\eta$, whereas RODE sampling employs a time-varying $\eta_{t_i}$. This mismatch may push the latents into regions where the training data is sparse and $s_\theta(\cdot)$ is therefore under-regularized, leading to larger score-estimation error;

2. The standard SDE sampling adopts a Wiener process to explore all directions in the $\epsilon_{t_i}$ space uniformly, which shows *error contraction effect* (Appx. D.4.3). However, the diffusive nature of $\mathrm{d}\overline{w}$ is absent in RODE sampling, and the directional noise lacks the ability to provide the same error-correcting benefits as the SDE's diffusive noise, potentially allowing errors to accumulate in underrepresented directions. Apart from restricted perturbation direction, the noise term in our RODE sampling scales with $\mathrm{d}t$ rather than $\sqrt{\mathrm{d}t}$ in standard SDE, which may limits its ability to correct accumulated error;

3. The $s_\theta(\cdot)$ approximates marginal distributions and is independent of sampling trajectories, and RODE sampling only alters the trajectories but **not** the input distribution, therefore reusing $s_\theta(\cdot)$ in RODE sampling does not introduce an additional error order.

More precisely, let the point-wise score-estimation error at timestep $t$ be

$$\mathcal{E}(z, t) = ||s_\theta(z, t) - s(z, t)|| \,, \tag{66}$$

where $s(z, t)$ is the groud-truth score function.

**Assumption 9.** *$\mathcal{E}(\cdot, t)$ is $L_{\mathcal{E},t}$-Lipschitz. Otherwise, we can resort to arguments in Birrell (2025), where a local Lipschitz condition is imposed to handle the unbounded cases.*

**Proposition 6.** (Prop. 3 in the main paper) *The difference of score-estimation error writes*

$$\left| \mathbb{E}_{p_t^{(R)}} \mathcal{E}(z,t) - \mathbb{E}_{p_t^{(S)}} \mathcal{E}(z,t) \right| \leq L_{\mathcal{E},t} \cdot W_1(p_t^{(R)}, p_t^{(S)}) \,, \tag{67}$$

*where the constant $L_{\mathcal{E},t} > 0$.*

**Proof.** The Kantorovich–Rubinstein duality (Villani et al., 2008) of Wasserstein-1 distances gives

$$
\begin{aligned}
\left| \mathbb{E}_{p_t^{(R)}} \mathcal{E}(z,t) - \mathbb{E}_{p_t^{(S)}} \mathcal{E}(z,t) \right| &\leq L_{\mathcal{E},t} \cdot \sup_{\text{Lip}(\mathcal{E}_0) \leq 1} \left| \mathbb{E}_{p_t^{(R)}} \mathcal{E}_0(z,t) - \mathbb{E}_{p_t^{(S)}} \mathcal{E}_0(z,t) \right| \\
&= L_{\mathcal{E},t} \cdot W_1(p_t^{(R)}, p_t^{(S)}) \,,
\end{aligned}
\tag{68}
$$

where $\text{Lip}(\mathcal{E}_0) \leq 1$ denotes the set of 1-Lipschitz functions $\mathcal{E}_0$. $\qquad\square$

Consequently, if the deviations $\mathbb{E}[\Delta\eta_{t_k}]$s are large, RODE sampling may increase the expected score estimation error relative to SDE sampling. Conversely, if $\mathbb{E}[\Delta\eta_{t_k}]$s are small, (*e.g.*, when the fixed $\eta$ in SDE sampling equals to the $\mathbb{E}[\Delta\eta_{t_k}]$ in RODE sampling), the above increase is small.

**Summary.** Our RODE sampling provides lower variance due to reduced randomness, but higher error due to biased exploration and time-varying $\eta_{t_i}$. Empirically, RODE sampling can still generate reasonable samples under larger errors (Sec. 5.2 in the main paper), which we attribute to:

1. The training process of DMs, such as the training of score functions (Song & Ermon, 2019) is intrinsically robust to noise variations (Song & Ermon, 2019; Song et al., 2021; Chen et al., 2025b);

2. DMs are robust to sampling variations, and adjusting noise schedules $\sigma(t)$ (indirectly influenced via $\eta(t)$ in our scenario) can shift sample characteristics (Karras et al., 2022);

3. Smaller noise magnitude limits dispersion, but can not prevent the "drift term" reaching the data manifold, thus ensuring plausibility.

Together, these factors justify the practical validity and stability of using RODE sampling for exploration.

### F.3.3 STRONGER PROPOSITION

**Assumption 10.** *The score-estimation error for SDE sampling decreases as $o_N(1)$ as the sample size $N \to \infty$. Note that the analogous error term may remain at the order of $\mathcal{O}_N(1)$ for RODE sampling.*

The right-hand side in Prop. 5 may not vanish as $\Delta t \to 0$, instead, it typically converges to a finite nonzero constant. Consequently, the right-hand side of Proposition 3 does not vanish either. Therefore, Proposition 3 only establishes a stability-type guarantee that the score-estimation error of RODE sampling does not blow up, rather than demonstrating convergence to the zero-error limit achieved by SDE sampling.

Actually, under stronger assumptions, one can show mathematically that $p^{(R)}$ and $p^{(S)}$ are sufficiently close.

**Assumption 11.** $\mathbb{E}[\Delta\eta_{t_k}] = \mathcal{O}(\Delta t^\nu)$, $\nu > 1$.

**Proposition 7.** *The right-hand side in Prop. 5 vanishes, therefore:*

1. *$p^{(R)}$ and $p^{(S)}$ are sufficiently close;*

2. *Prop. 6 demonstrate that the score-estimation error of RODE sampling converges to the zero-error limit as SDE sampling does.*

**Proof.** The first term of the right-hand side in Prop. 5 vanishes since

$$\sum_{k=0}^{n-1} \mathbb{E}[\Delta\eta_{t_k}] = n \cdot \mathcal{O}(\Delta t^\nu) = \frac{T}{\Delta t}\mathcal{O}(\Delta t^\nu) = T \cdot \mathcal{O}(\Delta t^{\nu-1}) \to 0, \tag{69}$$

while the second term also vanishes as $N \to \infty$. $\qquad\square$

### F.3.4 COMPARISON BETWEEN SDE SAMPLING AND RODE SAMPLING

**Pseudo-Code.** We provide pseudo-code in Alg. 1 and Alg. 2 to clearly demonstrate the behavior of SDE sampling and RODE sampling when generating a batch of samples.

---

**Algorithm 1** SDE Sampling for a Batch

**Input:**
- $B$: Batch size
- $n$: Inference step
- $\epsilon_\theta(\cdot, \cdot)$: Noise prediction network
- $\{t_i\}_{i=0}^{n-1}$: Timestep schedule ($t_0 > t_1 \cdots > t_n$)
- $\{z_{t_0}^{(k)}\}_{k=0}^{B-1}$: A batch of initial noise

1: **for** $i = 0$ **to** $(n-1)$ **do**
2:     **for** $k = 0$ **to** $(B-1)$ **do**
3:         $\epsilon_{\text{pred}}^{(k)} \leftarrow \epsilon_\theta(z_{t_i}^{(k)}, t_i)$
4:         $\epsilon_{\text{process}}^{(t_i,k)} \sim \mathcal{N}(0, I)$                      $\triangleleft$ Sample process noise on the fly
5:         $z_{t_{i+1}}^{(k)} \leftarrow \mathcal{G}(z_{t_i}^{(k)}, \epsilon_{\text{pred}}^{(k)}, \epsilon_{\text{process}}^{(t_i,k)}, \{t_i, t_{i+1}\})$
6:     **end for**
7: **end for**
8: **return** $\{z_{t_n}^{(k)}\}_{k=0}^{B-1}$

---

**Process Noise.** We provide an example to clarify the distinction of process noise adopted within SDE sampling and RODE sampling. Consider two samples $A$ and $B$ within a **single** experiment, which adopt $\{\epsilon_{t_i}^{(A)}\}_{i=0}^{n-1}$ and $\{\epsilon_{t_i}^{(B)}\}_{i=0}^{n-1}$ as process noise sequences, respectively. For SDE sampling, $\{\epsilon_{t_i}^{(A)}\}_{i=0}^{n-1}$ and $\{\epsilon_{t_i}^{(B)}\}_{i=0}^{n-1}$ may differ, while they are identical for RODE sampling.

**Comparison of Distribution.** For a shared realization of process noise sequence $\{\epsilon_{t_i}\}_{i=0}^{n-1}$, we argue that $p^{(S)}$ and $p^{(R)}$ overlap, but are not in an inclusion relationship:

- For time-invariant $\eta(t)$, perturbations in RODE sampling are smaller per step, exploring a less dispersed region than SDE due to $\Delta t < \sqrt{\Delta t}$ for small $\Delta t$;

- For time-variant $\eta(t)$, RODE sampling could venture into regions beyond SDE's reach due to varying perturbation amplitudes.

As discussed in Appx. F.3.2, $p^{(S)}$ and $p^{(R)}$ are close when the time-invariant $\eta(t)$ in SDE sampling is equal to the $\mathbb{E}[\eta(t)]$ in RODE sampling.

### F.4 AUGMENTED RODE SAMPLING

We take $m$ independent sets of process noise $\{\{\epsilon_{t_i}^{(j)}\}_{i=0}^{n-1}\}_{j=0}^{m-1}$ for our *augmented RODE sampling*. Note that this does **not** change the marginal distribution of RODE sampling.

**Proposition 8.** *As $m$ increases, $\{\epsilon_{t_i}^{(j)}\}_{j=0}^{m-1}$ spans $\mathbb{R}^d$ and shows approximate isotropy due to the independence of the process noise.*

*Proof.* The empirical covariance of the noise terms in Eq. 53

$$\frac{1}{m} \sum_{j=0}^{m-1} [\omega_{t_i} \eta_{t_i}^{(j)} \epsilon_{t_i}^{(j)} \Delta t][\omega_{t_i} \eta_{t_i}^{(j)} \epsilon_{t_i}^{(j)} \Delta t]^\top \tag{70}$$

approximates

$$\omega_{t_i}^2 \eta_{t_i}^2 (\Delta t)^2 I_d \tag{71}$$

---

**Algorithm 2** RODE Sampling for a Batch

---

**Input:**

- $B$: Batch size
- $n$: Inference step
- $\epsilon_\theta(\cdot, \cdot)$: Noise prediction network
- $\{t_i\}_{i=0}^{n-1}$: Timestep schedule ($t_0 > t_1 \cdots > t_n$)
- $\{z_{t_0}^{(k)}\}_{k=0}^{B-1}$: A batch of initial noise
- $\{\epsilon_{\text{process}}^{(t_i)}\}_{i=0}^{n-1}$: A sequence of process noise, $\epsilon_{\text{process}}^{(t_i)} \sim \mathcal{N}(0, I)$               ◁ Pre-sample

1: **for** $i = 0$ **to** $(n-1)$ **do**
2:      **for** $k = 0$ **to** $(B-1)$ **do**
3:          $\epsilon_{\text{pred}}^{(k)} \leftarrow \epsilon_\theta(z_{t_i}^{(k)}, t_i)$
4:          $z_{t_{i+1}}^{(k)} \leftarrow \mathcal{G}(z_{t_i}^{(k)}, \epsilon_{\text{pred}}^{(k)}, \epsilon_{\text{process}}^{(t_i)}, \{t_i, t_{i+1}\})$
5:      **end for**
6: **end for**
7: **return** $\{z_{t_n}^{(k)}\}_{k=0}^{B-1}$

---

since $\sum_{j=0}^{m-1} \epsilon_{t_i}^{(j)} (\epsilon_{t_i}^{(j)})^\top \approx m I_d$ holds for large $m$ when $\epsilon_{t_i}^{(j)}$ are *i.i.d.* random variables from $\mathcal{N}(0, I_d)$. Thus, it provides approximately isotropic exploration to all directions similar to a Wiener process, *i.e.*, the directionality of the process noise becomes less significant as $m$ increases.     $\square$

Note that the noise term is still scales by $\Delta t$ in RODE sampling (Eq. 49) rather than $\sqrt{\Delta t}$ in SDE sampling (Eq. 40).

### F.5 COMPARISON WITH PREVIOUS WORKS

Previously studies have examined the low-dimensional representations of diffusion sampling trajectories. For example, PAS (Wang et al., 2024) employs a 4D coordinate to specify the direction of a single denoising step for PF-ODE sampling (Song et al., 2021) (deterministic sampling). Differently, our approach adopts RODE sampling (based on stochastic sampling), where a single parameter $\eta_{t_i}$ suffices to determine a step. Based on these observations, we conjecture that, stochastic samplers can likewise admit low-dimensional parameterizations. We leave this for future work.

Previously, Flow-GRPO (Liu et al., 2025b) introduces stochasticity to flow matching (Albergo & Vanden-Eijnden, 2022; Lipman et al., 2022; Liu et al., 2022) by converting ODE sampling to SDE sampling. Our proposed RODE sampling offers an alternative solution to introduce randomness to ODE-based approaches, which achieves lower variance than SDE-based methods, and can be used as a stable and controllable exploration policy for RL training.

## G MORE DISCUSSION ON OUR MCTS

### G.1 OVERVIEW

Fig. 2 in the main paper presents an overview of our MCTS with online updated Beta policies. Specifically,

1. *Selection Phase*: Traverse the tree nodes with depth in range $[0, n']$, and select a not fully expanded node with maximum value-based UCB;

2. *Expansion Phase*: Sample an $a_{t_i}$ from the Beta policy of the selected node $s_{t_i}$ and create a new node $s_{t_{i+1}}$, then compute the intermediate reward $R(s_{t_i}, a_{t_i}, s_{t_{i+1}})$ with any latent reward policy. Note that the Beta policy of the expanded node is not initialized, which can also be seen as initialized as an uniform distribution. Additionally, for max-reward modeling, the pseudo-final samples obtained during the intermediate reward calculation are used to update the best trajectory if achieving higher rewards than the best-known trajectory;

3. *Simulation Phase*: Sample an $\eta_{t_{i+1}} \sim \mathcal{U}([0, 1])$ for the expanded node as the initial action, and initialize its unimodal Beta policy via mode re-parameterization. Then, perform a stochastic simulation with uniformly sampled actions sequentially until reaching a terminal state, while computing intermediate rewards along the roll-out. If a trajectory superior to the best-known one is found, the Beta policies from the current node to the root node in the MCTS tree are updated via mode re-parameterization;

4. *Backpropagation Phase*: Propagate the obtained merged reward of the terminal state back through the visited nodes with max-value policy;

5. Loop the above steps until the NFEs have elapsed.

## G.2 SUPPLEMENTARY TO MODE RE-PARAMETERIZATION

**Flaws of Directly Updating the Shape Parameters.** Directly update the shape parameters is straightforward, but hard to balance the exploration and exploitation of the Beta policy, since $\alpha$ and $\beta$ are prior counts of successes and failures rather than probabilities of sampling certain actions.

**The Concentration Controller $\zeta$.** Consider a unimodal Beta distribution $\text{Beta}(\alpha, \beta)$, where $\alpha, \beta > 1$, which are re-parameterized as $\alpha = 1 + e^a, \beta = 1 + e^b$, respectively. The mode and the variance of it are $\rho = \dfrac{\alpha - 1}{\alpha + \beta - 2}$ and $\sigma^2 = \dfrac{\alpha\beta}{(\alpha + \beta)^2(\alpha + \beta + 1)}$, respectively.

We introduce a hyper-parameter $\zeta > 0$ for concentration control, and set $a = \ln\zeta$, $b = \ln\zeta + \ln\dfrac{1 - \rho}{\rho}$. Actually, for a fixed $\rho \in (0, 1)$, the changes in $\sigma^2$ as $\zeta$ increases are non-monotonic. However, we empirically found that, setting $\zeta$ appropriately (we adopt $\zeta = 3$) can control the concentration of the unimodal Beta distribution to some extent (Appx. N.2). Nevertheless, the derivative of $\sigma^2$ with respect to $t = \ln\zeta$ changes sign at points denpent on the specific value of $\rho \in (0, 1)$. This means that, we can **not** directly determine a reasonable range for $\zeta$, so its selection relies on hyper-parameter grid search. We leave designing a better re-parameterization for the unimodal Beta distribution with its mode for future research.

## G.3 SUPPLEMENTARY TO INITIALIZATION AND ONLINE UPDATE

When a trajectory superior to the best-known one is found, the Beta policies from the current node to the root node within the MCTS tree are softly updated. Specifically, assume an action $\eta^*$ sampled from the Beta policy with mode of $\rho$ yields better performance, the mode is updated to

$$\rho' \leftarrow \begin{cases} \eta^*, & \text{if } |\eta^* - \rho| \leq \kappa \\ \rho + \kappa \cdot \text{sgn}(\eta^* - \rho), & \text{otherwise} \end{cases}, \quad \rho' \leftarrow \text{Clip}_\varepsilon^{1-\varepsilon}(\rho'), \tag{72}$$

where $\varepsilon$ is a small positive value to ensure unimodality, and $\kappa > 0$ is the update step size, which limits the amplitude amplitude to ensure policy stability (Lee et al., 2020).

## G.4 EFFICIENCY OPTIMIZATION

**Batch Processing.** The MCTS processes for all samples are conducted in a batch-wise manner, sharing a single search tree. Nodes at depth $i$ $(0 \leq i \leq n)$ can store at most a batch of states (latents). When expanding the $k$-th sample in a batch, if the node selected for expansion has a child node with an empty $k$-th state, we prioritize populating this child node rather than creating a new one. This reduces the number of tree nodes. Meanwhile, we sort each node's children in non-ascending order of their weight (the number of filled states), and place emptier nodes earlier in the children list for faster traversal.

**VRAM Management.** All node states expanded during MCTS process are stored on the GPU. To efficiently manage GPU memory (VRAM), we utilize a global LRU cache (Fricker et al., 2012) to maintain the states of tree nodes. We set a global limit $\chi$ for the number of GPU-resident states, *i.e.*, the capacity of the cache. During expansion, if the number of states on the GPU is not less than $\chi$, we offload the least recently used state in the cache to the CPU, and move the expanded one to the GPU. When accessing a node's state, if it resides on the CPU, we repeat the aforementioned cache-maintenance steps to ensure the returned latents on GPU. We set $\chi = 1000$ in our experiments.

## H EXPERIMENTAL SETTINGS, EVALUATION DIMENSIONS AND REWARD FUNCTIONS

### H.1 IMPLEMENTATION

We modified the the HuggingFace diffusers (von Platen et al., 2022) library for implementation.

### H.2 SETTINGS

**Environment.** All experiments are conducted on a *Ubuntu 20.04.6 LTS* system with *128× Intel(R) Xeon(R) Gold 6430* CPU and *1× RTX 6000 Ada Generation GPU with 48G VRAM*.

**Text-to-Image Models.** We adopt the following pre-trained models as our base models for alignment:

- *stabilityai/sd-turbo* (Sauer et al., 2024b) [1];
- *compvis/stable-diffusion-v-1-4-original* (Sauer et al., 2024b) [2];
- *stabilityai/stable-diffusion-xl-base-1.0* (Podell et al., 2023) [3];
- *PixArt-alpha/PixArt-XL-2-1024-MS* (Chen et al., 2023b) [4].

We use the full-precision versions of the models, because we empirically found that, the fp16 version can cause VAE (Kingma et al., 2013; 2021) precision issues, making the decoded images completely black (Madebyollin, 2024).

We do **not** adopt the current SoTA SD 3 (Esser et al., 2024), SD 3.5 (Esser et al., 2024), FLUX.1 (Labs et al., 2025; Labs, 2024) or Hunyuan DiT (Li et al., 2024b) models, as they are trained under flow-matching objectives (Albergo & Vanden-Eijnden, 2022; Lipman et al., 2022; Liu et al., 2022), and are incompatible with the DDIM scheduler (Song et al., 2020). We leave generalization to flow-matching schedulers for future work.

**Hyper-Parameters.** We adopt the following hyper-parameters unless specified.

- The global random seed is set to $42$;
- The seeds for initial noise is $\{0, 1, 2, \cdots\}$;
- For SD-Turbo and SD v1.4, the sample resolution is $512\times512$, with a CFG (Ho & Salimans, 2022) scale of 6.5; For SDXL base and PixArt-$\alpha$, the sample resolution is $1024\times1024$, with a CFG scale of 4.5. The *negative_prompt* is "low quality, blurry, ugly, oversaturated" for all models. Particularly, for SDXL base and PixArt-$\alpha$, the *prompt_2* and *nagative_prompt_2* is set to an empty string;
- For MCTS, the hyper-paramter that balance exploration and exploitation $\lambda_{\text{exploitation}} = 2.0$, the selection depth limit is $n' = \left\lceil \dfrac{4n}{5} \right\rceil$, and simulation actions are uniformly sampled from $[0, 1]$;
- For the Beta policies, the update step size $\kappa = 0.1$, and the clamp epsilon to $\varepsilon = 10^{-8}$. The concentration control scalar is set to $\zeta = 3$ unless specially specified.

### H.3 METRIC

#### H.3.1 MPD

We use LPIPS (Zhang et al., 2018) to evaluate the perceptual similarity between two images, where values closer to 0 indicate greater visual similarity. We calculate the mean pairwise distance

---

[1] https://huggingface.co/stabilityai/sd-turbo
[2] https://huggingface.co/CompVis/stable-diffusion-v-1-4-original
[3] https://huggingface.co/stabilityai/stable-diffusion-xl-base-1.0
[4] https://huggingface.co/PixArt-alpha/PixArt-XL-2-1024-MS

(MPD) (Kim et al., 2025) to assess the diversity of a set of samples $\{x_i\}_{i=0}^{N-1}$:

$$\text{MPD} = \frac{1}{N(N-1)} \sum_{i=0}^{N-2} \sum_{j=i+1}^{N-1} \text{LPIPS}(x_i, x_j), \tag{73}$$

where larger MPD means higher diversity. The MPD of generation results per prompt is calculated and averaged across all prompts to gauge the diversity of a specific inference-time scaling paradigm.

### H.3.2 PARAMETER EFFICIENCY

The number of parameters adopted for the scaling process is defined as

$$\#\text{Param.} = m \cdot n \cdot \dim(\text{action}), \tag{74}$$

where $\dim(\text{action})$ is the dimension of the adopted action. Specifically, in our experiments,

- If scaling with $\eta_{t_i}$, then $d_\eta \overset{\triangle}{=} \dim(\eta_{t_i}) = 1$;

- If scaling with $\epsilon_{t_i}$, then $d_\epsilon \overset{\triangle}{=} \dim(\epsilon_{t_i})$, which is equal to $4 \times 64 \times 64$ for SD v1.4, or $4 \times 128 \times 128$ for SDXL base and PixArt-$\alpha$.

The *Parameter Efficiency* (PE, $\uparrow$) quantifies how available parameters are efficiently used for the scaling process, which writes

$$\text{PE} = \frac{\text{reward} - \text{baseline}}{\#\text{Param.}}, \tag{75}$$

where *reward* is the maximum achieved reward, and *baseline* is the baseline reward for normalization.

### H.3.3 EVALUATION DIMENSIONS

We consider the following dimensions to evaluate our MCTS:

- *Best Reward*: The averaged highest reward achieved, which is the maximum final reward for cumulative-reward modeling, while it is the maximum encountered reward for max-reward modeling. It is rounded to 4 decimal places;

- *NFE-dynamics*: The averaged NFE consumed for computing the transition dynamics. It is rounded to 2 decimal places, so do *NFE-intermediate* and *NFE-final*;

- *NFE-intermediate*: The averaged NFE consumed for calculating intermediate rewards;

- *NFE-final*: The averaged NFE consumed for calculating final rewards. Note that the NFEs needed for calculating intermediate rewards are included;

- *Time Cost*: The averaged wall-clock time required per sample to obtain *Best Reward*. It is rounded to the nearest integer.

For each prompt, we run our MCTS with $m$ groups of process noise and take the optimal results (the samples with highest best rewards), and average the evaluation metrics like *Best Reward* across these results. If multiple samples with the same best rewards exist, the one with the smallest NFEs and the shortest time will be retained. Particularly, since our implement MCTS in batch (Appx. G.4), the time cost per sample is influenced by other samples in the batch. We record the wall-time consumed to achieve the best reward for each sample individually, and report the averaged time cost.

### H.3.4 REWARD FUNCTIONS

We adopt the following models or metrics as reward functions to align DMs with:

- **HPS v2** (Human Preference Score v2) (Wu et al., 2023b;a) [5]: Quantifies the human aesthetics preference of images. Higher scores indicate better aesthetics. Particularly, we report 100 times of the original reward;

---

[5] https://github.com/tgxs002/HPSv2

- **IR** (ImageReward) (Xu et al., 2023a) [6]: Evaluates the human preference of images. Higher scores indicate higher preference;

- **PS** (PickScore) (Kirstain et al., 2023) [7]: Predicts human preference of images. Higher scores indicate higher preference. Note that we evaluate the PS in sample-wise manner, instead of aggregating the samples into a batch and computing in a single pass;

- **CLIP Score** (Radford et al., 2021; Hessel et al., 2021) [8]: Evaluates the semantic alignment between the text descriptions and their generated images . Higher scores indicate higher alignment. We adopt *laion/CLIP-ViT-H-14-laion2B-s32B-b79K* [9] for feature extraction;

- **Compressibility Reward** (CR): The file size obtained by compressing the image with the standard JPEG compression (Raid et al., 2014) from Python's Pillow library (Appx. H.3.6), with a *quality* parameter of 30 and enabling optimization;

- **LAPV** (Laplacian Variance, LAPV) (Memon et al., 2015; GeeksforGeeks, 2024): The variance of the Laplacian matrix (GeeksforGeeks, 2024) of an image, which is an **unbounded** value that quantifies the sharpness of images. The larger, the sharper;

- **CCR** (Color Channel Reward, CCR) (Eyring et al., 2024): Measures how much an RGB image's overall hue leans towards red (R), green (G) or blue (B). For an RGB image $X_{3 \times h \times w}$ with height $h$ and width $w$, let the pixel value at coordinate $(i, j)$ in channel $c$ $(0 \le c \le 2)$ be $X_{c,i,j} \in [0, 1]$ (normalized). If the target channel is $c$, and the other two channels are $\overline{c}_1$ and $\overline{c}_2$, then the degree to which $X$ leans towards channel $c$ can be quantified as

$$\text{CCR}(c) = \sum_{i,j} X_{c,i,j} - \sum_{i,j} X_{\overline{c}_1,i,j} \sum_{i,j} X_{\overline{c}_2,i,j}. \tag{76}$$

### H.3.5 REWARD PRE-PROCESSING

Some rewards should be pre-processed to meet the non-negativity requirement for MCTS values. We first clip the rewards in a specific *range* (if specified), then perform normalization by diving them by a *scaling factor*, and finally add a *bias scalar*. These constants are listed in Tab. 11.

### H.3.6 METRIC COMPUTATION

We utilize the following third-party libraries for metric computation:

- *Human Preference Score v2* (1.2.0) (Wu et al., 2023a) [10];

- *pytorch-fid* (0.3.0) (Yu et al., 2021) [11];

- *lpips* (0.1.4) (Zhang et al., 2018) [12];

- *opencv-python* (4.11.0.86) (OpenCV, 2024) [13];

- *open-clip-torch* (2.23.0) (Ilharco et al., 2021) [14];

- *Pillow* (2.23.0) [15].

Table 11: **Reward pre-processing.**

| Reward Type | Clip Range | Scaling Factor | Bias Scalar |
|---|---|---|---|
| HPS v2 | / | 1.0 | 0.0 |
| PS | / | 1.0 | 0.0 |
| IR | $[-2, 2]$ | 1.0 | 2.0 |
| CLIP Score | / | 1.0 | 0.0 |
| CR | / | 1.0 | 3.0 |
| LAPV | / | 1.0 | 0.0 |
| CCR | / | 1.0 | 2.0 |

---

[6]https://github.com/THUDM/ImageReward
[7]https://github.com/yuvalkirstain/PickScore
[8]https://github.com/mlfoundations/open_clip
[9]https://huggingface.co/laion/CLIP-ViT-H-14-laion2B-s32B-b79K
[10]https://github.com/tgxs002/HPSv2
[11]https://github.com/mseitzer/pytorch-fid
[12]https://github.com/richzhang/PerceptualSimilarity
[13]https://github.com/opencv/opencv-python
[14]https://github.com/mlfoundations/open_clip
[15]https://github.com/python-pillow/Pillow

### H.4 Overview of Experimental Composition

Sec. 5.2 in the main paper and Appx. I provide the empirical foundation for our RODE-based sampling; Sec. 5.4 and Appx. J align DMs with aesthetics, and makes comparison between different inference-time scaling paradigms; Sec. 5.5 and Appx. K align DMs with semantics, and benchmark our method against Z-Sampling (Bai et al., 2024); Sec. 5.6 and Appx. L show our method's ability to align with composite rewards; Appx. M show the generalization of our method across various latent reward policies; Sec. 5.7, Appx. N and Appx. O presents ablation studies and applications. Appx. P presents failure cases.

## I Experiments: Supplementary to RODE Sampling

### I.1 Supplementary to Trajectory Simulation

In Fig. 1 in Sec. 5.2 of the main paper, we simulate the trajectories produced by three sampling paradigms using an Ornstein-Uhlenbeck (OU) process (Karatzas & Shreve, 2012)

$$dx_t = -\mu x_t dt + \sigma_t dw_t \tag{77}$$

with the discretized step

$$x_{t_{i+1}} = x_{t_i} - \mu x_{t_i} \Delta t + \sigma_{t_i} \sqrt{\Delta t} \epsilon_{t_i} \tag{78}$$

for $n$ times sequentially.

Specifically,

- For PF-ODE sampling (Song et al., 2021), we fix $\sigma_{t_i} = 0$ for $i = 0, 1, \cdots, n-1$;
- For SDE sampling (Song et al., 2021), we fix $\sigma_{t_i} = 1$, and re-sample the process noise $\epsilon_{t_i}$ at every timestep $t_i$;
- For RODE sampling, we pre-sample $\{\epsilon_{t_i}\}_{i=0}^{n-1}$ and re-use them across trajectories. We sample $\sigma_{t_i}$ from $\mathcal{U}([0, 1])$ at each timestep $t_i$.

We set $n = 1000$, $\mu = 1.5$, $\Delta t = \frac{1}{n}$. The background heatmap depicting the evolution of data distributions is generated by applying a Gaussian filer with $\sigma = 3$ to the 2D histograms of 5k SDE trajectories.

### I.2 Supplementary to Sample Diversity

**Settings.** We randomly sample 100 prompts from the HPD v2 dataset (Wu et al., 2023a). For each prompt, $N = 5$ images are generated under the conditions in Tab. 13 using 2-step SD-Turbo (Sauer et al., 2024a) and SD v1.4 (Rombach et al., 2022) with 15, 20, and 25 steps, respectively. we adopt the $m = 3$ sets of process noise for Aug. RODE sampling. The MPD of generation results per prompt is calculated and averaged across all prompts to gauge the diversity.

Table 12: **Seeds used for each paradigm.**

| Paradigm | Initial Noise | Process Noise |
|----------|---------------|---------------|
| a.0 | $\{0, 1, 2, 3, 4, 5\}$ | / |
| a.1 | $\{0, 1, 2, 3, 4, 5\}$ | $\mathcal{U}([3072, 4095])$ |
| a.2 | $\{0, 1, 2, 3, 4, 5\}$ | $\{3072, 3073, \cdots\}$ |
| b | $\{0, 0, 0, 0, 0\}$ | $\mathcal{U}([3072, 4095])$ |
| c.0 / c.1 | $\{0, 0, 0, 0, 0\}$ | $\{3072, 3073, \cdots\}$ $\{4096, 4097, \cdots\}$ $\{5120, 5121, \cdots\}$ |

**Polices of Different Inference-Time Scaling Paradigms.** The initial noise and process variance are easy to fix by keeping their values constant. However, it is infeasible to fix the process noise by setting all $\{\epsilon_{t_i}\}_{i=0}^{n-1}$ to a constant vector $\epsilon$, because this time-invariant perturbation will cause large accumulative error in SDE sampling, leading to significant divergence of true trajectories and producing completely noisy images in multi-step inference scenarios. For fair comparison, we set seeds for each paradigm as in Tab. 12. Particularly, for *paradigm c*, we run with $m = 3$ sets of process noise, and compute MPD for each set of the process noise **respectively**, which is averaged across the $m = 3$ sets as the result.

Table 13: **Policies and MPDs (↑) of different inference-time scaling paradigms.** Cells with a red / orange / yellow background: the best / second-best / third-best performance for each model (column), respectively.

| Paradigm | Index | Sampler | $\{z_{t_0}^{(j)}\}_{j=0}^{N-1}$ | $\{\epsilon_{t_i}\}_{i=0}^{n-1}$ | $\{\eta_{t_i}\}_{i=0}^{n-1}$ | SD-Turbo | | SD v1.4 | |
|---|---|---|---|---|---|---|---|---|---|
| | | | Components | | | 2-step | 15-step | 20-step | 25-step |
| a) initial-noise | **a.0)** | ODE | random | × | $\eta_{t_i} = 0$ | 0.5867 | 0.7542 | 0.7584 | 0.7594 |
| | **a.1)** | SDE | random | random | $\eta_{t_i} = 1.0$ | 0.6437 | 0.7653 | 0.7617 | 0.7650 |
| | **a.2)** | Aug. RODE | random | $m$ sets | $\eta_{t_i} = 1.0$ | 0.6200 | 0.6861 | 0.6887 | 0.6914 |
| b) process-noise | | SDE | fixed | random | $\eta_{t_i} = 1.0$ | 0.3823 | 0.6848 | 0.6939 | 0.6946 |
| c) process-noise -variance | **c.0)** | Aug. RODE | fixed | $m$ sets | $\eta_{t_i} \sim \mathcal{U}([0,1])$ | **0.1628** | **0.4717** | **0.4852** | **0.4899** |
| | **c.1)** | Aug. RODE | fixed | $m$ sets | $\eta_{t_i} \sim \mathcal{U}([0.5,1])$ | 0.1295 | 0.3556 | 0.3649 | 0.3686 |

**Results and Analysis.** We provide analysis of results in Tab. 13 as follows:

1. Under random initial noise, all sampling paradigms exhibit high sample diversity. Specifically, the initial noise (*paradigm a*) predominately determines the sampling diversity, especially in few-step inference (*e.g.*, $n = 2$ and $n = 15$);

2. The process noise (*paradigm b*) impacts sampling diversity more significantly than the process variance (*paradigm c*);

3. Though we fixed the process noise and randomize the variance, *paradigm c* shows no mode collapse (Barsha & Eberle, 2025) but mid-to-high diversity, especially when $n$ is large;

4. As the number of inference steps increases, diversity rises across all paradigms. Particularly, in few-step inference, *paradigm b* and *paradigm c* show much lower sampling diversity than *paradigm a*. However, as $n$ increases, the diversity in *paradigm b* and *paradigm c* significantly rises, with *paradigm b* even surpassing *paradigm a.2*. Our Aug. RODE sampling (*paradigm c.0*) even achieves an MPD of nearly $0.5$ when generated with 25-step SD v1.4 (Rombach et al., 2022). It indicates that, the sampling diversity in multi-step inference can be boosted by fully utilizing the process noise and process variance at each denoising step to perturb the denoising trajectories;

5. Interestingly, for Aug. RODE sampling, adopting a broader range $[0, 1]$ for $\eta_{t_i}$ (*paradigm c.0*) yields higher diversity than a larger scale $[0.5, 1]$ (*paradigm c.1*). On the one hand, this inspires us to adopt $\eta_{t_i} \sim \mathcal{U}([0, 1])$ in our experiments. On the other hand, this highlights that, the exploration can by easily adjusted by altering the perturbation magnitude (*i.e.*, the range of $\eta_{t_i}$);

6. Our Aug. RODE sampling (*paradigm c*) partially integrates the ability of SDE sampling (*paradigm b*);

7. The empirical observations above suggest that, our method is more suitable for multi-step inference scenarios due to broader exploration.

In summary, the augmented RODE sampling successfully balances the controllability of ODE sampling and the exploration of SDE sampling, serving as a qualified and efficient sampler for denoising trajectory search.

### I.3 Supplementary to Empirical Distribution

**Motivation.** To deploy RODE sampling in practical generation tasks, we should additionally verify that, its empirical distribution does not excessively deviate from the real data distribution even with larger errors (Prop. 5, Prop. 6).

**Settings.** The MS-COCO 5k validation set (Lin et al., 2014) comprises 5,000 images, each paired with 5 or 6 captions for a total of 25,010 prompts. Selecting SD v1.4 Rombach et al. (2022) as the base model, we generate $N = 1$ image per prompt with various inference steps using ODE (Song et al., 2020), SDE (Song et al., 2021) and Aug. RODE sampling, respectively. We adopt $m = 3$ sets of process noise for SDE sampling and Aug. RODE sampling, the zero-shot FID is computed for each set of the process noise **respectively**, and is averaged across the $m = 3$ sets as the result.

**Results and Analysis.** Results in Tab. 3 demonstrate that,

1. Aug. RODE sampling attains FIDs that lie between those of ODE and SDE sampling, indicating that the novel paradigm using $\eta_{t_i}s$ as random variables does not significantly shift the distribution. We attribute this to the close resemblance of the drift term of Aug. RODE to that of the original SDE (Appx. F.3). Additionally, we empirically find it hard to distinguish RODE-sampled results from standard SDE/ODE-sampled ones (*e.g.*, Fig. 1);

2. The ordering $\text{FID}_{\text{SDE}} < \text{FID}_{\text{Aug. RODE}} < \text{FID}_{\text{ODE}}$ highlights the forgetting effect induced by injecting process noise within stochastic sampling (Appx. D.4.3, Appx. F.3);

3. This empirically corroborate the observations in Song et al. (2021) and Karras et al. (2022) that, given sufficient NFE, introducing process noise improves the sample quality.

### I.4 VISUALIZATION OF SAMPLING TRAJECTORIES

**Visualization.** We detail the estabilishment of the polar plot for visualizing DM's high-dimensional denoising trajectories. Consider a series of concentric semicirles $\{\Gamma_i\}_{i=1}^n$ in polar coordinates, with an angular range of $[0, \pi]$, and radii of $\frac{1}{n}, \frac{2}{n}, \cdots, 1$, respectively. The origin, as the 0-th level semi-circle, represents the initial noise $z_{t_0}$ which is the denoising start point. The $i$-th ($1 \leq i \leq n$) semi-circle denotes the data manifold of the noise level at time step $t_i$. By linear mapping each $\eta_{t_i} \in [0, 1]$ to $\eta'_{t_i} \in [0, \pi]$, the denoising trajectory parameterized by the action sequence $\eta_{t_0, \cdots, t_{n-1}}$ can be visualized as follows: starting from the origin, at each $i$-th step ($0 \leq i \leq n - 1$), moving from $i$-th level manifold along the $\eta'_{t_i}$-angle direction to $(i+1)$-th level manifold, eventually reaching $n$-th level and obtaining a final latent $z_{t_n}$.

**Settings.** We visualize the sampling trajectories and the corresponding qualitative results for the prompt "`A symmetrical oil painting of two waterfalls in a dense forest.`" in Fig. 4. It demonstrate that,

1. The trajectory of deterministic DDIM (Song et al., 2020) consistently follows the 0-angle direction, the one of DDPM (Ho et al., 2020) consistently follows the $\pi$-angle direction, while our RODE sampling navigates time-variant directions;

2. Our RODE sampling explores various paths from the origin to the $n$-th level data manifold (the outmost semi-circle), seeking for higher rewards.

A comprehensive study on the relationship between action sequences and sampling results is left for future work.

**Comparison with Previous Works.** Previous works (Zhou et al., 2024a; Chen et al., 2024; Wang et al., 2024; Wang & Vastola, 2023) that low-dimensional visualizations of diffusion sampling trajectories are confined to deterministic PF-ODE (Song et al., 2021) sampling. Our approach offers a novel 2D perspective that renders stochastic sampling trajectories more readily and intuitively interpretable than prior representations, advancing the visualization of stochastic sampling paradigm.

## J EXPERIMENT: SUPPLEMENTARY TO ALIGNING WITH AESTHETICS

### J.1 SETTINGS

**The Dataset, Model and Reward Function.** We randomly sample 50 prompts from the HPD v2 dataset (Wu et al., 2023a), and sample $N = 2$ images for each prompt. We align 15-step SD v1.4 (Rombach et al., 2022) to HPS v2 (Wu et al., 2023a), aming at enhancing the sample aesthetics.

**NFE Budgets.** The limit of NFE dynamics is set to 999.

**Methods.** We contend that, existing inference-time scaling approaches with $\epsilon_{t_i}$ actions (Oshima et al., 2025; He et al., 2025; Liu et al., 2025a; Mao et al., 2025; Singhal et al., 2025) all boil down to uniformly sampling $\epsilon_{t_i} \in \mathbb{R}^{c \times h \times w}$ and applying diverse and elaborate optimizations. We strip away

additional embellishments, and essentially evaluate $\epsilon_{t_i}$-action scaling in isolation. Specifically, we instantiate beam search (BS) (Fernandes et al., 2025; Oshima et al., 2025), global search (GS, the BS with a single beam) and MCTS Coulom (2006); Browne et al. (2012) as base planners, and equip each with either $\epsilon_{t_i}$ or $\eta_{t_i}$ actions. Particularly, for $\epsilon_{t_i}$-action methods, we adopt $m = 3$ random sets of process noise generated using random seeds sampled from $\mathcal{U}([3072, 4095]), \mathcal{U}([4096, 5119])$, and $\mathcal{U}([5120, 6143])$, respectively, and retain the highest rewards (similar to $m = 3$ fixed sets of process noise that $\eta_{t_i}$-action methods do). In summary, we make comparison between:

- Vanilla DDIM (Song et al., 2020) sampling as baseline;

- Best-of-$N$ (Ma et al., 2025), which is **not** directly comparable as it is $z_{t_0}$-action and falls outside our setting, but is included here as a widely used baseline. To respect the NFE budget, we require $n \cdot N \leq 999$, yielding $N \leq 66$; thus we set $N = 66$. Note that this $N$ refers to the number of candidate initial noise in best-of-$N$, which is **not** related to the $N$ appearing in "sample $N = 2$ images for each prompt";

- Inference-step scaling paradigm (Nichol & Dhariwal, 2021) using DDPM (Ho et al., 2020) and DDIM (Song et al., 2020). Note that the DDPM sampling with $m = 3$ sets of process noise can also be seen as an $\epsilon_{t_i}$-action scaling. For both Best-of-$N$ and inference-step scaling, the reported *Time Cost* refers exclusively to the sampling time and does **not** include the time required for reward computation. This is because *Time Cost* is defined as the average wall-clock time per sample needed to obtain the *Best Reward*, and these two methods must generate a fixed number of samples regardless of when the best rewards are found;

- Beam search (Fernandes et al., 2025; Oshima et al., 2025) with $\epsilon_{t_i}/\eta_{t_i}$ actions that employs the conventional max-final-reward modeling and *immediate-ddim* as latent reward policy. It should be noted that, our definitions of $B$ and $K$ might slightly differ from those in the original paper (Oshima et al., 2025). Specifically, for a fixed initial noise, we maintain $B$ beams at every depth of the search tree. Each beam stochastically spawns $K$ candidate nodes, of which only the top-$B$ performers are retained for expansion at the next layer. Particularly, at depth 1, we directly initialize with $B \cdot K$ candidates in one shot. Besides, to respect the NFE budget, we enforce $n \cdot B \cdot K \leq 999$, yielding $B \cdot K \leq 66$. Following Oshima et al. (2025), we restrict $B$ and $K$ to powers of two, which further tightens this constraint to $B \cdot K$. we exhaustively enumerate all $(B, K)$ pairs that satisfies $B \cdot K = 64$, and report the best-performing configuration. We also report the performance of greedy search (GS), which corresponds to $(B, K) = (1, 64)$;

- Our MCTS with $\epsilon_{t_i}/\eta_{t_i}$ actions. Particularly, for MCTS-eps ($\epsilon_{t_i}$ action), we adopt settings identical to McTS-eta ($\eta_{t_i}$ action), except the expansion policy for $\epsilon_{t_i}$ being changed as described above.

**Notes.** It is worth mentioning that, since deterministic DDIM (Song et al., 2020) does not employ process noise, a direct comparison with DDPM (Ho et al., 2020) and other methods — which use $m$ sets of process noise — is unfair if strictly speaking. Nevertheless, we retain this evaluation to:

- Highlight the performance gains afforded by leveraging multiple sets of process noise;

- Treat the performance of the deterministic DDIM as the baseline from which relative improvements are computed.

## J.2 RESULTS AND ANALYSIS

**Comparison on the Scaling Process.** Fig. 5 shows that, increasing inference steps yields limited sample quality enhancement, and can even degrade performance with too many steps, which is consistent with the observations in Nichol & Dhariwal (2021); Li et al. (2023). In contrast, our MCTS achieves higher rewards and faster convergence, which underscores the superiority of allocating the NFE budgets across multiple denoising trajectories rather than concentrating it on a single one. BS methods are omitted because they only yield samples at the maximum depth.

**Generalization to Other Aesthetic Scores.** We also evaluate the optimization process of HPS v2 (Wu et al., 2023a) with PickScore (PS) (Kirstain et al., 2023) and ImageReward (IR) (Xu et al.,

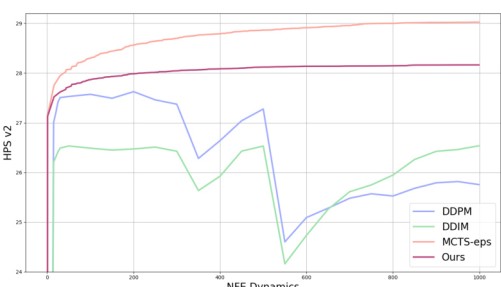
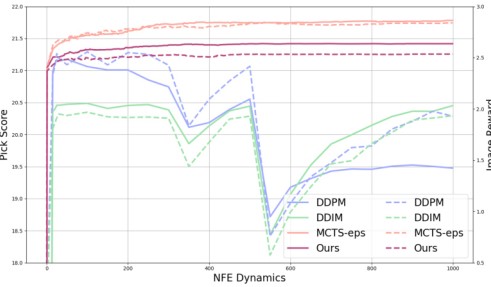

Figure 5: **Comparison on the optimization process of HPS v2 between different methods.**

Figure 6: **Comparison of PS and IR between our RODE-based scaling and inference-step paradigm.** Solid/dashed lines for PS/IR, respectively.

2023a). Results in Fig. 6 show improvements in two additional human aesthetics preference predictor that weren't directly optimized, indicating that our MCTS truly enhances image aesthetics rather than just over-optimizing HPS v2. We omit the evaluation of Aesthetic Score (Romain Beaumont, 2022) here and in Appx. O.2, since it gives discrete scores for each image.

Details of quantitative results for inference-step scaling and our method are provided in Tab. 15 and Tab. 16, respectively.

**Quantitative Results and Parameter Efficiency.** Quantitative results are shown in Tab. 14, where relative improvement is computed based on 15-step DDIM. It demonstrates that,

1. GS, BS and our MCTS all outperform the scaled DDIM and DDPM;

2. Ours (MCTS-eta) surpasses traditional inference-step scaling. Note that scaling DDPM is essentially an $\epsilon_{t_i}$-action method;

3. When scaling with $\eta_{t_i}$, ours outperform both GS and BS, highlighting MCTS's ability to allocate NFEs judiciously, and Beta policies' capacity for accurate low-dimensional search;

4. Although $\epsilon_{t_i}$-action methods achieve the highest absolute score, its relative improvement is only $\sim 2\times$ better than $\eta_{t_i}$-action ones, but with $\sim 16\text{k}\times$ more parameters, revealing low parameter efficiency and severe parameter redundancy;

5. When naively transferring our MCTS to $\epsilon_{t_i}$-actions, performance drops below GS and BS, underscoring the limitation of vanilla MCTS in high-dimensional continuous action spaces (Bianchi et al., 2023);

6. Ours runs significantly faster than GS and BS. This is because GS and BS expand nodes in strictly increasing depth order. In contrast, MCTS can revisit shallow but high-value nodes in later search, and these nodes spend more NFEs for expansion and simulation;

7. MCTS obviates the need to enumerate $(B, K)$ pairs, which is more adaptive for deployment.

**Qualitative Results.** Samples displayed in Fig. 1 (a) are generated with text prompt "A tea kettle sits on the burner of stove.", where samples for DDPM and DDIM are selected from the inference steps that yields the

Table 16: **Aesthetic scores of our method.**

| Metric | HPS v2 | PS | IR |
|---|---|---|---|
| Reward | 28.1639 | 21.4383 | 2.5350 |
| Improvement (%) | 7.45 | 5.27 | 40.23 |

highest HPS v2. It can be observed that, trajectory search-based methods render more reasonable kettles and avoid artifacts (*e.g.*, the fire on a cup in DDPM). Note that it is hard to distinguish samples derived by SDE sampling and RODE sampling.

More qualitative results in Fig. 7 highlight our superior visual aesthetics over baselines. The text prompts are "Yoda performing at Woodstock.", "A dreamlike scene with a vaporwave aesthetic.", "A manga-style illustration of a submachine gun in 2050 by Moebius and Stephan Martiniere.", "Psytrance artwork by Lisa Frank.", and "The image is a digital art poster sized in the

Table 14: **Comparison between different methods when optimizing HPS v2.** "*Relative Improv.*": relative improvement.

| Method | Action Type | Best Reward | Relative Impro. (%) | #Param. | Param. Efficiency | Time Cost (s) |
|---|---|---|---|---|---|---|
| DDIM (15 steps, baseline) | / | 26.2122 | / | / | / | 147 |
| Best-of-N (DDIM, $N = 66$) | $z_{t_0}$ | 28.4056 | 8.37 | $66 \cdot d_\epsilon$ | 2.03e-6 | 4480 |
| Best-of-N (DDPM, $N = 66$) | $z_{t_0}$ | 28.4297 | 8.46 | $66 \cdot d_\epsilon$ | 2.05e-6 | 4513 |
| DDIM (999 steps) | $\epsilon_{t_i}$ | 26.5362 | 1.24 | / | / | 6694 |
| DDPM (200 steps) | $\epsilon_{t_i}$ | 27.6256 | 5.39 | $3 \cdot 200 \cdot d_\epsilon$ | 1.44e-7 | 1382 |
| GS ($B = 1, K = 64$) | $\epsilon_{t_i}$ | 29.7386 | 13.45 | $3 \cdot 15 \cdot d_\epsilon$ | 4.78e-6 | 5684 |
| BS ($B = 8, K = 8$) | $\epsilon_{t_i}$ | 30.2095 | 15.25 | $3 \cdot 15 \cdot d_\epsilon$ | 5.42e-6 | 5469 |
| **MCTS-eps** | $\epsilon_{t_i}$ | 29.0241 | 10.73 | $3 \cdot 15 \cdot d_\epsilon$ | 3.81e-6 | 1437 |
| GS ($B = 1, K = 64$) | $\eta_{t_i}$ | 27.8116 | 6.10 | $3 \cdot 15 \cdot d_\eta$ | 0.0355 | 5697 |
| BS ($B = 8, K = 8$) | $\eta_{t_i}$ | 28.0833 | 7.14 | $3 \cdot 15 \cdot d_\eta$ | 0.0416 | 5273 |
| **Ours** (MCTS) | $\eta_{t_i}$ | 28.1644 | 7.45 | $3 \cdot 15 \cdot d_\eta$ | 0.0434 | 2736 |

Table 15: **Quantitative results of the inference-step scaling for SD v1.4.** HPS v2 ($\uparrow$), PickScore ($\uparrow$), and Image Reward ($\uparrow$), from top to bottom.

| Steps | 15 / 250 / 650 | 20 / 300 / 700 | 25 / 350 / 750 | 30 / 400 / 800 | 50 / 450 / 850 | 100 / 500 / 900 | 150 / 550 / 950 | 200 / 600 / 999 |
|---|---|---|---|---|---|---|---|---|
| DDPM | 27.0064 | 27.1961 | 27.4134 | 27.5067 | 27.5283 | 27.5727 | 27.4923 | 27.6256 |
|  | 27.4605 | 27.3734 | 26.2814 | 26.6403 | 27.0405 | 27.2764 | 24.6028 | 25.0923 |
|  | 25.2805 | 25.4803 | 25.5706 | 25.5252 | 25.6798 | 25.7909 | 25.8164 | 25.7555 |
| DDIM | 26.2122 | 26.3084 | 26.4150 | 26.4887 | 26.5333 | 26.4881 | 26.4502 | 26.4725 |
|  | 26.5097 | 26.4264 | 25.6347 | 25.9236 | 26.4288 | 26.5312 | 24.1603 | 24.7339 |
|  | 25.2594 | 25.6117 | 25.7486 | 25.9464 | 26.2603 | 26.4241 | 26.4608 | 26.5362 |
| DDPM | 20.9863 | 21.0963 | 21.1661 | 21.1489 | 21.1745 | 21.0616 | 21.0089 | 21.0098 |
|  | 20.8538 | 20.7450 | 20.1144 | 20.1872 | 20.3922 | 20.5539 | 18.7192 | 19.1784 |
|  | 19.3145 | 19.4295 | 19.4641 | 19.4592 | 19.5066 | 19.5255 | 19.5041 | 19.4777 |
| DDIM | 20.3647 | 20.3925 | 20.4592 | 20.4584 | 20.4728 | 20.4875 | 20.4081 | 20.4550 |
|  | 20.4686 | 20.3856 | 19.8591 | 20.1367 | 20.3700 | 20.4452 | 18.4348 | 19.0692 |
|  | 19.5242 | 19.8527 | 19.9955 | 20.1458 | 20.2830 | 20.3644 | 20.3636 | 20.4516 |
| DDPM | 2.3326 | 2.4108 | 2.5399 | 2.4956 | 2.4315 | 2.5571 | 2.4299 | 2.5497 |
|  | 2.5331 | 2.4270 | 1.8368 | 2.0955 | 2.2742 | 2.4163 | 0.7615 | 1.0822 |
|  | 1.3388 | 1.4753 | 1.6227 | 1.6395 | 1.8046 | 1.8789 | 1.9772 | 1.9300 |
| DDIM | 1.8078 | 1.8577 | 1.9151 | 1.9508 | 1.9366 | 1.9685 | 1.9234 | 1.9172 |
|  | 1.9220 | 1.9097 | 1.4397 | 1.6762 | 1.9015 | 1.9294 | 0.5751 | 0.9958 |
|  | 1.2412 | 1.4644 | 1.4950 | 1.6642 | 1.7719 | 1.8908 | 1.9116 | 1.9339 |

style of Utamaro Kitagawa featuring Lil Wayne.", respectively. It can be observed that, trajectory search-based methods yield samples that are visually more appealing than those produced by inference-step scaling, whereas ours additionally avoids color over-saturation.

Figure 7: **Qualitative results of optimizing HPS v2.**

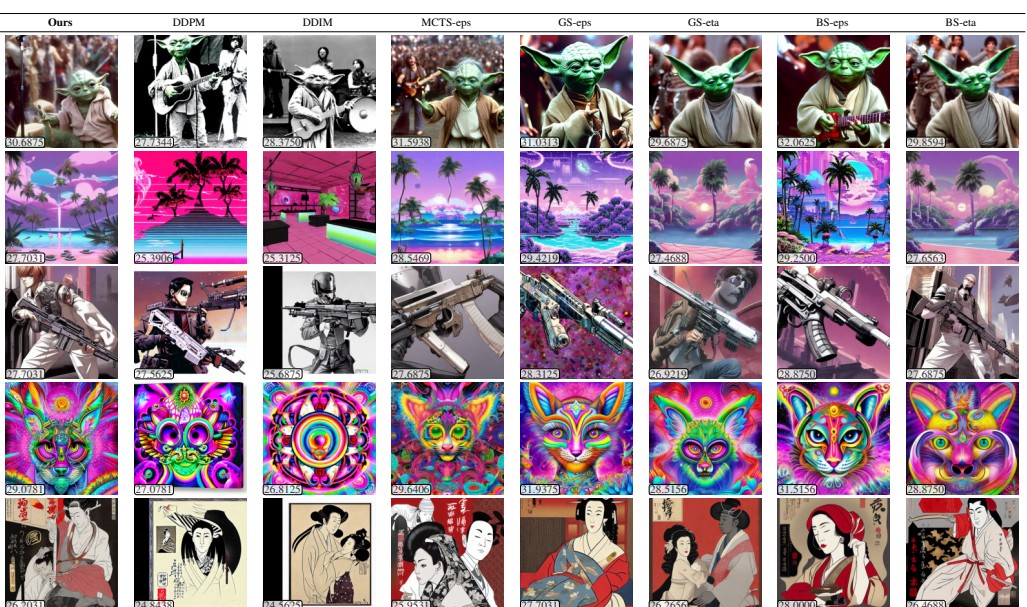

**Note.** We provide the following clarification regarding the global search capabilities of our method and absolute best rewards:

1. Given the limited expressive capacity of 1-D $\eta$-actions, $\eta$-based search behaves more like a *local search* method, and thus it is natural that its absolute best rewards may fall short of $\epsilon$-based *global search* methods. However, experiments in Sec. 5.4 and Sec. 5.5 show that even with an extremely low-dimensional action space, our method outperforms several eps-action baselines (e.g., inference-step scaling and Z-Sampling), highlighting its high parameter efficiency;

2. As noted in Appx. Q.3, future work may explore *global–local hybrid strategies*, e.g., using $\epsilon$-actions to identify high-reward regions, followed by $\eta$-actions for refined local search;

3. Nevertheless, the difference between the *relative improvement* achieved by $\eta$- and $\epsilon$-based methods — when viewed through the lens of their vastly different parameter dimensionalities — reveals a broader conceptual insight: *parameter efficiency* plays a crucial role in search-based alignment.

## K EXPERIMENT: SUPPLEMENTARY TO ALIGNING WITH SEMANTICS

### K.1 SETTINGS

**The Dataset, Model and Reward Function.** We randomly sample 30 prompts from the *colors*, *descriptions* and *positional* categories in DrawBench (Saharia et al., 2022) dataset, and align 30-step SDXL base (Podell et al., 2023) to CLIP score (Radford et al., 2021; Hessel et al., 2021), aiming at enhancing semantic alignment.

**NFE Budgets.** The limit of NFE dynamics is set to 999.

**Methods.** We benchmark our method against:

- 30-step DDPM (Ho et al., 2020) and DDIM (Song et al., 2020). We do **not** adopt the scaled ones since the number of inference steps is not directly related to prompt alignment;

- Z-Sampling (Bai et al., 2024), which simultaneously enhances sample aesthetics and semantic alignment. We adopt deterministic DDIM sampling (since stochastic samplers

underperform their deterministic counterparts owing to larger approximation errors) with hyper-parameters recommended in the official implementation [16] of Z-Sampling. Specifically, we apply the zig-zag operation along the entire trajectory, executing it $k$ time(s) per denoising step for scaling ($k = 1$ in the official release of Z-Sampling), and retain the samples with the highest CLIP Score across all $k$s. The *back-tracking stepsize* is set to 1 for each zig-zag operation. The guidance scale for denoising and inversion are set to 5.5 and 0.0, respectively. Note that, both denoising and inversion steps are counted toward the *NFE dynamics*. To respect the NFE budget of 999, the constraint $k \cdot 3(n - 1) + 1 \leq 999$ yields $k \leq 11$.

### K.2 RESULTS AND ANALYSIS

**Qualitative Results.** Note that, the displayed samples of Z-Sampling in Fig. 1 (d) in the main paper and below are produced with $k = 2$ (highest CLIP Score). We provide more qualitative results in Fig. 8. The text prompts are "`A separate seat for one person, typically with a back and four legs.`", "`A mechanical or electrical device for measuring time. `", "`An umbrella on top of a spoon.`", "`A tennis racket underneath a traffic light.`", "`A donut underneath a toilet.`", and "`A train on top of a surfboard.`", respectively.

It can be observed that,

1. For line 1 to line 4, our method produces high-quality images that align well with the prompts, whereas baselines render the semantics to some extent with lower quality;

2. For line 5 to line 6, our samples accurately depict the positional relationship between the objects, which the baselines fail to capture.

**Discussion on Quantitative Results.** Tab. 6 in Sec. 5.5 of the main paper presents quantitative results, where the relative improvement is computed based on 30-step DDIM. It demonstrates that,

1. Our method achieves the most significant performance enhancement;

2. Z-Sampling yields larger gains in aesthetic scores. However, aesthetic scores and CLIP score are only weakly — and sometimes even negatively — correlated, *i.e.*, samples judged more aesthetically pleasing may receive lower CLIP scores, which is called *inconsistent preferences* in Zhang et al. (2025). Consequently, such undirected and inconsistent optimization fails to maximize the CLIP score.

**Scaling Z-Sampling.** As shown in Tab. 17, scaling Z-Sampling does not guarantee improved performance, and can induce quality degradation, *e.g.*, color saturation and noise. In contrast, our method exhibits robust scaling behavior. Qualitative results are shown in Fig. 9), with text prompts "`A bicycle on top of a boat.`" and "`A red colored car.`", respectively. We attribute this to:

Table 17: **Quantitative results of scaling Z-Sampling.** "*max*": retain the samples with the highest CLIP Score across all $k$s.

| $k$ | 1
7 | 2
8 | 3
9 | 4
10 | 5
11 | 6
max |
|---|---|---|---|---|---|---|
| CLIP Score (↑) | 0.3443
0.3449 | 0.3508
0.3418 | 0.3500
0.3423 | 0.3484
0.3441 | 0.3464
0.3467 | 0.3459
**0.3676** |
| HPS v2 (↑) | 29.0346
28.4049 | 29.2260
28.1979 | 29.0893
28.0820 | 28.8760
28.0305 | 28.7349
27.8464 | 28.6070
**29.5635** |

1. Information corruption introduced by repeated noising-denoising cycles (Lugmayr et al., 2022; Wang et al., 2022; Bansal et al., 2023; Karras et al., 2022);

2. Z-Sampling progressively refines the attention masks of the foreground subjects (Bai et al., 2024). An excessive number of zig-zag operations tend to blur the background or yield an overly monotonous backdrop.

---

[16] https://github.com/xie-lab-ml/Zigzag-Diffusion-Sampling/

Figure 8: **Qualitative results of optimizing CLIP score.**

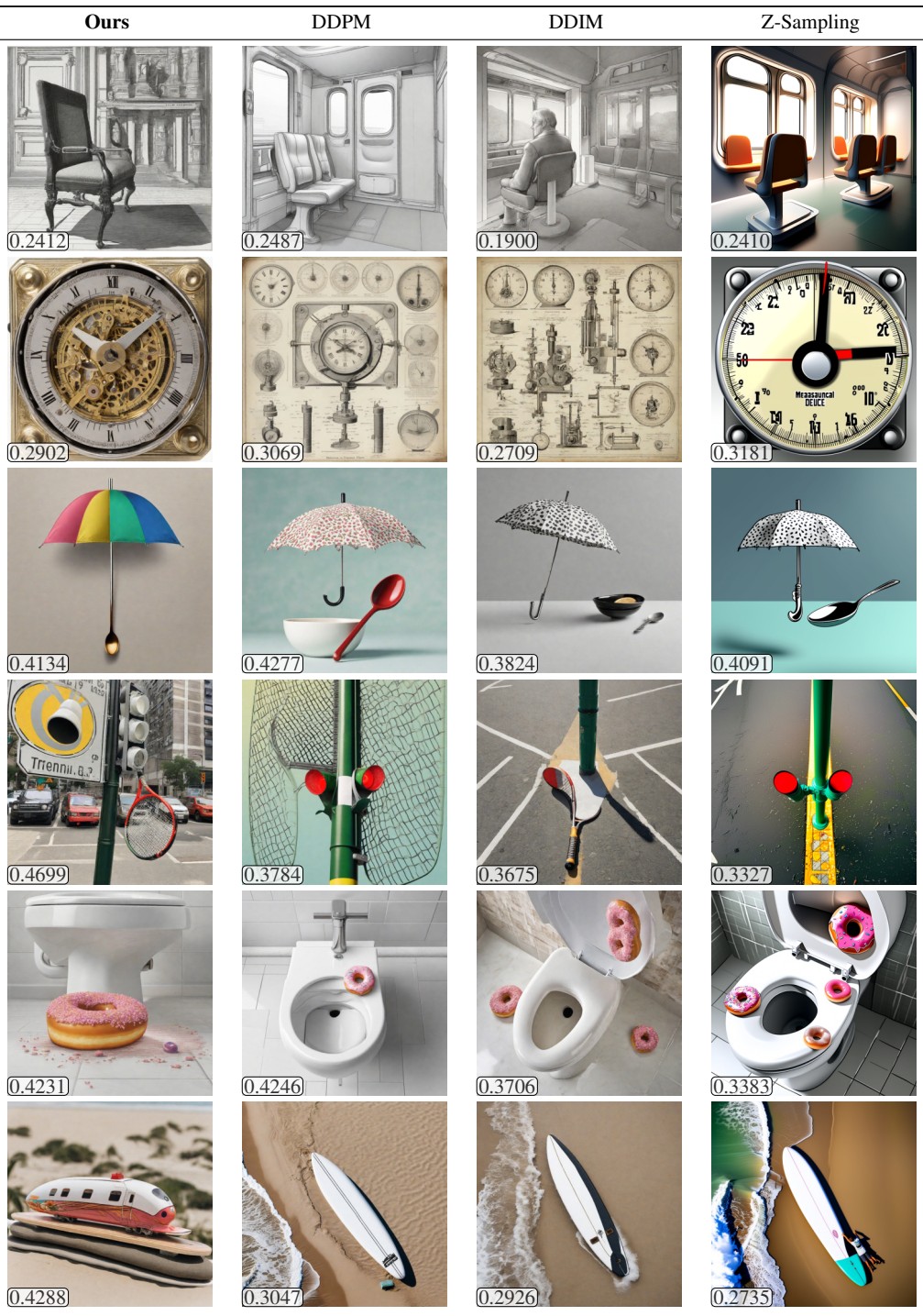

## L    EXPERIMENT: SUPPLEMENTARY TO ALIGNING WITH COMPOSITE REWARDS

### L.1    SETTINGS

**The Dataset and Model.**    We randomly sample 20 prompts from the HPD v2 dataset (Wu et al., 2023a), and align 50-step Pixart-$\alpha$ (Chen et al., 2023b) to different reward functions.

Figure 9: **Qualitative results of scaling Z-sampling (various $k$s).**

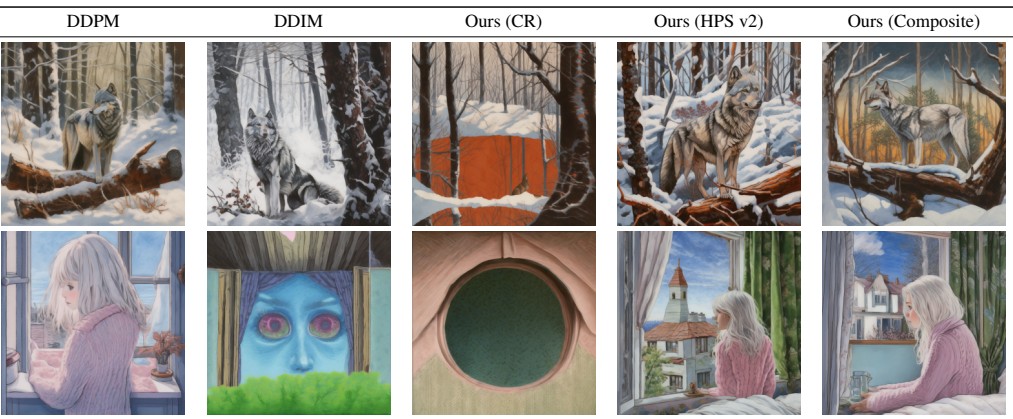

**Composite Reward.** The *composite reward* that combines CR and HPS v2 (Wu et al., 2023a) is defined as

$$R_{\text{composite}} = R_{\text{compressibility}} + \lambda \cdot R_{\text{HPS\_v2}}. \tag{79}$$

The following discusses the selection of hyper-parameter $\lambda$. After normalizing both compressibility reward and HPS v2 to $[0, 1]$, the former remains tightly clustered above $0.95$, whereas the latter spans a much wider range. We tune $\lambda$ to balance these two objectives, prioritizing samples with higher HPS v2 while still optimizing compressibility. The value of $\lambda = 0.02$ is chosen to approximate the ratio of the standard deviations of these two rewards across several random samples.

**NFE Budgets.** The limit of NFE dynamics is set to 999.

**Methods.** We conduct comparison between *Ours (CR)*, *Ours (HPS v2)*, and *Ours (Composite)*, where *Ours (R)* denotes aligning with reward $R$ using our method.

### L.2 RESULTS AND ANALYSIS

Figure 10: **Qualitative results of aligning PixArt-$\alpha$ with different objectives.**

| DDPM | DDIM | Ours (CR) | Ours (HPS v2) | Ours (Composite) |
|---|---|---|---|---|

Qualitative results in Fig. 1 (b) in the main paper are generated with the text prompt "A beautiful blue and pink sky overlooking the beach.". We provide more qualitative results in Fig. 10. The text prompts are "A black wolf standing on a fallen tree in a winter forest." and "A white-haired girl in a pink sweater looks out a window in her bedroom.", respectively. It demonstrates that,

1. When optimizing compressibility reward alone, a form of reward hacking emerges: the model collapses to introducing unreasonable smooth abnormalities to inflate the reward (*e.g.*, Fig. 1 (b)), producing only smooth backgrounds and omits the foreground subject (*e.g.*, Fig. 10 line 1), or even exhibit complete semantic loss (*e.g.*, Fig. 10 line 2);

2. Optimizing the composite reward yields samples that are perceptually smoother yet retain clear semantics, validating our method's ability to align DMs with various rewards simultaneously.

## M  EXPERIMENT: SUPPLEMENTARY TO GENERALIZATION TO OTHER LATENT REWARD POLICIES

### M.1  SETTINGS

**The Dataset, Model and Reward Function.**
We randomly sample 20 prompts from the HPD v2 dataset (Wu et al., 2023a). We align 15-step SD v1.4 (Rombach et al., 2022) with HPS v2 (Wu et al., 2023a).

Table 18: **Performance of different latent reward policies when optimizing HPS v2.**

| Reward Policy | Best Reward | NFE Dynamics | NFE Inter. | NFE Final | Time Cost |
|---|---|---|---|---|---|
| *immediate-ddim* | 29.2582 | 316.52 | 279.38 | 595.90 | 686 |
| *immediate-score* | 29.0293 | 325.88 | 289.27 | 615.15 | 1458 |
| *LA-2* | 29.3957 | 376.90 | 334.95 | 711.85 | 1693 |
| *LA-3* | 29.3938 | 326.00 | 290.82 | 616.83 | 1528 |
| *sequential* | 28.7070 | 292.07 | 35.90 | 74.55 | 314 |

**Latent Reward Policies.**  We adopt different latent reward policies in Appx. D.2 respectively for evaluation. Particularly, the actions used for computing intermediate rewards in *sequential* policy are uniformly sampled from $[0, 1]$.

**NFE Budgets.**  The limit of NFE dynamics is set to 500.

### M.2  RESULTS AND ANALYSIS

Quantitative results in Tab. 18 demonstrate that,

1. Different latent reward policies vary in the accuracy of intermediate rewards, the quality of pseudo-final latents, and computational cost;

2. The *LA-2* and *LA-3* policies yield the 1st/2nd best performance, respectively. They take two or three steps to compute an intermediate reward, which increases the probability of obtaining better pseudo-final samples, leading to performance superior to that of the two *immediate* policies;

3. The *sequential* policy, which spends more steps calculating intermediate rewards, achieves the worst performance, as it wastes a significant amount of NFEs to calculate intermediate rewards, which should be used for exploration in other latent reward policies. This underscores the importance of efficient intermediate reward computation for effective search.

## N  EXPERIMENT: SUPPLEMENTARY TO ABLATIONS

### N.1  SUPPLEMENTARY TO REWARD HACKING AND EFFECTS OF $\tau$

**Settings.**  We randomly sample 20 prompts from the HPD v2 dataset (Wu et al., 2023a), and align 50-step SD v1.4 (Rombach et al., 2022) to the Laplacian variance (LAPV) (Memon et al., 2015; GeeksforGeeks, 2024) and 3 types of color channel reward (CCR) (Eyring et al., 2024) (*CCR-R/G/B* for the R/G/B channel), respectively. The limit of NFE dynamics is set to 500.

**Supplementary to Reward Hacking.**  Sec. 5.7 in the main paper illustrates a reward hacking phenomenon (Eisenstein et al., 2023) that, the optimization process identifies vulnerabilities in LAPV and CCRs, where noisy and blurred samples can yield higher rewards (Black et al., 2023). Exploiting this, the process leverages early-stage pseudo-final samples from our max-reward modeling to maximize rewards with unreasonable results. Luckily, this form of reward hacking can be mitigated by excluding pseudo-final samples from valid sample consideration. But this might compromise the core features of the max-reward modeling, thus potentially degrading the performance.

**Results and Analysis.**  The displayed samples in Fig. 1 (e) in the main paper are generated with text prompts "A tall chicken standing next to a farmer.", "A close-up hyperrealistic oil painting of a nurse fashion model with red lipstick, ginger hair, freckles, in a style mixing classicism and 80s sci-fi, set in complete darkness.", "A symmetrical oil painting of two waterfalls in a dense forest." and "Pippi is tethered to the international space station in her space suit amidst stars and galaxies.", respectively. We present the following discussion:

1. Even though LAPV is an unbounded reward whose values span a wide range, our method achieves robust inference-time DM alignment;

2. Optimizing CCR is really challenging for our approach. Specifically, we fix the parameters of the pre-trained SD v1.4 and treat it as the environment model. Since RODE sampling can only induce limited perturbations along certain directions (determined by the process noise), our method can **not** achieve an overall color-tone effect like that in Eyring et al. (2024), as this would deviate from the original data distribution embeded in SD v1.4. However, qualitative results in Fig. 1 (e) show that, our method can sometimes obtain higher rewards in tricky ways. For example, the search process resorts to adding small components with colors closely relative to the target channel (*i.e.*, the primary color R/G/B and its secondary colors (Hunt, 2005)) for higher rewards, while maintaining reasonable coherence. This highlights the adaptability of our method in extreme cases.

## N.2 ABLATIONS ON $m$ AND $\zeta$

**Effects of $m$.** We study the role of the hyper-parameter $m$ in Aug. RODE sampling by limiting the number of groups of process noise. Specifically, when the full budget comprises $m$ sets of process noise, an ablation that uses only $m' \in [1, m]$ sets is conducted by enumerating all $C_m^{m'}$ possible combinations, and reporting the best result across them. We randomly sample 20 prompts from the HPD v2 dataset (Wu et al., 2023a), and align 15-step SD v1.4 (Rombach et al., 2022) to HPS v2 (Wu et al., 2023a). Results in Tab. 19 shows that, too few groups confine exploration to limited perturbation directions, causing performance degradation. It underscores the need to augment RODE sampling.

Table 19: **Ablations on $m$ for augmented RODE sampling.**

| $m$ | 1 | 2 | 3 (Ours) |
|---|---|---|---|
| **Best Reward** | 28.7344 | 29.1270 | 29.2582 |

Table 20: **Ablations on $\zeta$ for Beta policies.** Performance worse than uniform policies (29.1152 in Tab. 21) is marked with ↓.

| $\zeta$ | 1 9 | 2 10 | 3 11 | 4 12 | 5 13 | 6 14 | 7 15 | 8 |
|---|---|---|---|---|---|---|---|---|
| **Best Reward** | 29.1102 29.0988 ↓ | 29.1254 29.0199 ↓ | 29.2582 29.0371 ↓ | 29.1410 29.0625 ↓ | 29.1148 ↓ 29.1273 | 29.0547 ↓ 29.0859 ↓ | 29.0973 ↓ 29.1227 | 29.0961 ↓ |

**Effects of $\zeta$.** We also examined the concentration control scalar $\zeta$ in Beta policies. Specifically, we randomly sample 20 prompts from the HPD v2 dataset (Wu et al., 2023a), and align 15-step SD v1.4 (Rombach et al., 2022) to HPS v2 (Wu et al., 2023a). Tab. 20 presents the quantitative results, whose plot is shown in Fig. 11. It indicates that, $\zeta = 3$ achieves the best performance, which we attribute to an appropriate balance between exploration and exploitation. Moreover, the performance of Beta policy with an inappropriate selection of $\zeta$ may fall behind that of the naive uniform policy, especially when $\zeta$ is large. We attribute this to the Beta policy becoming trapped in local optima due to over-exploitation, whereas the uniform policy enables more thorough exploration with sufficient NFEs.

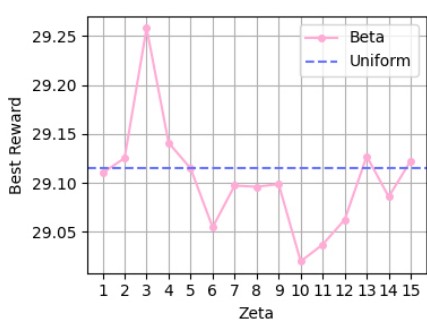

Figure 11: **Ablations on $\zeta$.**

## N.3 SUPPLEMENTARY TO MAIN ABLATIONS

**Settings.** We conduct ablations on key components of our method. Specifically, we randomly sample 20 prompts from the HPD v2 dataset (Wu et al., 2023a), and align 15-step SD v1.4 (Rombach et al., 2022) to HPS v2 (Wu et al., 2023a).

**Results and Analysis.** Quantitative results in Tab. 21 demonstrate that:

Table 21: **Ablations.** "*Ours w/o pseudo-final*": disabling pseudo-final samples as valid samples. "*Ours w/o Beta policy*": uniform policy.

| Method | Best Reward | NFE Dynamics | NFE Inter. | NFE Final | Time Cost |
|---|---|---|---|---|---|
| **Ours** | 29.2582 | 316.52 | 279.38 | 595.90 | 686 |
| Ours w/o pseudo-final | 29.0199 | 293.73 | 259.82 | 553.55 | 612 |
| Ours w/o expansion depth limit | 29.1082 | 298.45 | 259.18 | 557.62 | 631 |
| Ours w/o online update | 29.1547 | 325.88 | 287.18 | 613.05 | 1028 |
| Ours w/o Beta policy | 29.1152 | 297.02 | 264.75 | 561.77 | 528 |

1. Excluding pseudo-final samples from valid samples leads to the worst performance, since the samples encountered during the search are not fully leveraged. This underscores the advantage of our max-encountered-reward formulation over the conventional max-final-reward formulation;

2. In the absence of an expansion-depth cap, a disproportionate amount of NFEs is assigned to the nodes whose depth is close to $n$ during the later search process. This is because, deeper nodes are more likely to yield higher rewards when using *immediate-ddim* policy. However, modifying $\eta_{t_i}$ at late stages leads to limited alterations to latents, leaving the shallower and potentially more promising nodes under-explored;

3. Disabling the online update of Beta policy freezes the peak of the Beta distribution at its initial value, preventing the search focus from incorporating best-so-far knowledge;

4. The uniform policy underperforms Beta policy, highlighting the appropriate balance between exploitation and exploration;

5. Any component is indispensable.

## O APPLICATIONS

### O.1 SYNERGY WITH COMMUNITY MODULES

Our method lies in trajectory search techniques, which is plug-and-play compatible with community modules and other inference-time scaling approaches, further boosting performance. For example, Promptist (Hao et al., 2023) can be introduced to optimize the input prompts into model-preferred ones, and the initial noises for each prompt can be respectively optimized with Golden Noise (GN) (Zhou et al., 2024b) before the MCTS process.

**Settings.** We randomly sample 20 prompts from the HPD v2 (Wu et al., 2023a) dataset, and align the 30-step SDXL base (Podell et al., 2023) to HPS v2 (Wu et al., 2023a). We sequentially integrate our method with Promptist (Hao et al., 2023) and GN (Zhou et al., 2024b). The limit of NFE dynamics is set to 500 to highlight how optimizing the prompt and the initial noise contributes to the early search of our method.

Table 22: **Quantitative results of synergy with community modules.**

| Method | HPS v2 (↑) | CLIP Score (↑) |
|---|---|---|
| Ours | 28.2766 | 0.3564 |
| + Promptist | 29.1418 | 0.4177 |
| + GN | 29.1945 | 0.4181 |

**Results and Analysis.** Quantitative and qualitative results are presented in Tab. 22 and Fig. 12. The samples in Fig. 12 in the main paper is generated with text prompts:

1. Line 1: "A man smiles as he stirs his food in the pot.", and its optimized version "a man smiling as he stirs his food in the pot by greg rutkowski, digital art, realistic painting, fantasy, very detailed, trending on artstation";

2. Line 2: "A painting of a Persian cat dressed as a Renaissance king, standing on a skyscraper overlooking a city.", and its optimized version "painting of a Persian cat dressed as a

Figure 12: **Qualitative results of synergy with community modules.**

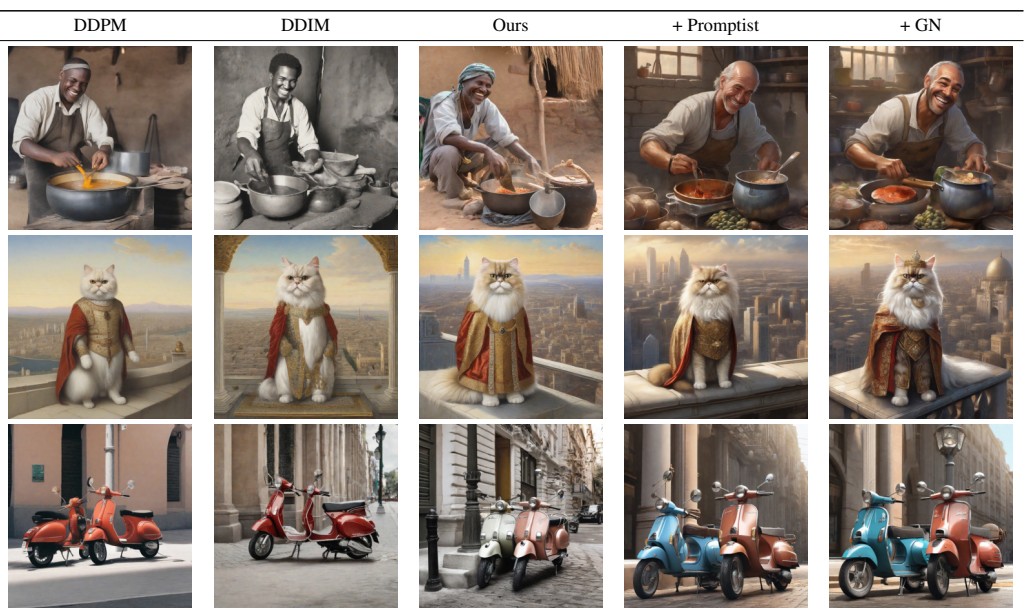

| DDPM | DDIM | Ours | + Promptist | + GN |

> Renaissance king, standing on a skyscraper overlooking a
> city, by Greg Rutkowski and Raymond Swanland, Trending on
> Artstation, ultra realistic digital art";

3. Line 3: " Two vespas parked next to a light post." and its optimized
   version "two vespas parked next to a light post, hyperdetailed,
   artstation, cgsociety, 8 k".

Note that the reward used during search process is computed with the original prompts, whereas the metrics reported in Tab. 22 are evaluated with the optimized prompts. Results demonstrate that,

1. Robust improvements persist even after prompt or initial-noise refinement;

2. Our approach is orthogonal to both Promptist and GN, and these three techniques can synergistically elevate the performance of inference-time DM alignment, achieving samples with higher aesthetics.

O.2    ROBUSTNESS OF IMAGE REWARD FUNCTIONS

Robustness evaluation is often overlooked in existing studies on reward models for text-to-image generation (Liu et al., 2024), particularly in terms of quantitative assessment. Our RODE-based scaling method introduces limited and controllable perturbations to the denoising trajectory, resulting in a sequence of progressively similar intermediate images. This enables us to assess the robustness of the optimized (Lipschitz continuous) reward models.

**Quantifying Robustness.**    We approximate the absolute value of the derivative of the reward change with respect to the perceptual change (measured by LPIPS (Zhang et al., 2018)) to quantify robustness. Formally, let $z_{t_n}$ and $z'_{t_n}$ be two consecutive intermediate samples during the optimization of a reward model $\phi$. The *local robustness* of $\phi$ on this sample pair is measured as

$$\text{Robustness}(\phi; z_{t_n}, z'_{t_n}) = \left| \frac{\phi(z_{t_n}) - \phi(z'_{t_n})}{\text{LPIPS}(\mathcal{D}(z_{t_n}), \mathcal{D}(z'_{t_n})) + \varepsilon} \right|, \tag{80}$$

where $\mathcal{D}(\cdot)$ is the decoder of the VAE (Kingma et al., 2013; 2021) for latent DMs (Sauer et al., 2024a), and $\varepsilon > 0$ is a small constant for numerical stability. A higher robustness value indicates a less robust $\phi$. The *robustness* is measured as the average of its local robustness for all initial noises and prompts.

Figure 13: **Failure cases of optimizing HPS v2.**

**Settings.** We randomly sample 20 prompts from the HPD v2 dataset (Wu et al., 2023a), and set $\varepsilon = 10^{-6}$. We adopt HPS v2 (Wu et al., 2023a), PS (Kirstain et al., 2023) and IR (Xu et al., 2023a) as reward models, respectively. The limit of NFE dynamics is set to 500.

Table 23: **Robustness ($\downarrow$) of three popular aesthetics models.**

| Reward Model | HPSv2 | PS | IR |
|---|---|---|---|
| **Robustness** | 1.5926 | 1.0674 | 0.6491 |

**Results.** Tab. 23 reports the robustness of three popular aesthetic reward models, where IR exhibits the highest robustness.

### O.3 QUANTITATIVE EVALUATION OF INITIAL NOISE POTENTIAL

Our method can be seen as a technique that sufficiently utilizes the potential of initial noise *i.e.*, it reveals the best achievable performance of an initial noise under given conditions and computational constraints. This can be used to quantitatively evaluate the *quality* or *generation potential* of initial noise, *e.g.*, serves as a verifier. We leave this for future work.

## P FAILURE CASES

### P.1 ALIGNING WITH AESTHETICS

We provide failure cases when optimizing HPS v2 (Wu et al., 2023a) (Sec. 5.4, Appx. J) in Fig. 13. The text prompts are "A portrait of a woman with a paper bag over her head." and "A woman that is standing near an open oven.", respectively. It can be observed that,

1. For line 1, both ours and GS-eps introduce anomalies in limbs and fingers;
2. For line 2, all three $\eta_{t_i}$-action methods underperform the baselines in score, and fails to render the semantics of "a woman" correctly.

### P.2 ALIGNING WITH SEMANTICS

We provide failure cases when optimizing CLIP score (Radford et al., 2021; Hessel et al., 2021) (Sec. 5.5, Appx. K) in Fig. 14. The text prompts are "A hair drier underneath a sheep." and "A zebra underneath a broccoli.", respectively. It can be observed that, although our method attains higher CLIP scores, it fails to render accurate positional relationship (line 1) and reasonable composition (line 2), while Z-sampling achieves high prompt alignment.

## Q BROADER IMPACTS, LIMITATIONS AND FUTURE WORK

### Q.1 BROADER IMPACTS

We outline the following potential broader impacts:

1. Our augmented RODE sampling provides an alternative solution for introducing randomness into ODE sampling without converting it into SDE sampling. This paradigm can po-

Figure 14: **Failure cases of optimizing CLIP score.**

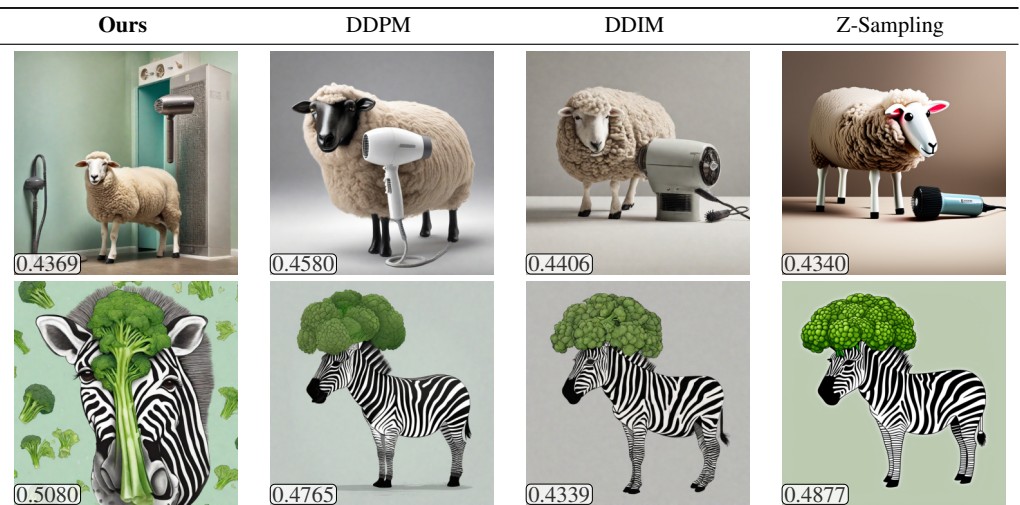

| **Ours** | DDPM | DDIM | Z-Sampling |
|---|---|---|---|

tentially be adopted for RL with flow matching (Albergo & Vanden-Eijnden, 2022; Lipman et al., 2022; Liu et al., 2022) models;

2. This study enhances the capabilities of pre-trained models in a training-free manner, such as enabling few-step inference performance to surpass that of multi-step inference in the same model. This suggests that pre-trained image generation models may not be fully utilized, prompting the community to explore the deeper potential of pre-trained models rather than solely focusing on training larger models.

### Q.2 LIMITATIONS

We highlight the following limitations:

1. The computational overhead of our method precludes its deployment in real-time scenarios;

2. Our method attempt to fully utilized the pre-trained DMs, whose performance is bounded by the capabilities of the adopted base models;

3. Our method is more effective in multi-step scenario, because the performance of our RODE sampling is highly dependent on a sufficient number of usable process noise. Despite our augmentation strategy partially alleviates this problem, we empirically observe dramatic performance degradation in very few-step inference scenarios;

4. The *NFE dynamics* required for the simulation phase of MCTS increases linearly with the number of inference steps, leading to higher computational demands in scenarios with larger step counts;

5. When using dense rewards from latent reward shaping, the extensive queries of intermediate rewards necessitate a large number of final reward calculation calls. This can become a notable efficiency bottleneck when reward function queries are costly;

6. In the context of training-free DM alignment, our approach often lags behind gradient-based methods. This is because the later — which optimize the latent variables or noise prediction with gradient — can modify the latent representations to a greater extent and with higher accuracy, thereby holding greater promise for achieving superior performance.

### Q.3 FUTURE WORK

We list the following potential directions for future work:

1. Generalize to stochastic samplers other than DDIM (Song et al., 2020);

2. Leverage advanced heuristic techniques to accelerate the search process for the optimal control sequence, thereby improving efficiency. For example, a problem-specific admissible heuristic that predicts the maximum attainable reward along a path can be introduced to prune unpromising branches at an early stage;

3. Propose a global-local search strategy. Specifically, first conduct a global search that treating $\epsilon_{t_i}$s as actions to identify high-reward regions. Then, perform a local search with $\eta_{t_i}$ actions to fine-tune the denoising trajectory to release the potential of an initial noise to the maximum extent;

4. Explore other parameter-efficient scaling methods;

5. While we have used low-dimensional action sequences to easily visualize the high-dimensional denoising trajectories of DMs, the relationship between sampling results and action sequences requires further investigation.

