# OpenReview forum: "Inference-Time Diffusion Model Alignment via Random Ordinary Equations"
_ICLR.cc/2026/Conference — Submitted to ICLR 2026_

### Official Review · Reviewer_cCHK · 2025-10-25

**Soundness:** 2
**Presentation:** 3
**Contribution:** 3
**Rating:** 4
**Confidence:** 2

**Summary:**

- This paper proposes a novel diffusion alignment method that operates efficiently at inference time.
- The approach is based on Monte Carlo Tree Search (MCTS), with the key originality lying in the combination of RODE sampling, a one-dimensional action space, and a max-reward control strategy.
- The proposed method is validated through extensive experiments, demonstrating that
  (a) it efficiently and actively searches for high-reward trajectories, and
  (b) its sampling process is both stable and diversity-controllable owing to the RODE formulation.
- The method is applicable to non-differentiable rewards, since it operates in a gradient-free manner.
- The authors also provide theoretical guarantees regarding sampling stability and score estimation error bounds.

Note: I used ChatGPT for minor language editing and phrasing assistance; all technical assessments are my own.

**Strengths:**

- This paper presents a novel diffusion-alignment method that operates efficiently at inference time.
- The proposed method is supported by extensive experiments, evaluated in terms of "Best Reward" and "Parameter Efficiency".
- The experiments also provide useful insights into the behavior of the proposed alignment method:
  (a) Proposition 1 and the experimental result on sample diversity (measured by the mean pairwise distance) are well connected, as discussed around line 405.
  (b) The visualization of the denoising path on the data manifold (Figure 4) effectively illustrates the exploration capability of the proposed approach.
- The authors also provide theoretical guarantees regarding the stability of the method and the bound on score-estimation error.

Note: I used ChatGPT for minor language editing and phrasing assistance; all technical assessments are my own.

**Weaknesses:**

- The theoretical argument, especially Proposition 2, needs to be clarified.
- It is not entirely clear whether the authors can rigorously justify the following conclusions based on their propositions:
  (1) that $p^{(R)}$ approximates the true data distribution as well as $p^{(S)}$; and
  (2) that using an SDE-trained score network within RODE sampling does not significantly increase the score-estimation error.
- See Question (2) below for more detailed comments.

Note: I used ChatGPT for minor language editing and phrasing assistance; all technical assessments are my own.

**Questions:**

1. **On the max-reward control strategy**
   Can the authors provide a more detailed explanation of why the proposed max-reward control strategy performs better compared to FK steering [1] (which is also cited in the paper)?
   This kind of approach seems natural since the intermediate reward is explicitly defined (see the theoretical discussions in, e.g., [2]).

2. **On Proposition 2 and its interpretation**
   (a) Was $M_{t_{k+1}}$ introduced before Proposition 2?
   It might be helpful if the authors explicitly describe all quantities appearing on the right-hand side.
   Also, is $M_{t_{k+1}}$ exponential in the number of steps?
   (b) It seems that the right-hand side does not vanish even if $\Delta t \to 0 $.
   Can the authors still claim that $p^{(S)}$ and $p^{(R)}$ are sufficiently close under some limiting regime?

3. **On Assumption 8**
   Is Assumption 8—stating that the mapping $\Psi_t$ is twice continuously differentiable—reasonable or realistic in practice?
   One of the main strengths of your work is that the proposed MCTS framework can handle **non-differentiable rewards**.
   Therefore, it would be desirable if the underlying diffusion dynamics (and the mapping $\Psi_t$ ) could also relax the differentiability requirement, so that both the diffusion model and the reward function may be non-differentiable.


---

**References**

[1] Raghav Singhal, Zachary Horvitz, Ryan Teehan, Mengye Ren, Zhou Yu, Kathleen McKeown, and Rajesh Ranganath. *A general framework for inference-time scaling and steering of diffusion models.* arXiv preprint arXiv:2501.06848, 2025.

[2] Uehara, M., Zhao, Y., Black, K., Hajiramezanali, E., Scalia, G., Diamant, N. L., ... & Levine, S. (2024). *Fine-tuning of continuous-time diffusion models as entropy-regularized control.* arXiv preprint arXiv:2402.15194.

Note: I used ChatGPT for minor language editing and phrasing assistance; all technical assessments are my own.

---

> ### Author Response · Authors · 2025-11-21
> **Reply to Reviewer cCHK - Round 1 - (1/3)**
>
> Reply to Reviewer cCHK - Round 1 - (1/3)
>
> Thank you for the thoughtful consideration of the paper and constructive feedback.
>
> **1. Advantage of max-encountered-reward modeling**:
>
> (1) FK steering [1] is a particle-potential-based *search strategy*, whereas max-encountered-reward control is a *modeling strategy*. They are not directly comparable since they describe different dimensions of the problem.
>
> (2) We acknowledge that FK steering [1] and related methods adopt a natural idea — using intermediate rewards as potentials to guide the search direction. However, as discussed in Sec. 3.1, when operating under max-final-reward modeling, the estimated intermediate rewards often suffer from inconsistency and inadmissibility, therefore potentially misleading the search process.
>
> To address this issue, we observe that almost all intermediate reward estimation techniques involve deriving some form of pseudo-final samples. Instead of treating them merely as intermediate constructs, our method directly regards them as valid samples for reward computation. This leads to the formulation of our max-encountered-reward modeling.
>
> (3) The intermediate rewards in both FK steering [1] and our approach are explicitly defined. Nevertheless, our contribution goes further in the design of intermediate reward construction. Specifically, we introduce a unified latent-reward-shaping formulation (Sec. 2.3, Appx. D.2), in which the mapping function $\mathcal{F}$ is modular and easily replaceable. This design enables users to flexibly choose an implementation of $\mathcal{F}$ that best fits their computational constraints or task-specific requirements.
>
> [1] Raghav Singhal, Zachary Horvitz, Ryan Teehan, Mengye Ren, Zhou Yu, Kathleen McKeown, and Rajesh Ranganath. *A general framework for inference-time scaling and steering of diffusion models.* arXiv preprint arXiv:2501.06848, 2025.
>
> Note: We used ChatGPT for language polishing. All technical content was written by ourselves.

---

> ### Author Response · Authors · 2025-11-21
> **Reply to Reviewer cCHK - Round 1 - (2/3)**
>
> Reply to Reviewer cCHK - Round 1 - (2/3)
>
> **2. Clarification of Proposition 2, 3**:
>
> (1) We apologize for the misunderstanding caused by our writing. Our conclusion is based on bounded-error analysis, not on proving any form of equivalence. Specifically, Proposition 2 is *not* intended to show that $p^{(R)} \approx p^{(S)} \approx p_{data}$, but rather to demonstrate that the discrepancy between $p^{(R)}$ and $p^{(S)}$ is Lipschitz-bounded. This ensures that RODE sampling can approximate the data distribution to a comparable extent.
>
> (2) Proposition 3 further shows that, when an SDE-trained score network is used within RODE sampling, the point-wise score-estimation error between RODE and SDE sampling remains bounded. In other words, reusing a pretrained score network does not introduce a significant increase in score-estimation error. This stems from the intrinsic robustness of score-network training to variations in noise (see Appx. F.3.2). That is, the score network approximates marginal distributions and is independent of sampling trajectories, and RODE sampling only alters the trajectories but *not* the input distribution, therefore it does not introduce an additional error order.
>
> (3) In the earlier version of the manuscript, notation explanations were omitted in the main paper due to space constraints and were presented in Appx. F.1 and F.3. In the revised version, we have added explicit notation descriptions near Proposition 2 in the main paper.
>
> (4) In Proposition 2, $M_{t_{k + 1}} = \prod_{j = k + 1}^{n - 1} L_{t_j} (0 \leq j \leq n - 1)$, $M_{t_n} = 1$, and it may grow exponentially if each stepwise Lipschitz constant exceeds 1. However, since the total time horizon $T = \sum \Delta t_k$ is finite, if all stepwise constants are uniformly upper-bounded by $L < \infty$, then one can show that $M_{t_{k + 1}} \leq \mathrm{e}^{LT}$.
>
> We appreciate your suggestion, which has helped us further polish this theoretical statement.
>
> (5) It is correct that the right-hand side in Proposition 2 may not vanish as $\Delta t \rightarrow 0$; instead, it typically converges to a finite, nonzero constant. Strictly guaranteeing $W_1(p^{(R)}, p^{(S)}) \rightarrow 0$ requires much stronger assumptions. Nevertheless, in practice, RODE sampling can still be regarded as a valid approximation because:
>
> (I) Higher-order residual terms diminish as $\Delta t \rightarrow 0$.
>
> (II) A well-trained score network has a small intrinsic estimation error.
>
> (III) Proposition 3 shows that using an SDE-trained score network in RODE sampling does not significantly increase this error;
>
> (IV) Although a theoretical gap exists, empirical results (Tab. 1 and Fig. 1) show that the generated samples are indistinguishable in practice, indicating that RODE and SDE sampling yield visually comparable distributions.
>
> Together, these factors justify the practical validity and stability of using RODE sampling for exploration.
>
> Note: We used ChatGPT for language polishing. All technical content was written by ourselves.

---

> ### Author Response · Authors · 2025-11-21
> **Reply to Reviewer cCHK - Round 1 - (3/3)**
>
> Reply to Reviewer cCHK - Round 1 - (3/3)
>
> **3. Clarification of Assumption 8**:
>
> (1) The mapping $\Psi_{t_i}$ arises from a local linearization of the diffusion dynamics (see Assumption 4 in Appx. F.1). Even when the underlying diffusion dynamics are not differentiable, it is still possible to locally approximate them with a differentiable linear mapping $\Psi_{t_i}$ for the purpose of theoretical analysis. The assumption that $\Psi_{t_i}$ is twice continuously differentiable ensures that the remainder term in the Taylor expansion is bounded. This assumption is standard and widely accepted in theoretical analysis in machine learning, including in classical diffusion model theory such as consistency models [1].
>
> (2) Our method does not involve sample gradients nor reward function gradients. Therefore, it does not require the diffusion model itself to be differentiable. In other words, our ability to handle non-differentiable rewards does not come from designing a gradient-estimation mechanism for reward functions (which may require model gradients), but rather from the fact that our method does not explicitly use gradients at all.
>
> (3) In practice, diffusion dynamics are governed by neural networks and are therefore intrinsically differentiable, though gradient computation is often expensive. As a result, most prior work on diffusion model alignment focuses primarily on the differentiability of reward functions rather than that of the diffusion model.
>
> [1] Song Y, Dhariwal P, Chen M, et al. Consistency models[J]. 2023.
>
> Note: We used ChatGPT for language polishment. All technical content was written by ourselves.

---

> > ### Comment · Reviewer_cCHK · 2025-11-26
> >
> > Thank you for your detailed response. My concerns about the motivation and Assumption 8 have been resolved.
> >
> > However, I am still uncertain about the interpretation of **Propositions 2 and 3**. In particular:
> >
> > - Under standard assumptions, the **score‐estimation error with SDE sampling** can be $o_N(1)$ as the sample size $N\to\infty$.
> >   By contrast, with ODE (or RODE) sampling, the corresponding error may remain **$O_N(1)$** (as I read).
> > - Given this, I do not find it sufficiently strong—mathematically—that **the pointwise score‐estimation error between RODE and SDE remains merely bounded**. Since the target procedure (SDE) has vanishing estimation error, whereas the RODE error may not vanish, it is not yet clear to me whether the boundedness result is nontrivial, or whether it should be interpreted only as a qualitative stability statement.
> >
> > It would help if the authors could clarify the intended scope of these propositions.

---

> > > ### Author Response · Authors · 2025-12-02
> > > **Reply to Reviewer cCHK - Round 2 - (1/1)**
> > >
> > > Reply to Reviewer cCHK - Round 2 - (1/1)
> > >
> > > **1. The strength of the proposition**:
> > >
> > > Thank you for the insightful comment — you are correct.
> > >
> > > Under the assumptions used in the earlier version, the right-hand side of Proposition 2 does not vanish, which in turn makes Proposition 3 establish only a trivial stability statement (i.e., the error does not blow up).
> > > To make the conclusion of Proposition 3 nontrivial, we have added Appx. F.3.3 and explicitly referenced it in the main paper following the statement of Prop. 3.
> > >
> > > In this new section, we first discuss the trivial case identified in your comment, and then introduce additional assumptions under which we can derive a stronger proposition. This strengthened result ensures that the right-hand side of Proposition 2 does vanish, and consequently, the right-hand side of Proposition 3 also vanishes. Under this regime, the pointwise score–estimation errors of RODE sampling and SDE sampling become asymptotically equivalent.
> > >
> > > We sincerely appreciate your careful reading and your contribution to improving the correctness, completeness, and clarity of our theoretical results.
> > >
> > > Note: We used ChatGPT for language polishing.

---

### Official Review · Reviewer_67PD · 2025-10-31

**Soundness:** 2
**Presentation:** 3
**Contribution:** 2
**Rating:** 4
**Confidence:** 4

**Summary:**

This paper studies inference-time alignment of diffusion models with respect to general reward functions. The key idea is to fix the process noise and treat the interpolation coefficients between ODE and SDE updates as optimization variables, a scheme we term RODE sampling. These coefficients form a low-dimensional vector that we optimize using Monte Carlo tree search. Experiments first confirm that RODE sampling provides sufficient sample diversity for effective exploration, and then demonstrate strong performance on several text-to-image diffusion alignment tasks.

**Strengths:**

1. Framing the ODE–SDE interpolation factors as optimization variables is an elegant idea. This parameterization is inherently low-dimensional and parameter-efficient, making it a natural fit for bandit-style algorithms such as MCTS.
2. The authors validate the method’s effectiveness across diffusion models with diverse architectures.

**Weaknesses:**

1. Baseline comparisons are limited. It’s unclear how RODE fares against simple baselines, especially best-of-n sampling under pure ODE or pure SDE trajectories. Since the process noise is fixed, the method largely behaves like a local search (though augmented RODE partially mitigates this), raising doubts about its advantage over global search strategies such as best-of-N.
2. The paper lacks an analysis of optimization time, which is critical for assessing practicality. Reporting wall-clock runtime, the number of reward evaluations, MCTS budgets, and how costs scale with image resolution and model size would greatly clarify the method’s efficiency.

**Questions:**

Some works like DNO explore gradient-estimation optimization approach for non-differentaible reward. How do you compare the bandit-type algorithm like MTCS to those gradient-estimation based optimization approach?

DNO: Tang, Zhiwei, et al. "Inference-Time Alignment of Diffusion Models with Direct Noise Optimization." Forty-second International Conference on Machine Learning.

---

> ### Author Response · Authors · 2025-11-21
> **Reply to Reviewer 67PD - Round 1 - (1/3)**
>
> Reply to Reviewer 67PD - Round 1 - (1/3)
>
> Thank you for the thoughtful consideration of the paper and constructive feedback.
>
> **1. Comparison with global search methods such as best-of-N**:
>
> (1) As stated in Sec. 5.1, our experimental setting fixes the initial noise and performs diffusion model alignment by optimizing the denoising trajectory itself.
> Our goal is to fully exploit the generation potential in any single initial noise. The central focus of this study is to demonstrate that, under fixed initial noise, trajectory search in an extremely low-dimensional action space can be used effectively for diffusion model alignment.
>
> (2) Given the limited expressive capacity of 1-D $\eta$-actions, $\eta$-based search behaves more like a *local search* method, and thus it is natural that its absolute best reward may fall short of $\epsilon$-based *global search* methods. However, experiments in Sec. 5.3 and Sec. 5.4 show that even with an extremely low-dimensional action space, our method outperforms several eps-action baselines (e.g., inference-step scaling and Z-Sampling), highlighting its high parameter efficiency.
>
> (3) As noted in Appx. R.3, future work may explore *global–local hybrid strategies*, e.g., using $\epsilon$-actions to identify high-reward regions, followed by $\eta$-actions for refined local search.
>
> (4) Differently, best-of-N selects the top-performing results from multiple initial noises, and therefore it does not fall under our setting. However, since best-of-N is a widely used baseline in diffusion model alignment, we are currently conducting experiments and will include the results in Sec. 5.3 and Appx. K. Note that best-of-N is essentially an $\epsilon$-action method and thus suffers from low parameter efficiency.
>
> Note: We used ChatGPT for language polishment. All technical content was written by ourselves.

---

> ### Author Response · Authors · 2025-11-21
> **Reply to Reviewer 67PD - Round 1 - (2/3)**
>
> Reply to Reviewer 67PD - Round 1 - (2/3)
>
> **2. Analysis of computational budgets and scalability**:
>
> (1) In the earlier version of the manuscript, Sec. 5.3 could not include detailed runtime metrics (wall-clock time, number of reward evaluations, and MCTS budgets) due to space constraints. These metrics were reported in Tab. 13 (Appx. K.2) and Tab. 17 (Appx. N.2). In the revised version, we now include these key efficiency indicators directly in Sec. 5.3 to facilitate clearer comparison across methods.
>
> (2) We appreciate your suggestion that better clarifying the efficiency of $\eta$-based search with scalability beyond parameter efficiency. We are currently conducting experiments and will incorporate the results and extended discussion.
>
> The supplementary experiments are currently underway, and the results will be reported later.
>
> Note: We used ChatGPT for language polishment. All technical content was written by ourselves.

---

> ### Author Response · Authors · 2025-11-21
> **Reply to Reviewer 67PD - Round 1 - (3/3)**
>
> Reply to Reviewer 67PD - Round 1 - (3/3)
>
> **3. Comparison with gradient-estimation-based optimization approach**:
>
> (1) We thank the reviewer for the valuable suggestion. We agree that comparing our method with gradient-estimation-based approaches would more clearly demonstrate the advantages of operating in a low-dimensional action space, and would further highlight the different balances and trade-offs between computational cost and performance under high-dimensional versus low-dimensional action spaces. We are currently conducting experiments and will include the results.
>
> (2) To avoid potential misunderstandings, we would like to clarify the following points.
>
> (I) DNO [1] does explicitly involve gradients of the reward functions, but it adopts gradient estimation to relax the constraint of differentiability. In contrast, our method does not necessitate differentiable reward functions because it is entirely simulation-based and *does not incorporate any reward gradients*.
>
> (II) An intuitive way to understand this distinction: Gradient-based navigation tends to be more beneficial or even necessary to efficiently explore high-dimensional action spaces. However, our action space is extremely low-dimensional, allowing MCTS to cover it effectively with a limited search budget. As a result, gradient computation or estimation is unnecessary for improving navigation efficiency. Moreover, involving gradient estimation in such a low-dimensional space may waste NFE dynamics and NFE reward evaluations, potentially leading to suboptimal performance.
>
> The supplementary experiments are currently underway, and the results will be reported later.
>
> Note: We used ChatGPT for language polishing. All technical content was written by ourselves.

---

### Official Review · Reviewer_8dAj · 2025-10-31

**Soundness:** 2
**Presentation:** 2
**Contribution:** 2
**Rating:** 4
**Confidence:** 4

**Summary:**

This paper developed a randomised ordinary differential equation (RODE) serving as value estimation for the MCTS algorithm. The theoretical guarantee ensures that the Lipschitz constant bounds the mean of the distribution between SDE and RODE.

**Strengths:**

## Presentation: ~45th percentile

The mathematical writing and explanation are intelligible, but they have some issues (see below).

## Soundness: 30th-70th percentile

This paper provides a theoretical guarantee for RODE to estimate the MCTS value, offering a transparent analysis with mathematical arguments.

## Contribution: 40th~75th percentile

The formulation of RODE is new to me and may be useful for further work other than alignment tasks.

## Note:

I hope the AC is aware that the rating is calibrated using estimation of percentiles to reduce evaluation noise effectively.
The rating is simply the mean of the three aspects.

**Weaknesses:**

## Presentation

### Writing

My reading flow was sometimes interrupted due to the lack of connection between sentences/paragraphs. For example, your abstract reads like a list of bullet points without a strong hierarchical presentation; the structure of the first introduction paragraph is

```
for work from a list of literature:

  Introduce literature

  Explain the weakness of the literature, compared to your method.

for work from a list of literature:

  Suggest your solution to address the weakness you mentioned earlier.
```

You can save your readers' effort by putting the weakness and the solution together so that they don’t need to frequently move their pointer while consuming a paragraph of 20 lines.

Such writing is pervasive throughout the manuscript, but I only list the abstract and introduction as they are the most important.

### Literature survey
**Citing Wikipedia** is not considered standard academic practice (and you did it five times). As an encyclopedia, it is a tertiary source that summarises information and is the last type of citation source. You must conduct a thorough literature review and replace this citation with the original peer-reviewed sources that contain the foundational knowledge. I have to admit that these citations leave a terrible impression when reviewing this paper, although I have to make it independent of judgement of other aspects.

### Minors
1. I suggest using Big-O/littel-O notation in equation 10.
2. In Equation 8, it would be better if you juxtapose the original SDE induced by DDIM and the RODE, so your reader can compare them.
3. If your work does not involve a reward encountered during the process (despite the reward at t=0), it is a bad idea to introduce them in Section 3.1 and throughout your entire paper, as I feel confused and effortful when reading the paragraphs from lines 160-179.


## Soundness
**ODE-based baseline:** It is known that DDPM, DDIM, and any score-based SDE can be reformulated as Karra’s SDE [1] bidirectionally. Therefore, these formulations are theoretically equivalent, although the conversion in practice is not very trivial. Assuming the equivalence, you state

> We are the first to extract an RODE sampling from the DDIMs

> To the best of our knowledge, we are the first to model DM alignment as a max-reward control/RL problem.

I expect you to clarify the difference between your and ODE-based value estimation baselines [2, 3] in your paper, and the experimental comparison of them is also missing. I raise a doubt about your performance because the figures in [3] are significantly larger than yours.

[1] Karras, Tero, et al. "Elucidating the design space of diffusion-based generative models"

[2] Ma, Nanye, et al. "Inference-time scaling for diffusion models beyond scaling denoising steps"

[3] PH, Yeh et al. “Training-free Diffusion Model Alignment with Sampling Demons”

**Questions:**

See weaknesses

---

> ### Author Response · Authors · 2025-11-21
> **Reply to Reviewer 8dAj - Round 1 - (1/4)**
>
> Reply to Reviewer 8dAj - Round 1 - (1/3)
>
> Thank you for the thoughtful consideration of the paper and constructive feedback.
>
> **1. Writing**:
>
> In the revised version, we have incorporated the following improvements:
>
> (1) We have updated the writing of the Abstract and Introduction to enhance clarity and coherence.
>
> (2) We have replaced citations to Wikipedia with original peer-reviewed sources.
>
> (3) We now use big-O notation on the right-hand side of Eq. (10).
>
> (4) We place the original SDE alongside Eq. (8) to facilitate easier comparison.
>
> **2. Clarification regarding the first paragraph of Sec. 3.1 (lines 160-179 in the earlier version)**:
>
> (1) We apologize for the misunderstanding caused by our earlier writing. In fact, our *max-encountered-reward modeling* (abbreviated as *max-reward modeling*) does involve intermediate rewards, i.e., rewards of noisy latents throughout the denoising process.
>
> (2) The intended role of the first paragraph in Sec. 3.1:
>
>  (I) To motivate the introduction of max-encountered-reward modeling. We first point out that, under max-final-reward modeling (**not** the full meaning of “max-reward modeling”) with dense rewards, directly or indirectly optimizing only the final reward is suboptimal. This motivates adopting max-encountered-reward modeling, which yields better results.
>
>  (II) To connect notation and avoid ambiguity. We formalize the (pseudo-)final latents used to compute the maximum reward under max-encountered-reward modeling. This removes potential ambiguity in the textual description and links back to the notation $\hat{\phi}(z_{t_i})$ introduced in Sec. 2.3, which is subsequently used in Sec. 3.3(d).
>
> Note: We used ChatGPT for language polishing. All technical content was written by ourselves.

---

> ### Author Response · Authors · 2025-11-21
> **Reply to Reviewer 8dAj - Round 1 - (2/4)**
>
> Reply to Reviewer 8dAj - Round 1 - (2/4)
>
> **3. Difference and comparison between our proposed method and ODE-based value estimation baselines, e.g., zero-order search [1] and Demons [2]**:
>
> Zero-order search, Demons, and our method all evaluate the quality of intermediate latents with ODE-based value estimation and use this information to guide navigation in the action space. However, they differ in several important aspects:
>
> (1) Representation of intermediate rewards:
>
> (I) Zero-order search and Demons both map intermediate latents to intermediate or final samples via an ODE step, and then feed them into a verifier to compute scores. However, neither approach provides a unified mathematical formulation for this mapping. Instead, the ODE-based estimation appears as an implementation detail of each search algorithm rather than a general, interchangeable framework.
>
> (II) Our method introduces a unified latent-reward-shaping formulation (Sec. 2.3, Appx. D.2), where the mapping function $\mathcal{F}$ is modular and replaceable. This allows users to flexibly choose an appropriate implementation of $\mathcal{F}$ depending on computational or task requirements. For example, a fast approximation using DDIM posterior means, or a more accurate estimation like a Heun step (Sec. 4.2 in [2]) followed by a deterministic ODE mapping (Sec. 4.1 in [2]).
>
> Moreover, experiments in Appx. N show that different latent-reward policies substantially affect the achievable best reward due to the nature of max-reward modeling. Designing latent-reward policies that better balance prediction accuracy and computational overhead is an important direction for future work.
>
> (2) Candidate expansion strategy and computational cost:
>
> (I) Zero-order search and Demons operate in a high-dimensional action space and therefore suffer from sparsity issues. To improve navigation efficiency, both methods generate $K$ candidate noises for each pivot, evaluate them separately, and then aggregate the results to identify a promising direction.
>
> (II) Our method navigates in a low-dimensional action space. During MCTS expansion, we select only one node and sample a single $\eta$ to generate one candidate latent.
>
> Because the candidate-expansion mechanisms differ fundamentally, the corresponding computational costs (e.g., NFE dynamics, NFE intermediate reward estimation) also differ and depend on the implementation of ODE steps and latent reward policies. Direct comparison is therefore non-trivial. We are running experiments under a unified setting and will report results.
>
> (3) Treatment of process noise and optimization paradigm:
>
> (I) Zero-order search and Demons are stochastic optimization methods: the noise used in each run of the search is resampled every time. Specifically, zero-order search samples new candidate noise from the neighborhood of the current pivot (Sec. 3.2 in [1]); Demons also resamples the noise used for interpolation at every step (Sec. 4.2 in [2]).
>
> (II) Our method avoids the high variance inherent in stochastic control by pre-sampling and fixing a set of process-noise realizations, which are then reused throughout the search (Appx. F.3.3). Consequently, our method is not a stochastic optimization method. (Further discussion is provided in *4. Clarification of our novelty*)

---

> ### Author Response · Authors · 2025-11-21
> **Reply to Reviewer 8dAj - Round 1 - (3/4)**
>
> Reply to Reviewer 8dAj - Round 1 - (3/4)
>
> (4) Action-space design:
>
>  (I) Zero-order search and Demons both adopt high-dimensional noise as actions. This leads to difficult action-space navigation, resulting in poor parameter efficiency.
>
> (II) Our method deliberately avoids high-dimensional eps-actions and instead uses a 1-D eta as the action variable. Such a low-dimensional action space can be effectively explored with a limited number of MCTS expansions, making it particularly suitable for bandit-style search methods.
>
> The experiments in Sec. 5.3 and Sec. 5.4 show that even with an extremely low-dimensional action space, our method outperforms several eps-action baselines (e.g., inference-step scaling), highlighting its high parameter efficiency.
>
> (5) Modeling formulation:
>
> (I) Zero-order search and Demons both aim to maximize the reward of the final generated sample. They enforce a full denoising process and therefore fall under the max-final-reward modeling framework.
>
> (II) Our method treats pseudo-final latents derived within the latent-reward policies as valid samples for reward computation. Thus, our method does not require samples to complete the entire denoising trajectory and belongs to the max-encountered-reward modeling paradigm.
>
> This also clarifies our statement: “To the best of our knowledge, we are the first to model DM alignment as a max-reward control/RL problem.”
>
> The supplementary experiments are currently underway, and the results will be reported later.
>
> [1] Ma, Nanye, et al. "Inference-time scaling for diffusion models beyond scaling denoising steps"
>
> [2] PH, Yeh et al. “Training-free Diffusion Model Alignment with Sampling Demons”
>
> Note: We used ChatGPT for language polishing. All technical content was written by ourselves.

---

> ### Author Response · Authors · 2025-11-21
> **Reply to Reviewer 8dAj - Round 1 - (4/4)**
>
> Reply to Reviewer 8dAj - Round 1 - (4/4)
>
> **4. Clarification of our novelty**:
>
> We appreciate the reviewer’s concern regarding the novelty of our method and the comment “We are the first to extract an RODE sampling from the DDIMs.” Below, we clarify the conceptual and technical distinctions between Demons [1] and our approach.
>
> (1) Difference in the role of scalar variables in Demons [1] and in our method:
>
> (I) After randomly sampling a set of candidate noise vectors {$z^{(k)}$}$^K_{k = 1}$, Demons [2] form a new noise vector $z^*$ by weighting them with a set of scalars {$b_k$}$^K_{k=1}$ (Sec. 4.2 in [2]). Importantly, each $b_k$ is *computed adaptively* according to the relative advantage of $z^{k}$ among {$z^{(k)}$}$_{k = 1}^K$ rather than from an optimization process.
>
> (II) Differently, we pre-sample and fix a set of process-noise directions, and the MCTS optimization explicitly searches for the scalar sequence {$\eta_{t_i}$}$_{i = 0}^{n - 1}$. The scalars in our formulation are not adaptively computed; instead, they are variables that are being optimized.
>
> (2) Clarification regarding RODE sampling:
>
> (I) The candidate noise set of Demons [2] {$z^{(k)}$}$_{k = 1}^K$ is resampled at every expansion step. From a continuous-time perspective, this repeated re-sampling induces a stochastic term — a Wiener increment $\mathrm{d} w$ — when writing its differential-equation formulation. Consequently, the continuous-time interpretation of Demons corresponds to an SDE sampling process.
>
> (II) Differently, we pre-sample and fix a set of process-noise realizations and reuse them during search instead of resampling (Appx. F.3.3). This replaces the stochastic term $g(t)  \mathrm{d}w$ with an anisotropic drift term $\omega(t) \eta(t) h(t) \mathrm{d}t$ (Eq. (8)). With process noise fixed, both $\omega(t)$ and $h(t)$ are deterministic, and the only stochastic component is the scalar-valued process $\eta(t)$. Consequently, the resulting dynamics contain no Wiener-process term, making the formulation a random ODE (RODE) rather than an SDE.
>
> To the best of our knowledge, prior works have not formalized sampling via pre-fixed process-noise realizations as an RODE for inference-time alignment.
>
> (3) Novelty and significance of this study:
>
> The central focus of this study is to demonstrate that, with fixed initial noise, trajectory search in an extremely low-dimensional action space can be effectively used for diffusion model alignment. This also represents the first attempt at RODE sampling of diffusion models.
>
> Given the limited expressive capacity of 1-D $\eta$-actions, $\eta$-based search behaves more like a *local search* method, and thus it is natural that its absolute best reward may fall short of $\epsilon$-based *global search* methods. As noted in Appx. R.3, future work may explore *global–local hybrid strategies*, e.g., using $\epsilon$-actions to identify high-reward regions, followed by $\eta$-actions for refined local search.
>
> Nevertheless, the difference between the *relative improvement* achieved by $\eta$- and $\epsilon$-based methods — when viewed through the lens of their vastly different parameter dimensionalities — reveals a broader conceptual insight: *parameter efficiency* plays a crucial role in search-based alignment.
>
> For these reasons, we believe that this study offers substantial novelty and significance.
>
> [1] PH, Yeh et al. “Training-free Diffusion Model Alignment with Sampling Demons”
>
> Note: We used ChatGPT for language polishing. All technical content was written by ourselves.

---

> > ### Comment · Reviewer_8dAj · 2025-11-25
> >
> > There may be a misunderstanding regarding my comment on the Big O notation. Mathematically, there is a strict difference between $\mathcal{O}(ax + b)$ and $ax + \mathcal{O}(b)$. However, I acknowledge that in this context, it supports the claim that RODE and SDE have a minor distributional gap, so I do not consider this a critical issue holding back the paper.
> >
> > While you are preparing the experimental results, I strongly encourage you to prioritise improving the presentation. This significantly impacts the paper's acceptance chances, as presentation accounts for 1/3 portion of my assessment.
> >
> > While the mathematical symbols are legible, the paper currently lacks intuition. The text often describes what the math is doing, but fails to explain why. In my second reading, I still found it difficult to grasp the underlying logic behind the mechanism, even though I could follow the derivation steps. If the paper were written more coherently, connecting the symbols to the concepts, it would be much more sound and impactful for the reader.

---

> > > ### Author Response · Authors · 2025-12-02
> > > **Reply to Reviewer 8dAj - Round 2 - (1/2)**
> > >
> > > Reply to Reviewer 8dAj - Round 2 - (1/2)
> > >
> > > **1. Usage of the Big-O notation**:
> > >
> > > Thank you for pointing this out.
> > > We have revised the bound to the form $ax + \mathcal{O}(b)$, which cleanly separates the main contribution from the residual term and highlights that the primary discrepancy between RODE and SDE trajectories arises from the
> > > $\eta$-bias:
> > >
> > > $$
> > > W_1\big(p^{(S)},p^{(R)}\big)
> > > \leq
> > > \sum_{k=0}^{n-1} M_{t_{k+1}} \Big( \omega_{t_k} C_{\epsilon,t_k} \mathbb{E}[\Delta\eta_{t_k}] \Big)
> > > +
> > > \mathcal{O}\Big( \sum_{k=0}^{n-1} M_{t_{k+1}} \gamma_{t_k} \Big).
> > > $$
> > >
> > > We also added an additional explanation in Appendix F.3.1.
> > >
> > > Note: We used ChatGPT for language polishing.

---

> > > > ### Author Response · Authors · 2025-12-02
> > > > **Reply to Reviewer 8dAj - Round 2 - (2/2)**
> > > >
> > > > Reply to Reviewer 8dAj - Round 2 - (2/2)
> > > >
> > > > **2. Presentation**:
> > > >
> > > > In our earlier version, we had already described the motivation of our approach in the introduction, provided textual intuition preceding each proposition (Sec. 3.2), and included further discussion and visualization in Sec. 5.2, Appx. E, and Appx. I. The flow and visualization were positively noted by Reviewers Rs8u and cCHK.
> > > >
> > > > In the revised version, we have further clarified the exposition:
> > > > (I) We refined part of the introduction, moved several intuition-related visualizations and discussions from the appendix into the main paper (Sec. 3.1);
> > > > (II) We expanded the explanatory text surrounding the propositions in both the main paper (Sec. 3.2) and the appendix (Appx. F).
> > > > All revisions are highlighted in red in the updated manuscript.
> > > >
> > > > We hope that these improvements make the underlying logic behind the necessary equations clearer and more accessible to readers.
> > > >
> > > > Note: We used ChatGPT for language polishing.

---

### Official Review · Reviewer_Rs8u · 2025-11-05

**Soundness:** 3
**Presentation:** 3
**Contribution:** 2
**Rating:** 4
**Confidence:** 3

**Summary:**

This paper proposes a novel inference-time scaling method for diffusion model alignment via modifying the denoising trajectory with the strength of the added noise as the exploration action (while the added noise is fixed). Compared to the commonly used noise/eps as action (which is high-dimensional), the proposed method adopts such a low-dimensional action that can be easily integrated into UCB-based Monte Carlo Tree Search (MCTS) for efficient online exploration. The authors show (both theoretically and empirically) that this new action space admits less variance than the noise/eps as actions paradigm. This method is plug-and-play for various reward signals (including non-differentiable ones).

**Strengths:**

+ The formulation of using the perturbation strength as an action while fixing the added noise is novel and intriguing. I like the explanation and the visualization of the reduced variance.
+ I am a bit surprised the method, while simple conceptually, can do inference-time scaling pretty well. A major challenge in visual diffusion model fine-tuning is the high-dimensional action space. Since the action space is low-dimensional, it might open up a door for other online tuning methods, such as policy optimization (also optimizing the diffusion model itself).

**Weaknesses:**

+ I am not particularly impressed by the claimed advantage of parameter efficiency. Can the authors point out scenarios where it is critical, given that eps-based action space still performs better in some cases (maybe not for MCTS)?
+ I am curious if the authors can provide more evidence that the proposed method alleviates reward hacking compared to eps-action alternatives. This will make the contribution of this work significant.

I am willing to raise my score after seeing the authors' response.

**Questions:**

See weaknesses

---

> ### Author Response · Authors · 2025-11-21
> **Reply to Reviewer Rs8u - Round 1 - (1/2)**
>
> Reply to Reviewer Rs8u - Round 1 - (1/2)
>
> Thank you for the thoughtful consideration of the paper and constructive feedback.
>
> **1. Scenarios where parameter efficiency (PE) is critical.**:
>
> (1) **When and why eps-based actions appear to perform better**:
>
>  We acknowledge that eps-based action spaces can sometimes yield superior empirical performance. This improvement largely stems from the substantially higher dimensionality of eps-actions: each eps-action contains many parameters, which increases the action’s expressiveness and allows the search to explore a wider region of the solution space (i.e., it exhibits higher variance as stated in our paper). Moreover, inference-time alignment methods often retain the best outcome encountered during the search procedure; under such a mechanism, a higher-variance search may discover improved results, making eps-based approaches appear advantageous.
>
> (2) **Why we report PE**:
>
> We report the PE of both eps- and eta-based methods to reveal potential parameter redundancy in the eps-action representation. Concretely, in Sec. 5.3 (Tab. 3), we show that scaling DDPM (essentially an eps-based search) attains a best reward that is **inferior** to that of three distinct eta-based methods. This observation suggests that:
>
>  (I) Increasing parameters does not necessarily lead to better performance;
>
>  (II) The relative improvement attained by the eps-based search is only on the order of $\sim 2\times$ compared to the eta-based methods, while requiring roughly $\sim 16\text{k}\times$ more parameters, which strongly implies substantial parameter redundancy;
>
>  (III) The blind high-dimensional search is computationally inefficient. Further detailed analysis is provided in Appx. K.2.
>
> (3) **Scenarios where PE is critical**:
>
> (I) Resolution scaling. The number of parameters in an eps-action grows significantly with image resolution. As resolution increases, the low PE of eps-based searches leads to a proliferation of redundant parameters, which does not necessarily yield higher rewards and incurs heavy computational and memory costs.
>
> (II) Bandit-style and gradient-based procedures. Methods that require evaluating or differentiating with respect to each action parameter (e.g., bandit-style searches, gradient-based searches, or computing theoretically optimal controls based on local linearization) must "probe" or compute gradients per parameter. Low PE in these settings causes excessive and often prohibitive computational overhead. In many cases, gradient computation over a very high-dimensional action becomes intractable.
>
> In summary, considering PE in trajectory search methods is beneficial: it enables the design of alignment techniques that achieve comparable or better performance while substantially reducing computational and memory requirements.
>
> The supplementary experiments are currently underway, and the results will be reported later.
>
> Note: We used ChatGPT for language polishing.

---

> ### Author Response · Authors · 2025-11-21
> **Reply to Reviewer Rs8u - Round 1 - (2/2)**
>
> Reply to Reviewer Rs8u - Round 1 - (2/2)
>
> **2. Can our proposed method alleviate reward hacking compared to eps-action alternatives**:
>
> We apologize for the confusion our earlier writing may have caused.
>
> In our original manuscript, the term *reward hacking* referred specifically to the phenomenon observed in max-reward modeling, where pseudo-final latents from early denoising steps are often blurry or contain high-frequency artifacts. These latents can serve as "hack samples" for certain reward functions (e.g., LAPV). To mitigate this issue, we restrict the earliest denoising depth at which a latent can be designated as the final one. This phenomenon is unrelated to whether the action space is eps-based or eta-based.
>
> We assume that you are instead asking whether, in the broader context of aligning diffusion models with reward functions, eta-actions may alleviate reward hacking compared with eps-actions. We believe the answer is yes. Without additional constraints, higher reward values encountered during search often strongly correlate with reward hacking. Since eps-actions span a substantially larger search space, they are more likely to enter regions associated with reward hacking. In contrast, eta-actions yield a more confined search space. Therefore, because vanilla diffusion sampling predominantly produces natural images, eta-based sampling deviates less from vanilla trajectories. As a result, the samples that achieve the best rewards from eta-based search are more likely to be natural images than hacked ones.
>
> We thank the reviewer for raising this insightful point. We will revise the manuscript to clarify that our proposed strategy targets the reward hacking that arises from max-reward modeling. Moreover, motivated by your suggestion, we will include a discussion on the potential advantages of eta-actions over eps-actions for reducing reward hacking.
>
> Note: We used ChatGPT for language polishing.

---

### Author Response · Authors · 2025-12-02
**A Summary for AC (1/5)**

A Summary for AC (1/5)

We sincerely thank all reviewers for their constructive feedback on the writing, theory, and experimental evaluation of our work.

We also thank the Area Chair for the time and effort dedicated to handling this submission.

To assist your decision process, we provide a concise summary of the discussion and revisions below.

**1. What have we done?**

(1) A brief summary of this paper:

This paper proposes a novel inference-time alignment framework for diffusion models (DMs) based on **random ODE (RODE)** sampling. Specifically, by fixing the process noise and optimizing the perturbation strength, we obtain a **1-D action space**, which integrates naturally with Monte Carlo tree search. We can thus perform trajectory search in a gradient-free manner, therefore supporting non-differentiable rewards. Then, we model the inference process as a Markov decision process with dense rewards, and formulate DM alignment as a **max-encountered-reward** optimal control problem. Aside from algorithmic design, we also provide theoretical guarantees and empirical evidence on stability and score-estimation error to support and validate our method, showing that while the RODE trajectories exhibit low variance, they remain close to the pure SDE trajectories under certain assumptions.

Extensive experiments demonstrate sufficient sample diversity, stronger performance above traditional inference-step scaling, and significantly higher parameter efficiency compared with high-dimensional action approaches.
Our method can be plug-and-play integrated into any multi-step inference DMs.

(2) Our main contributions:

(I) We propose a novel parameter-efficient inference-time scaling framework for aligning DMs with any Lipschitz continuous reward functions (unnecessarily differentiable) by altering the denoising trajectory;

(II) **We are the first to** extract an RODE sampling from the DDIM's SDE sampling, which provides controllable randomness with lower variance;

(III) **We are the first to** formulate the inference-time DM alignment as a max-encountered-reward optimal control problem, and solve it using MCTS with augmented RODE-based scaling;

(IV) **We are the first to** scale DMs with 1-D actions.

---

> ### Author Response · Authors · 2025-12-02
> **A Summary for AC (2/5)**
>
> A Summary for AC (2/5)
>
> **2. What do reviewers appreciate?**
>
> (1) **Novelty**.
> Reviewer Rs8u found our approach **intriguing**, **conceptually simple**, and **empirically effective**, while Reviewer 67PD described the method as **elegant**.
>
> (2) **Soundness**.
> Reviewers Rs8u, 8dAj, and cCHK all agreed that the method is **theoretically grounded**.
>
> (3) **Presentation**.
> Reviewers Rs8u, 8dAj, and cCHK also expressed positive feedback regarding the clarity of our writing, the coherence of the flow, and the usefulness of the visualizations.

---

> > ### Author Response · Authors · 2025-12-02
> > **A Summary for AC (3/5)**
> >
> > A Summary for AC (3/5)
> >
> > **3. What are reviewers concerned about and how do we respond?**
> >
> > (1) **Theoretical soundness**.
> >
> > (i) Reviewer 8dAj helped us refine the formulation of Proposition 2.
> >
> > (ii) Reviewer cCHK raised important questions regarding the strength of Propositions 2 and 3. We have responded with detailed explanations and, under stronger assumptions, derived an enhanced proposition that strengthens the theoretical guarantees.
> >
> > (2) **Clarifying misunderstandings**.
> >
> > Some aspects of our initial writing led to misunderstandings among reviewers — for example, regarding mitigation of reward hacking (Reviewer Rs8u), the adoption of intermediate rewards (Reviewer 8dAj), and why support non-differentiable rewards (Reviewer 67PD). We clarified these points in the discussion and revised the corresponding parts of the manuscript (summarized in Summary 4).
> >
> > (3) **Clarification of key concepts**.
> >
> > (i) We clarified that "max-reward" refers to max-encountered reward, not max-final reward. We have clarified this in the revised version.
> > (ii) We addressed the differences between our method and Demons [1] (Reviewer 8dAj), DNO [2] (Reviewer 67PD), and FK steering [3] (Reviewer cCHK), and added corresponding discussion and citations to the manuscript.
> >
> > (4) **Clarification regarding global search capabilities and absolute best rewards**.
> >
> > Please refer to Summary 5 (3) for details.
> >
> > (5) **Experiments**.
> >
> > (i) We have moved the computational cost analysis (NFE, runtime, etc.), previously in the appendix, into the main paper. (Suggested by Reviewer 67PD)
> >
> > (ii) We have added important inference-time alignment baselines — pure ODE/SDE best-of-N sampling. (Suggested by Reviewer 67PD)
> >
> > (iii) Regarding parameter-efficiency experiments on cost scaling with image resolution and model size (Reviewer Rs8u, 67PD), we now report runtime for both SD and SDXL. We note that comparing different architectures directly would be unfair. Therefore, we are conducting additional scaling experiments on PixArt-Alpha-XL 512/1024 (same architecture, different resolution/model size) and will include the results.
> >
> > (iv) For comparisons with Demons [1] (Reviewer 8dAj), we believe the existing eps-action baselines already demonstrate the limitations of high-dimensional action spaces and their low parameter efficiency. We further discussed the differences with Demons in the revised manuscript. If the Area Chair considers additional experiments necessary, we will provide them.
> >
> > [1] PH, Yeh et al. Training-free Diffusion Model Alignment with Sampling Demons
> >
> > [2] Tang, Zhiwei, et al. Inference-Time Alignment of Diffusion Models with Direct Noise Optimization. Forty-second International Conference on Machine Learning.
> >
> > [3] Raghav Singhal, Zachary Horvitz, Ryan Teehan, Mengye Ren, Zhou Yu, Kathleen McKeown, and Rajesh Ranganath. A general framework for inference-time scaling and steering of diffusion models. arXiv preprint arXiv:2501.06848, 2025.

---

> ### Author Response · Authors · 2025-12-02
> **A Summary for AC (4/5)**
>
> A Summary for AC (4/5)
>
> **4. Revision of the manuscript.**
>
> (1) Abstract and Introduction.
>
> We reconstructed and polished the presentation of the abstract and the introduction to improve clarity and coherence.
>
> (2) Section 3.1.
>
> (I) We clarified that "*max-reward*" refers to *max-encountered reward* to eliminate potential ambiguity.
>
> (II) We moved the geometric interpretation of our modeling from the appendix to this section to better convey the intuition behind our method design.
>
> (3) Section 3.2.
>
> (I) According to reviewer 8dAj, we rewrite proposition 2 as a *main term + residual term* formulation, with additional explanation provided in Appendix F.3.1.
>
> (II) We added a more complete description of the notation for Proposition 2.
>
> (III) We now explicitly refer to Appendix F.3.3 in the main paper for the strengthened version of Proposition 2.
>
> (4) Section 5.
>
> (I) We improved the overall presentation and formatting of Section 5.
>
> (II) We moved the experiments originally in the appendix — comparison between different combinations of reward shaping, MDP modeling, and value-update policies — to the main paper (Sec. 5.3), as max-encountered-reward modeling is one of our key contributions.
>
> (5) Appendix D.2.1.
>
> We clarified the relationship between our latent reward shaping and existing ODE-based value-estimation methods.
>
> (6) Appendix E.4.
>
> We added comparisons with Demons and methods designed to handle non-differentiable rewards, which are summarized in Appendix E.4..
>
> (7) Appendix J.
>
> (I) We added the introduction and explanation of newly added baseline methods in Appendix J.1.
>
> (II) We provided additional clarification regarding the global-search capabilities of our method and the absolute best rewards to Appendix J.2.

---

> ### Author Response · Authors · 2025-12-02
> **A Summary for AC (5/5)**
>
> A Summary for AC (5/5)
>
> **5. Additional Remarks .**
>
> (1) **Our setting**.
>
> (i) Our setting fixes the initial noise and performs diffusion model alignment by optimizing the denoising trajectory.
>
> (ii) Best-of-N methods search over multiple initial noises, and therefore fall outside our setting.
>
> (2) **Our goal**.
>
> (i) Our primary goal is to fully exploit the generative potential of a single initial noise. We demonstrate that, even under fixed initial noise, trajectory search in an extremely low-dimensional action space can effectively align diffusion models.
>
> (ii) This work represents the first attempt at RODE sampling and diffusion-model alignment using 1-D actions, and highlights the broader conceptual importance of parameter efficiency in inference-time scaling.
>
> (3) **Local/global search and absolute best rewards**:
>
> (I) Given the limited expressive capacity of 1-D $\eta$-actions, $\eta$-based search behaves more like a *local search* method, and thus it is natural that its absolute best rewards may fall short of $\epsilon$-based *global search* methods. However, experiments in Sec. 5.4 and Sec. 5.5 show that even with an extremely low-dimensional action space, our method outperforms several eps-action baselines (e.g., inference-step scaling and Z-Sampling), highlighting its high parameter efficiency.
>
> (II) As noted in Appx. Q.3, future work may explore *global–local hybrid strategies*, e.g., using $\epsilon$-actions to identify high-reward regions, followed by $\eta$-actions for refined local search.
>
> (III) Nevertheless, the difference between the *relative improvement* achieved by $\eta$- and $\epsilon$-based methods — when viewed through the lens of their vastly different parameter dimensionalities — reveals a broader conceptual insight: *parameter efficiency* plays a crucial role in search-based alignment.
>
> (4) **Novelty regarding intermediate-reward estimation**.
>
> We do **not** propose a new intermediate-reward estimator. Instead, we introduce a unified latent-reward-shaping formulation that encompasses a broad class of existing ODE-based value/intermediate-reward estimation methods. The mapping $\mathcal{F}$ is modular and replaceable, allowing practitioners to choose an appropriate instantiation based on computational or task requirements.
>
> (5) **Clarification on max-reward**.
>
> "*Max-reward*" refers to *max-encountered reward*, **not max-final reward**. We have clarified this in the revised version. Accordingly, we are the first to formulate inference-time diffusion model alignment as a max-reward optimal control problem, and to solve it using MCTS with augmented RODE-based scaling.

---

### Meta-Review · Area_Chair_Spuw · 2026-01-07

**Summary:**

The main concern that informed my suggested decision is whether parameter efficiency is important, while empirical evidence shows that eps-action methods are almost always better or on par. In the response to Reviewer Rs8u, the authors said that there are two main scenarios where this can be critical:

*1. Resolution scaling as the number of parameters in an eps-action grows significantly with image resolution. As resolution increases, this can incur heavy computational and memory costs.*

However, the paper doesn't present quantified evidence for additional computational cost. Table 5 shows that, compared to eta-action space methods (the proposed approach), all eps-action methods achieve comparable or better reward with comparable or lower time cost actually. In addition, in the summary to AC, the authors stated *Regarding parameter-efficiency experiments on cost scaling with image resolution and model size (Reviewer Rs8u, 67PD), we now report runtime for both SD and SDXL. We note that comparing different architectures directly would be unfair. Therefore, we are conducting additional scaling experiments on PixArt-Alpha-XL 512/1024 (same architecture, different resolution/model size) and will include the results.*. Where are these results? They don't seem to appear in discussion and the revised manuscript.

Also, resolution scaling doesn't just introduce problems to inference-time scaling methods, but diffusion in general. I think the more common practice for super high resolution nowadays is latent diffusion. The paper not only didn't compare to latent diffusion, but didn't discuss it at all. I think a rough system-wise comparison to support the discussion would've been fine.

*2. Bandit-style and gradient-based procedures. Methods that require evaluating or differentiating with respect to each action parameter (e.g., bandit-style searches, gradient-based searches, or computing theoretically optimal controls based on local linearization) must "probe" or compute gradients per parameter. Low PE in these settings causes excessive and often prohibitive computational overhead. In many cases, gradient computation over a very high-dimensional action becomes intractable.*

Again, this lacks supporting evidence.

**Reviewer Concerns:**

Outstanding:
- The main outstanding concern is why parameter efficiency is important, as that's the main claim of this paper. This appears to be not fully addressed.

Addressed:
- Refinement of Propositions: Reviewer 8dAj suggested refining Proposition 2 to better isolate the primary source of error. The authors rewrote the proposition using a "main term + residual term" formulation to clearly show that the η-bias is the dominant discrepancy between RODE and SDE trajectories.
- Reviewer cCHK questioned the strength of Propositions 2 and 3, noting they initially only established "boundedness" rather than convergence. The authors added Appendix F.3.3, which introduces stronger assumptions under which the discrepancy and score-estimation error vanish, making the theoretical guarantees nontrivial.
- To eliminate ambiguity, the authors clarified in the revised manuscript that "max-reward" specifically refers to the max-encountered reward within an episode, rather than the more common max-final reward.
- The authors addressed concerns regarding how their method differs from existing paradigms like Demons, DNO, and FK Steering. Specifically, they clarified that unlike Demons (which resamples noise and behaves as an SDE), their method fixes process-noise realizations to operate as a Random ODE (RODE), which avoids the high variance of stochastic control.
- Reviewer Rs8u sought evidence that the method alleviates reward hacking. The authors clarified that their specific strategy—using a depth limit (τ)—mitigates hacking caused by blurry early-stage samples, and argued that 1-D η-actions are less prone to hacking than high-dimensional actions because they deviate less from natural image manifolds.
- Geometric Intuition: The authors moved the geometric interpretation of their MDP modeling (the directed acyclic graph shortcuts) from the appendix to Section 3.1 to better convey the underlying intuition of treating pseudo-final samples as valid.

**Reviewer Scores:**

In my prediction:
- Reviewer Rs8u would have maintained a 4 as the explanation about why parameter efficiency is important was not supported by evidence.
- Reviewer 8dAj would've maintained a 4 as they are not fully satisfied with the presentation.
- Reviewer 67PD would've +1 as the questions are answered in detail.
- Reviewer cCHK would've +1 as the questions are answered in detail.

---

### Decision · Program_Chairs · 2026-01-26

Reject